# AVOID: Alleviating VAE's Overestimation in Unsupervised OOD Detection

## Abstract

Deep generative models (DGMs) aim at characterizing the distribution of the training set by maximizing the marginal likelihood of inputs in an unsupervised manner, making them a promising option for unsupervised out-of-distribution (OOD) detection. However, recent works have reported that DGMs often assign higher likelihoods to OOD data than in-distribution (ID) data, *i.e.*, **overestimation**, leading to their failures in OOD detection. Although several pioneer works have tried to analyze this phenomenon, and some VAE-based methods have also attempted to alleviate this issue by modifying their score functions for OOD detection, the root cause of the *overestimation* in VAE has never been revealed to our best knowledge. To fill this gap, this paper provides a thorough theoretical analysis on the *overestimation* issue of VAE, and reveals that this phenomenon arises from two aspects: 1) the improper design of prior distribution; 2) the gap of dataset entropy-mutual integration (sum of dataset entropy and mutual information gap terms) between ID and OOD datasets. Based on these findings, we propose a novel score function to **A**lleviate **V**AE's **O**verestimation **I**n unsupervised OOD **D**etection, named **"AVOID"**, which contains two novel techniques, specifically *post-hoc prior* and *dataset entropy-mutual calibration*. Experimental results verify our theoretical analysis, demonstrating that the proposed method is effective in alleviating *overestimation* and improving unsupervised OOD detection performance.

## 1 Introduction

The detection of out-of-distribution (OOD) data, *i.e.*, identifying data that differ from the in-distribution (ID) training set, is crucial for ensuring the reliability and safety of real-world applications (Hendrycks & Gimpel, 2017; Goodfellow et al., 2015; Nguyen et al., 2015; Wei et al., 2022b). While the most commonly used OOD detection methods rely on supervised classifiers (Alemi et al., 2018; Liu et al., 2020; Wei et al., 2022a; Huang et al., 2022; Wei et al., 2022c; Yu et al., 2023; Galil et al., 2023), which require labeled data, this paper focuses on designing an unsupervised OOD detector. **Unsupervised OOD detection** refers to the task of designing a detector, based solely on the unlabeled training data, that can determine whether an input is ID or OOD (Ren et al., 2019a; Serrà et al., 2020; Xiao et al., 2020; Maaløe et al., 2019; Floto et al., 2023; Havtorn et al., 2021; Li et al., 2022). This unsupervised approach is more practical for real-world scenarios where the data lack labels.

Deep generative models (DGMs) are a highly attractive option for unsupervised OOD detection. DGMs, mainly including the auto-regressive model (van den Oord et al., 2016; Salimans et al., 2017), flow model (Dinh et al., 2017; Kingma & Dhariwal, 2018), diffusion model (Ho et al., 2020), generative adversarial network (Goodfellow et al., 2020), and variational autoencoder (VAE) (Kingma & Welling, 2014), are designed to model the distribution of the training set by explicitly or implicitly maximizing the likelihood estimation of $p(\boldsymbol{x})$ for its input $\boldsymbol{x}$ without category label supervision or additional OOD auxiliary data. They have achieved great success in a wide range of applications, such as image and text generation. Since generative models are promising at modeling the distribution of the training set, they could be seen as an ideal unsupervised OOD detector, where the likelihood of the unseen OOD data output by the model should be lower than that of the in-distribution data.

Unfortunately, developing a flawless unsupervised OOD detector using DGMs is not as easy as it seems to be. Recent experiments have revealed a paradoxical phenomenon that directly applying the likelihood of generative models as an OOD detector can result in **overestimation**, *i.e.*, **DGMs**

**assign higher likelihoods to OOD data than ID data** (Ren et al., 2019a; Serrà et al., 2020; Havtorn et al., 2021; Li et al., 2022). For instance, a generative model trained on the FashionMNIST dataset could assign higher likelihoods to data samples from the MNIST dataset (OOD) than those from the FashionMNIST dataset (ID), as shown in Figure 5(a). Since OOD detection can be viewed as a verification of whether a generative model has learned to model the distribution of the training set accurately, the counterfactual phenomenon of *overestimation* not only poses challenges to unsupervised OOD detection but also raises doubts about the generative model's fundamental ability in modeling the data distribution. Therefore, it highlights the need for developing more effective methods for unsupervised OOD detection and, more importantly, a more thorough understanding of the reasons behind the *overestimation* in deep generative models.

To develop more effective methods for unsupervised OOD detection, some approaches modified the likelihood to new score functions with empirical assumptions, such as likelihood-ratio methods (Havtorn et al., 2021; Li et al., 2022) and ensemble approaches (Choi et al., 2018). While these methods, particularly the VAE-based ones (Li et al., 2022), have achieved state-of-the-art (SOTA) performance in unsupervised OOD detection, none of them provides a clear explanation for the *overestimation* issue. To gain insight into the *overestimation* issue in DGMs, some works attribute the *overestimation* to the typical set hypothesis, where ID and OOD data have overlapping supports (Choi et al., 2018; Wang et al., 2020; Morningstar et al., 2021). However, Zhang et al. (2021) prove that this hypothesis is not correct and appeal for more future works on analyzing the estimation error of the DGMs. In contrast to the analysis for exact-marginal-likelihood DGMs like flow and auto-regressive models (Nalisnick et al., 2019a; Kirichenko et al., 2020), VAE utilizes a variational evidence lower bound (ELBO) of the likelihood, making it difficult to analyze. Overall, the reasons behind the *overestimation* issue of VAE are still not fully understood, especially for a trained VAE in practice without assuming the model distribution could exactly converge to the true data distribution.

In this paper, we try to address the research gap by providing a theoretical analysis of VAE's *overestimation* issue in unsupervised OOD detection and corresponding methods for alleviating it. Our contributions can be summarized as follows:

1. Through theoretical analyses, we are the first to identify two factors that cause the *overestimation* issue of VAE: 1) the improper design of prior distribution; 2) the intrinsic gap of ***dataset entropy-mutual integration*** (sum of the dataset entropy and the mutual information gap terms between the inputs and latent variables) between ID and OOD datasets;

2. Focused on these two discovered factors, we propose a new score function, named **"AVOID"**, to alleviate the *overestimation* issue by ensembling two remedies for each factor: i) *post-hoc prior* (PHP) for the improper design of prior distribution, *i.e.*, replacing the Gaussian prior $p(\boldsymbol{z})$ with the approximated ID training set's aggregated posterior distribution $\hat{q}_{id}(\boldsymbol{z})$ in the ELBO; ii) *dataset entropy-mutual calibration* (DEC) for the second factor, which employs a data compression method to calibrate the gap of entropy-mutual integration between ID and OOD datasets and designs a regularized scale to balance the weight of PHP method;

3. Extensive experiments, including the commonly acknowledged "harder" tasks, demonstrate our method's theoretically guaranteed effectiveness in improving the performance of VAE-based methods on unsupervised OOD detection.

## 2 PRELIMINARIES

### 2.1 UNSUPERVISED OUT-OF-DISTRIBUTION DETECTION

**Problem statement.** While deploying a machine learning system, it is possible to encounter inputs from unknown distributions that are semantically (*e.g.*, category) and/or statistically (*e.g.*, data complexity) different from the training data, and such inputs are referred to as OOD data (Choi et al., 2018; Serrà et al., 2020). Processing OOD data could potentially introduce critical errors that compromise the safety of the system (Hendrycks & Gimpel, 2017). Thus, the OOD detection task is to identify these OOD data, which could be seen as a binary classification task: determining whether an input $\boldsymbol{x}$ is more likely ID or OOD. An ID-OOD classifier $D(\boldsymbol{x})$ could be formalized as a level-set estimation:

$$D(\boldsymbol{x}) = \begin{cases} \text{ID}, & \text{if} \quad \mathcal{S}(\boldsymbol{x}) > \lambda, \\ \text{OOD}, & \text{if} \quad \mathcal{S}(\boldsymbol{x}) \leq \lambda, \end{cases} \tag{1}$$

where $\mathcal{S}(\boldsymbol{x})$ denotes the score function, *i.e.*, **OOD detector**, and the threshold $\lambda$ is commonly chosen to make a high fraction (*e.g.*, 95%) of ID data correctly classified (Wei et al., 2022c). In conclusion, OOD detection aims at designing the $\mathcal{S}(\boldsymbol{x})$ that could assign higher scores to ID data than OOD data.

**Setup.** Denoting the input space with $\mathcal{X}$, an *unlabeled* training dataset $\mathcal{D}_{\text{train}} = \{\boldsymbol{x}_i\}_{i=1}^{N}$ containing of $N$ data points can be obtained by sampling *i.i.d.* from a data distribution $\mathcal{P}_{\mathcal{X}}$. Typically, we treat the $\mathcal{P}_{\mathcal{X}}$ as $p_{\text{id}}$, which represents the in-distribution (ID) (Havtorn et al., 2021; Nalisnick et al., 2019a). With this *unlabeled* training set, unsupervised OOD detection is to design a score function $\mathcal{S}(\boldsymbol{x})$ that can determine whether an input is ID or OOD. This is different from supervised OOD detection, which typically leverages a classifier that is trained on labeled data and primarily focuses on semantic difference (Wei et al., 2022a;b;c). We provide a detailed discussion in Appendix A.

### 2.2 VAE-BASED UNSUPERVISED OOD DETECTION

Among DGMs, VAE can offer great flexibility and strong representation ability (Xiao et al., 2022), leading to a series of unsupervised OOD detection methods based on VAE that have achieved SOTA performance (Havtorn et al., 2021; Li et al., 2022). Specifically, VAE estimates the marginal likelihood by training with the variational evidence lower bound (ELBO), *i.e.*,

$$\text{ELBO}(\boldsymbol{x}) = \mathbb{E}_{q_\phi(\boldsymbol{z}|\boldsymbol{x})}\left[\log p_\theta(\boldsymbol{x}|\boldsymbol{z})\right] - D_{\text{KL}}(q_\phi(\boldsymbol{z}|\boldsymbol{x})||p(\boldsymbol{z})), \tag{2}$$

where posterior $q_\phi(\boldsymbol{z}|\boldsymbol{x})$ is modeled by an encoder, decoder distribution $p_\theta(\boldsymbol{x}|\boldsymbol{z})$ is modeled by a decoder, and prior $p(\boldsymbol{z})$ is set as a Gaussian distribution $\mathcal{N}(\boldsymbol{0}, \mathbf{I})$. After training, $\text{ELBO}(\boldsymbol{x})$ is an estimation of true data distribution $p(\boldsymbol{x})$ that could be seen as $\mathcal{S}(\boldsymbol{x})$ to do OOD detection. However, it would suffer from the *overestimation* issue. More **Related Work**, especially a comprehensive literature about analyzing DGMs' failure in OOD detection, can be seen in Appendix B.

## 3 ANALYSIS OF VAE'S *overestimation* IN UNSUPERVISED OOD DETECTION

We will first conduct an analysis to identify the factors contributing to VAE's *overestimation*, *i.e.*, the improper design of prior distribution and the gap of dataset entropy-mutual integration.

### 3.1 IDENTIFYING FACTORS OF VAE'S *Overestimation* ISSUE

Following the common analysis procedure (Nalisnick et al., 2019a), an ideal score function $\mathcal{S}(\boldsymbol{x})$ that could achieve well-behaved OOD detection performance with VAEs is expected to have the following property for any OOD dataset:

$$\mathcal{G} = \mathbb{E}_{\boldsymbol{x} \sim p_{\text{id}}(\boldsymbol{x})}[\mathcal{S}(\boldsymbol{x})] - \mathbb{E}_{\boldsymbol{x} \sim p_{\text{ood}}(\boldsymbol{x})}[\mathcal{S}(\boldsymbol{x})] > 0, \tag{3}$$

where $p_{\text{id}}(\boldsymbol{x})$ and $p_{\text{ood}}(\boldsymbol{x})$ denote the true distribution of the ID and OOD dataset, respectively. A larger gap between these two expectation terms can usually lead to better OOD detection performance.

Using the $\text{ELBO}(\boldsymbol{x})$ as the score function $\mathcal{S}(\boldsymbol{x})$, we could give a formal definition of the repeatedly reported VAE's *overestimation* issue in the context of unsupervised OOD detection (Choi et al., 2018; Havtorn et al., 2021; Li et al., 2022).

**Definition 1** (VAE's *overestimation* in unsupervised OOD Detection). Assume we have a VAE trained on a training set and we use the $\text{ELBO}(\boldsymbol{x})$ as the score function to distinguish data points sampled *i.i.d.* from the in-distribution testing set $p_{\text{id}}(\boldsymbol{x})$ and an OOD dataset $p_{\text{ood}}(\boldsymbol{x})$. When

$$\mathcal{G} = \mathbb{E}_{\boldsymbol{x} \sim p_{\text{id}}(\boldsymbol{x})}[\text{ELBO}(\boldsymbol{x})] - \mathbb{E}_{\boldsymbol{x} \sim p_{\text{ood}}(\boldsymbol{x})}[\text{ELBO}(\boldsymbol{x})] \leq 0, \tag{4}$$

which is called VAE's *overestimation* in unsupervised OOD detection.

With a clear set-level definition of *overestimation* (a discussion on the instance-level definition could be seen in Appendix K.8), we could now investigate the underlying factors causing the *overestimation* in VAE. After training a VAE, we could reformulate the expectation terms of $\mathbb{E}_{\boldsymbol{x} \sim p(\boldsymbol{x})}[\text{ELBO}(\boldsymbol{x})]$ from the perspective of information theory (Cover, 1999), *i.e.*,

$$\mathbb{E}_{\boldsymbol{x} \sim p(\boldsymbol{x})}[\mathbb{E}_{\boldsymbol{z} \sim q_\phi(\boldsymbol{z}|\boldsymbol{x})} \log p_\theta(\boldsymbol{x}|\boldsymbol{z})] = \mathbb{E}_{p(\boldsymbol{x})q_\phi(\boldsymbol{z}|\boldsymbol{x})}[\log \frac{p_\theta(\boldsymbol{z}|\boldsymbol{x})}{p(\boldsymbol{z})}p(\boldsymbol{x})] = \hat{\mathcal{I}}_{q,p}(\boldsymbol{x}, \boldsymbol{z}) - \mathcal{H}_p(\boldsymbol{x}),$$

$$\mathbb{E}_{\boldsymbol{x} \sim p(\boldsymbol{x})}[D_{\text{KL}}(q_\phi(\boldsymbol{z}|\boldsymbol{x})||p(\boldsymbol{z}))] = \mathbb{E}_{p(\boldsymbol{x})q_\phi(\boldsymbol{z}|\boldsymbol{x})}[\log \frac{q_\phi(\boldsymbol{z}|\boldsymbol{x})}{q(\boldsymbol{z})}\frac{q(\boldsymbol{z})}{p(\boldsymbol{z})}] = \hat{\mathcal{I}}_q(\boldsymbol{x}, \boldsymbol{z}) + D_{\text{KL}}(q(\boldsymbol{z})||p(\boldsymbol{z})),$$

where $q(\boldsymbol{z}) = \mathbb{E}_{\boldsymbol{x} \sim p(\boldsymbol{x})} q_\phi(\boldsymbol{z}|\boldsymbol{x})$ denotes the aggregated posterior distribution of the latent variables $\boldsymbol{z}$, and $\hat{\mathcal{I}}_{q,p}(\boldsymbol{x}, \boldsymbol{z})$ and $\hat{\mathcal{I}}_q(\boldsymbol{x}, \boldsymbol{z})$ is defined as

$$\hat{\mathcal{I}}_{q,p}(\boldsymbol{x}, \boldsymbol{z}) = -\hat{\mathcal{H}}_{q,p}(\boldsymbol{z}|\boldsymbol{x}) + \hat{\mathcal{H}}_{q,p}(\boldsymbol{z}) = \mathbb{E}_{p(\boldsymbol{x})q_\phi(\boldsymbol{z}|\boldsymbol{x})}[\log p_\theta(\boldsymbol{z}|\boldsymbol{x})] - \mathbb{E}_{q(\boldsymbol{z})}[\log p(\boldsymbol{z})], \quad (5)$$

$$\hat{\mathcal{I}}_q(\boldsymbol{x}, \boldsymbol{z}) = -\hat{\mathcal{H}}_q(\boldsymbol{z}|\boldsymbol{x}) + \hat{\mathcal{H}}_q(\boldsymbol{z}) = \mathbb{E}_{p(\boldsymbol{x})q_\phi(\boldsymbol{z}|\boldsymbol{x})}[\log q_\phi(\boldsymbol{z}|\boldsymbol{x})] - \mathbb{E}_{q(\boldsymbol{z})}[\log q(\boldsymbol{z})], \quad (6)$$

Therefore, the expectation of the ELBO$(\boldsymbol{x})$ on the data distribution $p(\boldsymbol{x})$ could be rewritten as

$$\mathbb{E}_{\boldsymbol{x} \sim p(\boldsymbol{x})}[\text{ELBO}(\boldsymbol{x})] = \mathbb{E}_{\boldsymbol{x} \sim p(\boldsymbol{x})}[\mathbb{E}_{\boldsymbol{z} \sim q_\phi(\boldsymbol{z}|\boldsymbol{x})} \log p_\theta(\boldsymbol{x}|\boldsymbol{z})] - \mathbb{E}_{\boldsymbol{x} \sim p(\boldsymbol{x})}[D_{\text{KL}}(q_\phi(\boldsymbol{z}|\boldsymbol{x})||p(\boldsymbol{z}))] \quad (7)$$

$$= -D_{\text{KL}}(q(\boldsymbol{z})||p(\boldsymbol{z})) - [\mathcal{H}_p(\boldsymbol{x}) + \hat{\mathcal{I}}_q(\boldsymbol{x}, \boldsymbol{z}) - \hat{\mathcal{I}}_{q,p}(\boldsymbol{x}, \boldsymbol{z})] = -D_{\text{KL}}(q(\boldsymbol{z})||p(\boldsymbol{z})) - \text{Ent-Mut}(\theta, \phi, p),$$

where we term the dataset entropy-mutual integration as $\text{Ent-Mut}(\theta, \phi, p)$, which is a constant that is only related to the data distribution and the model parameters, and its value would not be changed with our proposed methods. Thus, the gap $\mathcal{G}$ in Eq. 4 could be rewritten as

$$\mathcal{G} = [-D_{\text{KL}}(q_{\text{id}}(\boldsymbol{z})||p(\boldsymbol{z})) + D_{\text{KL}}(q_{\text{ood}}(\boldsymbol{z})||p(\boldsymbol{z}))] - [\text{Ent-Mut}(\theta, \phi, p_{id}) - \text{Ent-Mut}(\theta, \phi, p_{ood})], \quad (8)$$

where the prior $p(\boldsymbol{z})$ is typically set as a standard (multivariate) Gaussian distribution $\mathcal{N}(\boldsymbol{0}, \mathbf{I})$ to enable reparameterization for efficient gradient descent optimization (Kingma & Welling, 2014). Please kindly note that this work focuses on developing a general post-hoc method that could be directly applied to alleviate the VAE's *overestimation* regardless of its model architecture or training scheme, so we do not develop methods to change the value of the $\text{Ent-Mut}(\theta, \phi, p)$ term in this paper. We leave the detailed definition and derivation in Appendix C.1.

Through analyzing the most widely used criterion, specifically the expectation of ELBO reformulated in Eq. 8, for VAE-based unsupervised OOD detection, we find that there will be two potential factors that lead to the *overestimation* issue of VAE, *i.e.*, $\mathcal{G} \le 0$:

**Factor I: The improper design of prior distribution $p(\boldsymbol{z})$.** Several studies have also argued that the aggregated posterior distribution of latent variables $q(\boldsymbol{z})$ cannot always equal $\mathcal{N}(\boldsymbol{0}, \mathbf{I})$, particularly when the dataset exhibits intrinsic multimodality (Xiao et al., 2022; Rosca et al., 2018; Sohl-Dickstein et al., 2015; Feller, 2015). Thus, the value of $D_{\text{KL}}(q_{\text{id}}(\boldsymbol{z})||p(\boldsymbol{z}))$ could be overestimated, potentially contributing to $\mathcal{G} \le 0$. Please note that our analysis is applicable to all trained VAEs in practice because we do not assume that the ELBO$(\boldsymbol{x})$, a lower bound of the model distribution $p_\theta(\boldsymbol{x})$, can converge exactly to the true one $p(\boldsymbol{x})$. Even if it is possible in theory, the observations in (Dai & Wipf, 2019; Dai et al., 2021) will prevent this from happening in practice.

**Factor II: The gap of dataset entropy-mutual integration Ent-Mut$(\theta, \phi, p)$ between ID and OOD datasets .** Considering the dataset's statistics, such as the variance of pixel values, different datasets exhibit various levels of entropy. As an example, the FashionMNIST dataset should possess higher entropy compared to the MNIST dataset. Therefore, when the entropy of an observed ID dataset is too high, the value of $-\mathcal{H}_{p_{\text{id}}}(\boldsymbol{x}) + \mathcal{H}_{p_{\text{ood}}}(\boldsymbol{x})$ could be small, potentially leading to *overestimation*. The mutual information term $\hat{\mathcal{I}}_{q,p}(\boldsymbol{x}, \boldsymbol{z}) - \hat{\mathcal{I}}_q(\boldsymbol{x}, \boldsymbol{z})$ measures the optimality of the parameters $\theta$ and $\phi$ on data distribution $p$ and it could also cause *overestimation* for OOD detection since the trained VAEs in practice are always not optimal (Dai & Wipf, 2019; Dai et al., 2021).

## 3.2 FURTHER ANALYSIS ON FACTOR I

Since factor I can be counter-intuitive and challenging to comprehend, we provide a further analysis.

**When the design of prior is proper?** Assuming a dataset $\{\boldsymbol{x}_i\}_{i=1}^N$ sampled *i.i.d.* from $p(\boldsymbol{x}) = \mathcal{N}(\boldsymbol{x}|\boldsymbol{0}, \boldsymbol{\Sigma}_{\mathbf{x}})$ as shown in Figure 1(a), and we construct a linear VAE to estimate $p(\boldsymbol{x})$, formulated as:

$$p(\boldsymbol{z}) = \mathcal{N}(\boldsymbol{z}|\boldsymbol{0}, \mathbf{I}), q_\phi(\boldsymbol{z}|\boldsymbol{x}) = \mathcal{N}(\boldsymbol{z}|\mathbf{A}\boldsymbol{x} + \mathbf{B}, \mathbf{C}), p_\theta(\boldsymbol{x}|\boldsymbol{z}) = \mathcal{N}(\boldsymbol{x}|\mathbf{E}\boldsymbol{z} + \mathbf{F}, \sigma^2\mathbf{I}), \quad (9)$$

where all learnable parameters' optimal values can be obtained by the derivation in Appendix C.3. As depicted in Figure 1(c), we find that the linear VAE can accurately estimate the $p(\boldsymbol{x})$. Figures 1(b) and 1(d) indicate that the design of the prior distribution is proper, where $q(\boldsymbol{z})$ equals $p(\boldsymbol{z})$.

**When the design of prior is NOT proper?** Consider a more complex data distribution, *e.g.*, a mixture of Gaussians as shown in Figure 1(e) (More details are in Appendix C.2), we could also get the optimal parameters of the same linear VAE in Eq. 9. After the derivation in Appendix C.4, Figure 1(f) illustrates that $q(\boldsymbol{z})$ is a multi-modal distribution instead of $p(\boldsymbol{z}) = \mathcal{N}(\boldsymbol{z}|\boldsymbol{0}, \mathbf{I})$, *i.e.*, the design of the prior is not proper, which leads to *overestimation* as seen in Figure 1(g). However, as analyzed in

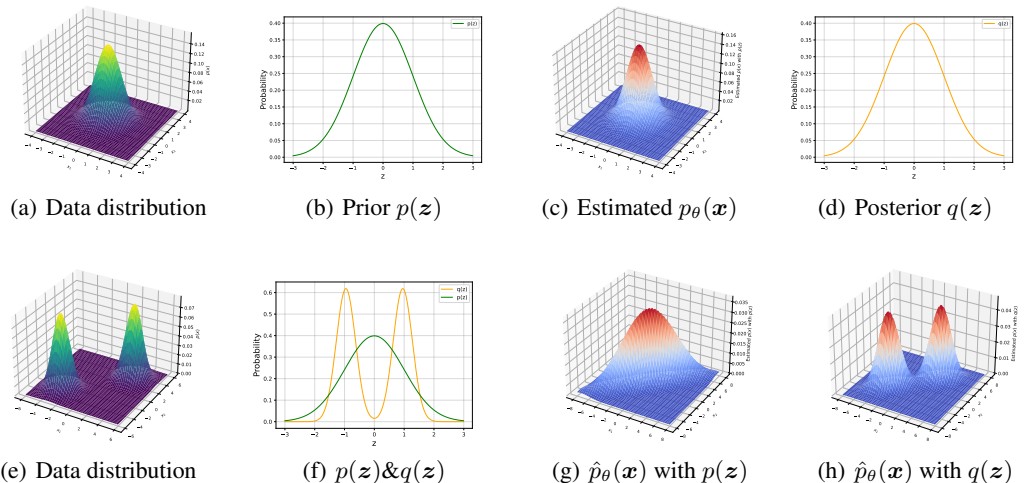

Figure 1: (**a-d**): Visualization of modeling a **single-modal** data distribution with a linear VAE; (**e-h**): Visualization of modeling a **multi-modal** data distribution with a linear VAE.

Factor I, we find that the *overestimation* issue is mitigated when replacing $p(z)$ in the KL term of the ELBO with $q(z)$, which is shown in Figure 1(h).

**More empirical studies on non-linear VAEs for the improper design of prior.** For more practical cases, we use non-linear deep VAEs to model $q_\phi(z|x)$ and $p_\theta(x|z)$ with $p(z) = \mathcal{N}(0, I)$ on the same multi-modal dataset in Figure 1(e) and image datasets. Implementation details are in C.5. For the low-dimensional multi-modal dataset, we observed that $q(z)$ still differs from $p(z)$, as shown in Figure 2(a). The ELBO still suffers from *overestimation*, especially in the region near $(0, 0)$, as shown in Figure 2(b). For the image datasets, please note that, if $q_{id}(z)$ is closer to $p(z) = \mathcal{N}(0, I)$, $z_{id} \sim q_{id}(z)$ should occupy the center of latent space $\mathcal{N}(0, I)$ and $z_{ood} \sim q_{ood}(z)$ should be pushed far from the center, leading to $p(z_{id})$ to be larger than $p(z_{ood})$. Surprisingly, we find this expected phenomenon does not exist, as shown in Figure 2(c) and 2(d), where the experiments are on two dataset pairs, Fashion-MNIST(ID)/MNIST(OOD) and CIFAR-10(ID)/SVHN(OOD). This still suggests that the prior $p(z)$ is improper, even $q_{ood}(z)$ for OOD data may be closer to $p(z)$ than $q_{id}(z)$.

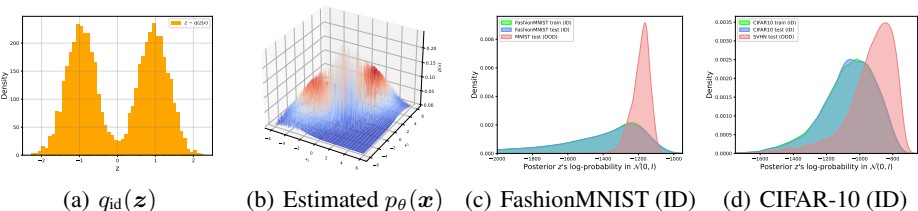

Figure 2: (**a**) and (**b**): visualization of $q_{id}(z)$ and estimated $p_\theta(x)$ by ELBO on the multi-modal data distribution with a non-linear deep VAE; (**c**) and (**d**): the density plot of the log-probability of posterior $z$ sampled from $q_{id/ood}(z)$ in prior $\mathcal{N}(0, I)$ on two dataset pairs.

**Brief summary.** Through analyzing *overestimation* scenarios from simple to complex, the answer to the beginning question of this part could be: *the prior distribution $p(z) = \mathcal{N}(0, I)$ may be an improper choice for VAE when modeling a complex data distribution $p(x)$*, leading to an overestimated $D_{KL}(q_{id}(z)||p(z))$ and further raising the *overestimation* issue in unsupervised OOD detection.

## 4 ALLEVIATING VAE'S *overestimation* IN UNSUPERVISED OOD DETECTION

In this section, we develop a new score function for OOD detection, named **"AVOID"** and denoted as $\mathcal{S}_{\text{AVOID}}(x)$, to alleviate the influence of two aforementioned factors in Section 3, combined up with two parts: **i)** post-hoc prior (PHP) and **ii)** dataset entropy-mutual calibration (DEC). Specifically, the expression of $\mathcal{S}_{\text{AVOID}}(x)$ is:

$$\mathcal{S}_{\text{AVOID}}(x) := \mathbb{E}_{q_\phi(z|x)}[\log p_\theta(x|z)] - D_{KL}[q_\phi(z|x)||\hat{q}_{id}(z)] + \mathcal{C}(x), \quad (10)$$

where the PHP method is replacing the Gaussian prior $p(\boldsymbol{z})$ with the estimated aggregated posterior distribution $\hat{q}_{\text{id}}(\boldsymbol{z})$, which is the approximation of $q_{\text{id}}(\boldsymbol{z}) = \mathbb{E}_{\boldsymbol{x} \sim p_{id}(\boldsymbol{x})} q_\phi(\boldsymbol{z}|\boldsymbol{x})$, in the original KL-term $D_{\text{KL}}[q_\phi(\boldsymbol{z}|\boldsymbol{x})||p(\boldsymbol{z})]$, *i.e.*,

$$\text{PHP}(\boldsymbol{x}) := \mathbb{E}_{\boldsymbol{z} \sim q_\phi(\boldsymbol{z}|\boldsymbol{x})} \log p_\theta(\boldsymbol{x}|\boldsymbol{z}) - D_{\text{KL}}(q_\phi(\boldsymbol{z}|\boldsymbol{x})||\hat{q}_{\text{id}}(\boldsymbol{z})); \tag{11}$$

and the DEC method is defined by introducing the additional $\mathcal{C}(\boldsymbol{x})$ term:

$$\text{DEC}(\boldsymbol{x}) := \mathbb{E}_{\boldsymbol{z} \sim q_\phi(\boldsymbol{z}|\boldsymbol{x})} \log p_\theta(\boldsymbol{x}|\boldsymbol{z}) - D_{\text{KL}}(q_\phi(\boldsymbol{z}|\boldsymbol{x})||p(\boldsymbol{z})) + \mathcal{C}(\boldsymbol{x}). \tag{12}$$

Both aforementioned PHP and DEC could be independent OOD detection methods targeted on different factors and combining them up could achieve the final OOD detection method "AVOID". In the following two parts, we will conduct a detailed analysis of the motivation of each part for why they could help alleviate the *overestimation* and both of them are implemented in a simple way to inspire related future work and can be further investigated for improvement.

### 4.1 POST-HOC PRIOR METHOD FOR FACTOR I

To provide a more insightful view to investigate the relationship between $q_{\text{id}}(\boldsymbol{z})$, $q_{\text{ood}}(\boldsymbol{z})$, and $p(\boldsymbol{z})$, we use t-SNE (Van der Maaten & Hinton, 2008) to visualize them in Figure 3. As the dark-blue points (latent representations of Fashion-MNIST) are much more distinguishable from the red points (MNIST) than the light-blue points (latent $\boldsymbol{z}$ sampled from $\mathcal{N}(\boldsymbol{0}, \mathbf{I})$) from the red points, this indicates that $p(\boldsymbol{z})$ cannot distinguish between the latent variables sampled from $q_{\text{id}}(\boldsymbol{z})$ and $q_{\text{ood}}(\boldsymbol{z})$, while $q_{\text{id}}(\boldsymbol{z})$ is clearly distinguishable from $q_{\text{ood}}(\boldsymbol{z})$. Therefore, to alleviate *overestimation*, we can explicitly modify the prior distribution $p(\boldsymbol{z})$ in Eq. 8 to force it to be closer to $q_{\text{id}}(\boldsymbol{z})$ and far from $q_{\text{ood}}(\boldsymbol{z})$, *i.e.*, decreasing $D_{\text{KL}}(q_{\text{id}}(\boldsymbol{z})||p(\boldsymbol{z}))$ and increasing $D_{\text{KL}}(q_{\text{ood}}(\boldsymbol{z})||p(\boldsymbol{z}))$.

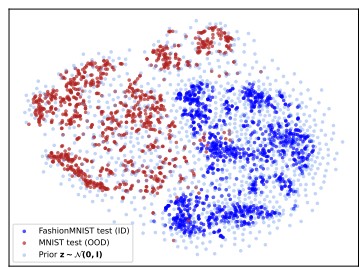

Figure 3: The t-SNE visualization of latent representations on FashionM-NIST(ID)/MNIST(OOD) dataset pair.

A straightforward modifying approach is to replace $p(\boldsymbol{z})$ in the KL-term of ELBO with an additional distribution $\hat{q}_{\text{id}}(\boldsymbol{z})$ that can fit $q_{\text{id}}(\boldsymbol{z})$ well **after training** the VAE, where the target value of $q_{\text{id}}(\boldsymbol{z})$ can be acquired by marginalizing $q_\phi(\boldsymbol{z}|\boldsymbol{x})$ over the training set, *i.e.*, $q_{\text{id}}(\boldsymbol{z}) = \mathbb{E}_{\boldsymbol{x} \sim p_{\text{id}}(\boldsymbol{x})}[q_\phi(\boldsymbol{z}|\boldsymbol{x})]$. Previous study on distribution matching (Rosca et al., 2018) has developed an LSTM-based method to efficiently fit $q_{\text{id}}(\boldsymbol{z})$ in the latent space, *i.e.*,

$$\hat{q}_{\text{id}}(\boldsymbol{z}) = \prod_{t=1}^{T} q(\boldsymbol{z}_t|\boldsymbol{z}_{<t}), \text{ where } q(\boldsymbol{z}_t|\boldsymbol{z}_{<t}) = \mathcal{N}(\mu_i, \sigma_i^2). \tag{13}$$

Thus, we could propose the *post-hoc prior* (PHP) method defined in Eq. 11 for Factor I and its expectation on a data distribution is

$$\mathbb{E}_{\boldsymbol{x} \sim p(\boldsymbol{x})}[\text{PHP}(\boldsymbol{x})] = -D_{\text{KL}}[q(\boldsymbol{z})||\hat{q}_{id}(\boldsymbol{z})] - \text{Ent-Mut}(\theta, \phi, p), \tag{14}$$

which could lead to better OOD detection performance since it could enlarge the gap $\mathcal{G}$, *i.e.*,

$$\mathcal{G}_{\text{PHP}} = [-D_{\text{KL}}(q_{\text{id}}(\boldsymbol{z})||\hat{q}_{\text{id}}(\boldsymbol{z}) + D_{\text{KL}}(q_{\text{ood}}(\boldsymbol{z})||\hat{q}_{\text{id}}(\boldsymbol{z}))] - [\text{Ent-Mut}(\theta, \phi, p_{id}) - \text{Ent-Mut}(\theta, \phi, p_{ood})].$$

### 4.2 DATASET ENTROPY-MUTUAL CALIBRATION METHOD FOR FACTOR II

While the term of dataset entropy-mutual integration is a constant that remains unaffected when the model parameters are fixed after training, it is still an essential factor that could lead to *overestimation*. To address this, a straightforward approach is to design a calibration method that ensures the value added to the ELBO of ID data will be larger than that of OOD data, expressed as

$$\text{Property 1: } \mathbb{E}_{\boldsymbol{x} \sim p_{\text{id}}(\boldsymbol{x})}[\mathcal{C}(\boldsymbol{x})] > \mathbb{E}_{\boldsymbol{x} \sim p_{\text{ood}}(\boldsymbol{x})}[\mathcal{C}(\boldsymbol{x})]. \tag{15}$$

Additionally, since this method needs to be incorporated with the PHP method to form the final AVOID method in Eq. 10, there should be a "weight" balance between the DEC and PHP methods. An ideal choice is to set a "weight" that could remove the influence of $\text{Ent-Mut}(\theta, \phi, p_{id})$ on the effectiveness of the PHP method, which could be achieved by regularizing the scale of the expectation of the DEC method on the ID training set, denoted as $\mathcal{C}_{id}^{\text{scale}}$, to have the following property:

$$\text{Property 2: } \mathcal{C}_{id}^{\text{scale}} = \mathbb{E}_{\boldsymbol{x} \sim p_{\text{id}}(\boldsymbol{x})}[\mathcal{C}(\boldsymbol{x})] = \text{Ent-Mut}(\theta, \phi, p_{id}) > 0. \tag{16}$$

With the property 2, the expectation of $\text{DEC}(\boldsymbol{x})$ on the ID data becomes

$$\mathbb{E}_{\boldsymbol{x}\sim p_{\text{id}}(\boldsymbol{x})}[\text{DEC}(\boldsymbol{x})] = \mathbb{E}_{\boldsymbol{x}\sim p_{\text{id}}(\boldsymbol{x})}[\text{ELBO}(\boldsymbol{x})] + \mathcal{C}_{\text{scale}} = -D_{\text{KL}}(q_{\text{id}}(\boldsymbol{z})||p(\boldsymbol{z})). \qquad (17)$$

Thus, DEC is totally focused on factor 2 and would not affect the PHP method's effectiveness, which also guarantees the AVOID method could always be better than both the PHP and DEC methods.

With the properties 1 and 2 of the DEC method, we could find that the new gap $\mathcal{G}_{\text{DEC}}$ becomes larger than the original gap $\mathcal{G}$ based solely on ELBO, as

$$\mathcal{G}_{\text{DEC}} = \mathcal{G} + \mathbb{E}_{\boldsymbol{x}\sim p_{\text{id}}(\boldsymbol{x})}[\mathcal{C}(\boldsymbol{x})] - \mathbb{E}_{\boldsymbol{x}\sim p_{\text{ood}}(\boldsymbol{x})}[\mathcal{C}(\boldsymbol{x})] > \mathcal{G}, \qquad (18)$$

which should alleviate the *overestimation* and lead to better unsupervised OOD detection performance.

**How to design the calibration $\mathcal{C}(\boldsymbol{x})$?** Since the PHP method is mainly focused on the semantic difference between ID and OOD data, we hope the DEC method could be targeted at the statistical difference. Inspired by the previous work (Serrà et al., 2020), we could use data compression methods like SVD (Stewart, 1993) to roughly measure the complexity of a data example, where the data examples from the same dataset should have similar complexity. An intuitive insight into this could be shown in Figure 4, where the ID dataset's statistical feature, *i.e.*, the curve, is distinguishable to other datasets. Based on this empirical study, we could first propose a **non-scaled** calibration function, denoted as $\mathcal{C}_{\text{non}}(\boldsymbol{x})$, to achieve the Property 1. First, we could set the number of singular values as $n_{\text{id}}$, which can achieve the reconstruction error

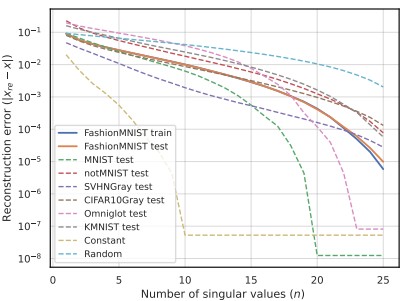

Figure 4: Visualization of the relationship between the number of singular values and the reconstruction error.

$|\boldsymbol{x}_{\text{recon}} - \boldsymbol{x}| = \epsilon$ in the ID training set; then for a test input $\boldsymbol{x}_i$, we use SVD to calculate the smallest $n_i$ that could also achieve a smaller reconstruction error $\epsilon$, then $\mathcal{C}_{\text{non}}(\boldsymbol{x})$ could be formulated as:

$$\mathcal{C}_{\text{non}}(\boldsymbol{x}) = \begin{cases} (n_i/n_{\text{id}}), & \text{if} \quad n_i < n_{\text{id}}, \\ (n_{\text{id}} - (n_i - n_{\text{id}}))/n_{\text{id}}, & \text{if} \quad n_i \geq n_{\text{id}}, \end{cases} \qquad (19)$$

which can give the ID dataset a higher expectation $\mathbb{E}_{\boldsymbol{x}\sim p_{\text{id}}(\boldsymbol{x})}[\mathcal{C}_{\text{non}}(\boldsymbol{x})]$ than that of other statistically different OOD datasets. More details to obtain $\mathcal{C}_{\text{non}}(\boldsymbol{x})$ can be found in Algorithm 1 of Appendix D.

Then, we need to get the scale $\mathcal{C}_{id}^{\text{scale}}$ for achieving the Property 2, which could be approximated by the PHP method, *i.e.*, $\mathcal{C}_{id}^{\text{scale}} = -\mathbb{E}_{\boldsymbol{x}\sim p_{\text{id}}(\boldsymbol{x})}[\text{PHP}(\boldsymbol{x})]$, as $\hat{q}_{id}(\boldsymbol{x})$ approaches $q_{id}(\boldsymbol{z})$ after post-hoc fitting:

$$\mathcal{C}_{id}^{\text{scale}} = \text{Ent-Mut}(\theta, \phi, p_{id}) + D_{\text{KL}}[q_{id}(\boldsymbol{z})||\hat{q}_{id}(\boldsymbol{z})] \approx \text{Ent-Mut}(\theta, \phi, p_{id}). \qquad (20)$$

Thus, by introducing a constant $c = \mathcal{C}_{id}^{\text{scale}}/\mathbb{E}_{\boldsymbol{x}\sim p_{id}}[\mathcal{C}_{\text{non}}(\boldsymbol{x})]$, we could implement the final **scaled** calibration function, formulated as

$$\mathcal{C}(\boldsymbol{x}) = \mathcal{C}_{\text{non}}(\boldsymbol{x}) \cdot c = \begin{cases} (n_i/n_{\text{id}}) \cdot c, & \text{if} \quad n_i < n_{\text{id}}, \\ [(n_{\text{id}} - (n_i - n_{\text{id}}))/n_{\text{id}}] \cdot c, & \text{if} \quad n_i \geq n_{\text{id}}. \end{cases} \qquad (21)$$

## 5 EXPERIMENTS

### 5.1 EXPERIMENTAL SETUP

**Baselines.** Our experiments primarily encompass two comparison aspects: **i**) evaluating our novel score function "AVOID" against previous unsupervised OOD detection methods to determine whether it can achieve competitive performance; and **ii**) comparing "AVOID" with VAE's ELBO to assess whether our method can mitigate *overestimation* and yield improved performance. For comparisons in **i**, we can categorize the baselines into three groups, as outlined in (Li et al., 2022): "**Supervised**" includes supervised OOD detection methods that utilize in-distribution data labels; "**Auxiliary**" refers to methods that employ auxiliary knowledge gathered from OOD data; and "**Unsupervised**" encompasses methods without reliance on labels or OOD-specific assumptions. For comparisons in **ii**, we compare our method with a standard VAE (Kingma & Welling, 2014), denoted as "ELBO", which also serves as the foundation our methods. Further details regarding these baselines and their respective categories can be found in Appendix E.3.

Due to the page limitation, we leave the detailed descriptions of the **Datasets** in Appendix E.1, **Evaluation Metrics** in Appendix E.2, and **Implementation details** in Appendix E.4.

Table 1: The comparisons of our method and other OOD detection methods. The best results achieved by the methods of the category "Not ensembles" of "Unsupervised" have been bold. We leave the citation and detailed description for each method's abbreviation in Appendix E.3.

| FashinMNIST(ID)/MNIST(OOD) | | | | | | CIFAR-10(ID)/SVHN(OOD) | | | | | |
|---|---|---|---|---|---|---|---|---|---|---|---|
| Supervised | | Auxiliary | | Unsupervised | | Supervised | | Auxiliary | | Unsupervised | |
| Method | AUROC↑ | Mehod | AUROC↑ | Method | AUROC↑ | Method | AUROC↑ | Mehod | AUROC↑ | Method | AUROC↑ |
| CP | 73.4 | LR(PC) | 99.4 | *-Ensembles* | | MD | 99.7 | LR(PC) | 93.0 | *-Ensembles* | |
| CP(Ent) | 74.6 | LR(BC) | 45.5 | WAIC(5VAE) | 76.6 | LMD | 27.9 | LR(VAE) | 26.5 | WAIC(5Glow) | 99.0 |
| ODIN | 75.2 | CP(OOD) | 87.7 | WAIC(5PC) | 22.1 | EN | 99.4 | OE | 98.4 | WAIC(5PC) | 62.8 |
| VIB | 94.1 | CP(Cal) | 90.4 | *-Not Ensembles* | | iDE | 95.7 | IC(Glow) | 95.0 | *-Not Ensembles* | |
| MD(CNN) | 94.2 | IC(Glow) | 99.8 | LRe | 98.8 | LN | 98.4 | IC(HVAE) | 83.3 | LRe | 87.5 |
| MD(DN) | 98.6 | IC(PC++) | 96.7 | HVK | 98.4 | ODIN | 82.9 | WOODS | 99.9 | HVK | 89.1 |
| DE | 85.7 | | | $\mathcal{LLR}^{ada}$ | 98.0 | GN | 76.7 | DCM | 99.8 | $\mathcal{LLR}^{ada}$ | 94.2 |
| | | | | AVOID(ours) | **99.2** | | | | | AVOID(ours) | **94.5** |

Table 2: The comparisons of our method with post-hoc prior (denoted as "PHP") or dataset entropy-mutual calibration (denoted as "DEC") individually and other unsupervised OOD detection methods. "PHP+DEC" is equal to our method "AVOID". Bold numbers are superior results.

| FashinMNIST(ID)/MNIST(OOD) | | | | CIFAR-10(ID)/SVHN(OOD) | | | |
|---|---|---|---|---|---|---|---|
| Method | AUROC↑ | AUPRC↑ | FPR80↓ | Method | AUROC↑ | AUPRC↑ | FPR80↓ |
| ELBO (Kingma & Welling, 2014) | 23.5 ±0.82 | 35.6 ±0.85 | 98.5 ±0.38 | ELBO (Kingma & Welling, 2014) | 24.9 ±1.41 | 36.7 ±1.52 | 94.6 ±0.96 |
| WAIC(5VAE) (Choi et al., 2018) | 76.6 ±0.83 | 78.1 ±0.85 | 51.1 ±0.80 | WAIC(5VAE) (Choi et al., 2018) | 71.9 ±0.95 | 73.2 ±1.02 | 49.1 ±0.89 |
| HVK (Havtorn et al., 2021) | 98.4 ±0.79 | 98.4 ±0.73 | 1.3 ±0.04 | HVK (Havtorn et al., 2021) | 89.1 ±2.32 | 87.5 ±2.96 | 17.2 ±2.00 |
| $\mathcal{LLR}^{ada}$(Li et al., 2022) | 97.0 ±0.58 | 97.6 ±0.72 | 0.9 ±0.03 | $\mathcal{LLR}^{ada}$(Li et al., 2022) | 92.6 ±0.41 | 91.8 ±0.54 | 11.1 ±0.27 |
| *-Ours:* | | | | *-Ours:* | | | |
| PHP | 89.7 ±0.54 | 90.3 ±0.50 | 13.3 ±0.24 | PHP | 39.6 ±1.37 | 42.6 ±1.53 | 85.7 ±0.69 |
| DEC | 34.1 ±0.00 | 40.7 ±0.00 | 92.5 ±0.00 | DEC | 87.8 ±0.00 | 89.9 ±0.00 | 17.8 ±0.00 |
| PHP+DEC | **99.2** ±0.51 | **99.4** ±0.60 | **0.00** ±0.00 | PHP+DEC | **94.5** ±1.44 | **95.3** ±1.48 | **4.24** ±0.36 |

## 5.2 COMPARISON WITH UNSUPERVISED OOD DETECTION BASELINES

First, we compare our method with other SOTA baselines in Table 1. The results demonstrate that our method achieves competitive performance compared to "Supervised" and "Auxiliary" methods and outperforms "Unsupervised" OOD detection methods. Next, we provide a more detailed comparison with some unsupervised methods, particularly the ELBO of VAE, as shown in Table 2. More results on VAEs trained on CelebA (ID) and vertical flip data detection experiments are shown in Table 5, 6, and 7 of Appendix F. Lastly, to assess our method's generalization capabilities, we test it on a broader range of datasets in Table 3. Experimental results, especially for "harder" dataset pairs as introduced in Appendix E.1, strongly verify our analysis of the VAE's *overestimation* issue and demonstrate that our method consistently mitigates *overestimation* on a wide range of OOD datasets.

## 5.3 ABLATION STUDY ON VERIFYING THE POST-HOC PRIOR METHOD

To evaluate the effectiveness of the Post-hoc Prior (PHP), we compare it with other unsupervised methods in Table 2. Moreover, we test the PHP method on additional datasets and present the results in Table 8 of Appendix G. The experimental results demonstrate that the PHP method can alleviate the *overestimation*. To provide a better understanding, we also visualize the density plot of ELBO and PHP for the "FashionMNIST(ID)/MNIST(OOD)" dataset pair in Figures 5(a) and 5(b), respectively.

The Log-likelihood Ratio ($\mathcal{LLR}$) methods (Havtorn et al., 2021; Li et al., 2022) are the current SOTA unsupervised OOD detection methods that also focus on latent variables. These methods are based on an empirical assumption that the bottom layer latent variables of a hierarchical VAE could learn low-level features and top layers learn semantic features. However, we discovered that while ELBO could already perform well in detecting some OOD data, the $\mathcal{LLR}$ method (Li et al., 2022) could negatively impact OOD detection performance to some extent, as demonstrated in Figure 5(c), where the model is trained on MNIST and detects FashionMNIST as OOD. On the other hand, our method can still maintain comparable performance since the PHP method can explicitly alleviate *overestimation*, which is one of the strengths of our method compared to the SOTA methods.

## 5.4 ABLATION STUDY ON VERIFYING DATASET ENTROPY-MUTUAL CALIBRATION METHOD

We evaluate the performance of *dataset entropy-mutual calibration* (DEC) method in Table 2 and Table 9 of Appendix H. Although the DEC method is simple, our results show that it effectively alleviates *overestimation*. To better understand DEC, we visualize the calculated $\mathcal{C}(\boldsymbol{x})$ of CIFAR-10

Table 3: The comparisons of our method "AVOID" and baseline "ELBO" on more datasets. Bold numbers are superior performance.

| ID | FashionMNIST | | | ID | CIFAR-10 | | |
|---|---|---|---|---|---|---|---|
| OOD | AUROC ↑ | AUPRC ↑ | FPR80 ↓ | OOD | AUROC ↑ | AUPRC ↑ | PFR80 ↓ |
| | ELBO / AVOID (ours) | | | | ELBO / AVOID (ours) | | |
| KMNIST | 60.03 / **78.71** | 54.60 / **68.91** | 61.6 / **48.4** | CelebA | 57.27 / **71.23** | 54.51 / **72.13** | 69.03 / **54.45** |
| Omniglot | 99.86 / **100.0** | 99.89 / **100.0** | 0.00 / **0.00** | SUN | 53.14 / **63.09** | 54.48 / **63.32** | 79.52 / **68.63** |
| notMNIST | 94.12 / **97.72** | 94.09 / **97.70** | 8.29 / **2.20** | Places365 | 57.24 / **68.37** | 56.96 / **69.05** | 73.13 / **62.64** |
| CIFAR-10-G | 98.01 / **99.01** | 98.24 / **99.04** | 1.20 / **0.40** | LFWPeople | 64.15 / **67.72** | 59.71 / **68.81** | 59.44 / **54.45** |
| CIFAR-100-G | 98.49 / **98.59** | 97.49 / **97.87** | 1.00 / **1.00** | CIFAR-100 | 52.91 / **55.36** | 51.15 / **72.13** | 77.42 / **73.93** |
| SVHN-G | 95.61 / **96.20** | 96.20 / **97.41** | 3.00 / **0.40** | Texture | 37.86 / **81.82** | 40.93 / **62.42** | 82.22 / **64.34** |
| CelebA-G | 97.33 / **97.87** | 94.71 / **95.82** | 3.00 / **0.40** | Flowers102 | 67.68 / **76.83** | 64.68 / **78.01** | 57.94 / **46.65** |
| SUN-G | 99.16 / **99.32** | 99.39 / **99.47** | 0.00 / **0.00** | GTSRB | 39.50 / **53.06** | 41.73 / **49.84** | 86.61 / **73.63** |
| Const | 94.94 / **95.20** | 97.27 / **97.32** | 1.80 / **1.70** | Const | 0.001 / **80.12** | 30.71 / **89.42** | 100.0 / **22.38** |
| Random | 99.80 / **100.0** | 99.90 / **100.0** | 0.00 / **0.00** | Random | 71.81 / **99.31** | 82.89 / **99.59** | 85.71 / **0.000** |

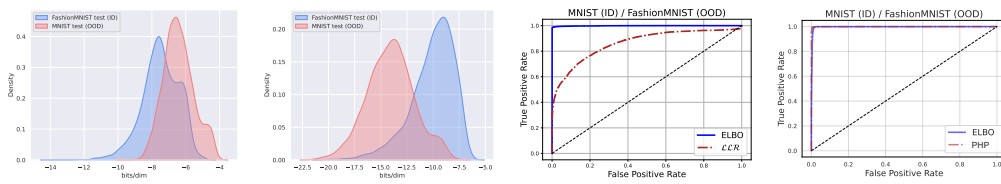

(a) Density plot of ELBO (b) Density plot of PHP (c) ROC curve of $\mathcal{LLR}$ (d) ROC curve of PHP

Figure 5: Density plots and ROC curves. **(a):** directly using ELBO$(\boldsymbol{x})$, an estimation of the $p(\boldsymbol{x})$, of a VAE trained on FashionMNIST leads to *overestimation* in detecting MNIST as OOD data; **(b):** using PHP method could alleviate the *overestimation*; **(c):** SOTA method $\mathcal{LLR}$ hurts the performance when ELBO could already work well; **(d):** PHP method would not hurt the performance.

(ID) in Figure 6(a) and other OOD datasets in Figure 6(b) when $n_{\text{id}} = 20$. Our results show that the $\mathcal{C}(\boldsymbol{x})$ of CIFAR-10 (ID) achieves generally higher values than that of other datasets, which is the underlying reason for its effectiveness in alleviating *overestimation*. Additionally, we investigate the impact of different $n_{\text{id}}$ on OOD detection performance in Figure 6(c), where our results show that the performance of dataset entropy-mutual calibration is consistently better than ELBO.

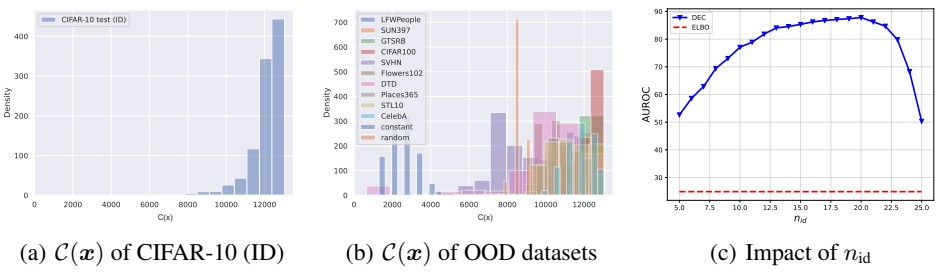

(a) $\mathcal{C}(\boldsymbol{x})$ of CIFAR-10 (ID) (b) $\mathcal{C}(\boldsymbol{x})$ of OOD datasets (c) Impact of $n_{\text{id}}$

Figure 6: **(a)** and **(b)** are respectively the visualizations of the calculated entropy-mutual calibration $\mathcal{C}(\boldsymbol{x})$ of CIFAR-10 (ID) and other OOD datasets, where the $\mathcal{C}(\boldsymbol{x})$ of CIFAR-10 (ID) could achieve generally higher values. **(c)** is the OOD detection performance of dataset entropy-mutual calibration with different $n_{\text{id}}$ settings, which consistently outperforms ELBO.

## 6 CONCLUSION

**Limitations** could be the the simplicity of the developed methods that may under-explore the full capabilities of VAEs on unsupervised OOD detection and the introduced extra computation burden.

In conclusion, this work highlights the underlying factors that lead to VAE's *overestimation* in unsupervised OOD detection and develops a novel score function called "AVOID", which is effective in alleviating *overestimation* and improving unsupervised OOD detection. This work may lead a research stream for improving unsupervised OOD detection by developing more efficient and sophisticated methods aimed at optimizing these revealed factors.

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

APPENDIX

## A    MORE BACKGROUND ON OOD DETECTION

To provide a clear distinction and avoid confusion between supervised and unsupervised OOD detection, we delineate the key differences here, primarily focusing on their respective setups.

**Setup of *unsupervised* OOD detection.** Denoting the input space with $\mathcal{X}$, an *unlabeled* training dataset $\mathcal{D}_{\text{train}} = \{x_i\}_{i=1}^N$ containing of $N$ data points can be obtained by sampling *i.i.d.* from a data distribution $\mathcal{P}_{\mathcal{X}}$. Typically, we treat the $\mathcal{P}_{\mathcal{X}}$ as $p_{\text{id}}$, which represents the in-distribution (ID) (Havtorn et al., 2021; Nalisnick et al., 2019a). With this *unlabeled* training set, unsupervised OOD detection is to design a score function $\mathcal{S}(x)$ that can determine whether an input is ID or OOD.

**Setup of *supervised* OOD detection.** Compared with the setup of unsupervised OOD detection, supervised one needs to additionally introduce a label space $\mathcal{Y} = \{1, ..., k\}$ with $k$ classes, and the training set becomes $\mathcal{D}_{\text{train}} = \{(x_i, y_i)\}_{i=1}^N$. Then, it typically needs to train a classifier $f : \mathcal{X} \to \mathbb{R}^k$, and OOD detection can be achieved based on the property of the classifier (Wei et al., 2022b;a;c).

We illustrate the distinction between supervised and unsupervised OOD detection in Figure 7.

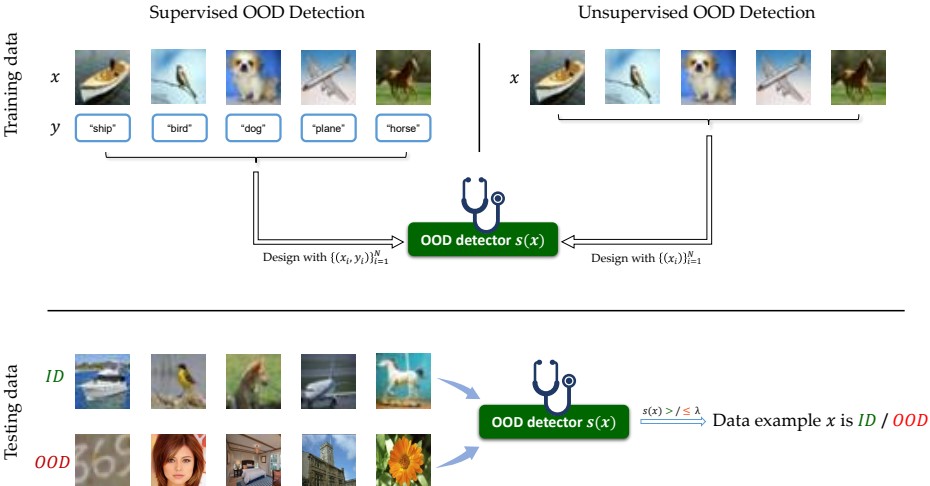

Figure 7: An illustration showcasing the difference between supervised and unsupervised OOD detection.

We also present a discussion here about the methods that people use for mitigating the overconfidence issue in the supervised case (especially the classifier-based methods), which could be divided into two categories, and draw parallels between the unsupervised and supervised cases:

1. Designing a score function based on the properties of a **trained** classifier, such as maximum softmax probability (Hendrycks & Gimpel, 2017), Mahalanobis distance-based score (Lee et al., 2018), energy-based score (Morteza & Li, 2022), and GradNorm score Huang et al. (2021). These methods operate on the premise that the statistical characteristics exhibited by the classifier when presented with an in-distribution (ID) data example are distinct from those observed when handling an out-of-distribution (OOD) data example. For unsupervised case, without the label to train a classifier, it is similar to designing a score function based on a trained deep generative model (DGM), such as HVK (Havtorn et al., 2021) that exploit the relationship between the posterior and prior latent distribution existed in a trained VAE to do OOD detection. More methods can be seen in Appendix B.3. Our method also focuses on a trained VAE, which has the advantage of "plug-and-play" and could be compatible with the existing VAEs;

2. Introducing regularization techniques during the classifier's **training phase**, such as adding a fix to the cross-entropy loss (Wei et al., 2022c), encouraging the classifier to give predictions with uniform distribution for OOD data (Hendrycks et al., 2019), and shaping the log-likelihood by energy-based regularizer (Katz-Samuels et al., 2022). After training with the regularizer, the classifier exhibits differing statistics between ID and OOD data. There are also some similar works that modify the training objective of the DGM in unsupervised cases, *e.g.*, the (Li et al., 2022) adds a partial generation term into the ELBO during training a VAE that could explicitly enhance the semantic information's quality in the latent variables that could be helpful for OOD detection. More related works could be seen in Appendix B.3. However, these methods avoid analyzing the root cause of the original DGMs' *overestimation* issue.

## B    RELATED WORK

### B.1    DEEP GENERATIVE MODELS

Deep Generative Models (DGMs) have been developed with the aim of modeling the true data distribution $p(x)$, leveraging deep neural networks to learn a generative process (Goodfellow et al., 2016). These models span several types, mainly including the autoregressive model (van den Oord et al., 2016; Salimans et al., 2017), flow model (Dinh et al., 2017; Kingma & Dhariwal, 2018), generative adversarial network (Goodfellow et al., 2020), diffusion model (Ho et al., 2020), and variational autoencoder (VAE) (Kingma & Welling, 2014). Below, we briefly introduce each of these models: The autoregressive model operates under the premise that a data sample $x$ is a sequential series, implying that the value of a pixel in an image is only dependent on the pixels preceding it. The flow model comes with an inherent requirement for the invertibility of the projection between $x$ and $z$, which imposes constraints on the implementation of its backbone. The generative adversarial network adopts an additional discriminator to implicitly learn the data distribution. Despite its power, it faces challenges such as unstable training and mode collapse (Xiao et al., 2022). The diffusion model, trained using a score-based method, has the drawback of being slow in sampling due to its multiple stochastic layers. Among these models, VAE stands out for its flexibility in implementation, comprehensive mode coverage, and fast sampling (Xiao et al., 2022). However, its training objective, an evidence lower bound of the data distribution, presents difficulties for analysis.

### B.2    ANALYSIS FOR DGMs' FAILURE IN OOD DETECTION

Some works attribute the *overestimation* to the typical set hypothesis, where ID and OOD data have overlapping supports (Nalisnick et al., 2019b; Choi et al., 2018; Wang et al., 2020; Morningstar et al., 2021). However, Zhang et al. (2021) prove that this hypothesis is not correct and appeal for more future works on analyzing the estimation error of the DGMs. Osada et al. (2023) also present the failure case of the typicality-based approach and use typicality as penality instead for improving the OOD detection performance of normalizing flow. Zhang et al. (2023) propose a perspective from the Kullback-Leibler divergence and the representation in the flow model to improve the OOD detection performance. Nalisnick et al. (2019a) find the *overestimation* issue could arise from the intrinsic model curvature brought by the invertible architecture in flow models. Kirichenko et al. (2020) claim that normalizing flows learns latent representations based on local pixel correlations and not in-distribution data's specific semantic content, which leads to *overestimation*. Serrà et al. (2020) empirically find that the image complexity could be a reason for the *overestimation* of flow and auto-regressive models, but this finding is not available in cases where the ID and OOD data share similar image complexity. Ren et al. (2019a) find that the background pixels could dominate the likelihood estimation of the flow and auto-regressive models, where the background is simulated by adding random noise to the original paper. Following this work, Cai & Li (2023) focus on the image data and find the high-frequency information in an image could cause the *overestimation*. Similar to this, Chauhan et al. (2022) claim the image intensity or contrast can empirically bias the VAE's likelihood. Le Lan & Dinh (2021) mentioned that the failure is due to the curse of dimensionality that makes the Gaussian Annulus theorem (Blum et al., 2020) fail. To cope with the curse of dimensionality, Jiang et al. (2022) propose a random projection method for flow models and Razavi et al. (2023) propose measuring the distance in a low-dimensional manifold to improve OOD detection performance of flow models. Schirrmeister et al. (2020) provide analysis for the deep

invertible networks through hierarchies of distributions and features. Caterini & Loaiza-Ganem break down the expectation of the likelihood into two terms: a KL term indicating the distance between the estimated data distribution and the true data distribution and an entropy term of the data distribution for analyzing the failure of DGMs in OOD detection. However, our analysis provides further insight into the expectation of the ELBO (a lower bound of the likelihood) directly instead of the likelihood. Loaiza-Ganem et al. (2022) analyze the failure of DGMs as a manifold overfitting issue and appeals for projecting the high-dimension input data to a lower-dimension representation first and then fitting the lower-dimension representation, instead of fitting the likelihood in the high-dimension space directly. It could be directly related to our PHP method but the DEC method is still needed to further alleviate the *overestimation* issue.

However, most of these analyses are for exact likelihood models like flow and auto-regressive models, whose training objective is the exact marginal likelihood estimation of the in-distribution data, and none of them provide a theoretical and comprehensive analysis for the VAEs, where VAEs' training objective is a lower bound of the marginal likelihood bringing extra difficulties for analysis.

### B.3 VAE-BASED UNSUPERVISED OOD DETECTION

Given the advantages of flexibility, comprehensive mode coverage, and fast sampling capabilities, variational autoencoder (VAE)-based methods have emerged as a promising choice for unsupervised out-of-distribution (OOD) detection. Based on the necessity to modify the training of VAE, these methods can be categorized into two groups. *i*) The first group includes methods that modify the training of VAE. Hierarchical VAE expands the VAE's layers to augment its representational capacity (Maaløe et al., 2019), yet the improvements in performance are marginal, and the issue of *overestimation* persists. The adaptive log-likelihood ratio method, $\mathcal{LLR}^{ada}$, is also grounded in the hierarchical VAE and introduces a generative skip connection to propagate information to higher layers of latent variables (Li et al., 2022). It utilizes the differences between each layer of latent variables for OOD detection, achieving state-of-the-art performance despite certain shortcomings as discussed in section 5.3. The tilted variational autoencoder enforces the latent variable to exist within the sphere of a tilted Gaussian (Floto et al., 2023), thereby disrupting the efficient, widely adopted reparameterization based on the Gaussian. It should be noted that modifying the training of VAE may be less practical as the proposed method cannot be directly applied to other VAEs. This implies that applying the OOD detection method to a new advanced VAE necessitates meticulous training using the new modification method. *ii*) The second group of methods attempts to utilize the properties of a trained VAE for OOD detection without modifying it. The likelihood-ratio method simulates the background using noise and employs the difference between the original and simulated background images for OOD detection (Ren et al., 2019a). The likelihood-regret method finetunes the trained VAE with the test sample to observe changes in likelihood (Xiao et al., 2020). The log-likelihood ratio method leverages the assumption that latent variables of lower layers capture low-level features of inputs while those of higher layers grasp semantic features (Havtorn et al., 2021). The difference between these latent variables can then be used for OOD detection. WAIC utilizes empirical ensemble methods for OOD detection (Choi et al., 2018). However, it should be stressed that none of these methods have strived to provide an exhaustive theoretical analysis of the VAE's *overestimation* issue.

## C DERIVATION OF THE ANALYSIS

### C.1 DERIVATION FOR EQ. 7

We first give the definition of the mutual information $\hat{\mathcal{I}}_q(\boldsymbol{x}, \boldsymbol{z})$ and $\hat{\mathcal{I}}_{q,p}(\boldsymbol{x}, \boldsymbol{z})$ as follows:

$$
\begin{aligned}
\hat{\mathcal{I}}_q(\boldsymbol{x}, \boldsymbol{z}) &= \int_{\boldsymbol{x}} \int_{\boldsymbol{z}} q(\boldsymbol{x}, \boldsymbol{z}) \log \frac{q(\boldsymbol{x}, \boldsymbol{z})}{p(\boldsymbol{x})q(\boldsymbol{z})} \\
&= \int_{\boldsymbol{x}} \int_{\boldsymbol{z}} q_\phi(\boldsymbol{z}|\boldsymbol{x})p(\boldsymbol{x}) \log \frac{q_\phi(\boldsymbol{z}|\boldsymbol{x})}{q(\boldsymbol{z})} \\
&= \mathbb{E}_{\boldsymbol{x} \sim p(\boldsymbol{x}), \boldsymbol{z} \sim q_\phi(\boldsymbol{z}|\boldsymbol{x})}[\log \frac{q_\phi(\boldsymbol{z}|\boldsymbol{x})}{q(\boldsymbol{z})}] \\
&= \mathbb{E}_{\boldsymbol{x} \sim p(\boldsymbol{x}), \boldsymbol{z} \sim q_\phi(\boldsymbol{z}|\boldsymbol{x})}[\log q_\phi(\boldsymbol{z}|\boldsymbol{x})] - \mathbb{E}_{\boldsymbol{x} \sim p(\boldsymbol{x}), \boldsymbol{z} \sim q_\phi(\boldsymbol{z}|\boldsymbol{x})}[\log q(\boldsymbol{z})] \\
&= \mathbb{E}_{\boldsymbol{x} \sim p(\boldsymbol{x}), \boldsymbol{z} \sim q_\phi(\boldsymbol{z}|\boldsymbol{x})}[\log q_\phi(\boldsymbol{z}|\boldsymbol{x})] - \mathbb{E}_{q(\boldsymbol{z})}[\log q(\boldsymbol{z})],
\end{aligned}
\tag{22}
$$

$$\hat{\mathcal{I}}_{q,p}(\boldsymbol{x}, \boldsymbol{z}) = \int_{\boldsymbol{x}} \int_{\boldsymbol{z}} q(\boldsymbol{x}, \boldsymbol{z}) \log \frac{p(\boldsymbol{x}, \boldsymbol{z})}{p(\boldsymbol{x})p(\boldsymbol{z})}$$

$$= \int_{\boldsymbol{x}} \int_{\boldsymbol{z}} q_\phi(\boldsymbol{z}|\boldsymbol{x})p(\boldsymbol{x}) \log \frac{p_\theta(\boldsymbol{z}|\boldsymbol{x})}{p(\boldsymbol{z})}$$

$$= \mathbb{E}_{\boldsymbol{x} \sim p(\boldsymbol{x}), \boldsymbol{z} \sim q_\phi(\boldsymbol{z}|\boldsymbol{x})}[\log \frac{p_\theta(\boldsymbol{z}|\boldsymbol{x})}{p(\boldsymbol{z})}] \qquad (23)$$

$$= \mathbb{E}_{\boldsymbol{x} \sim p(\boldsymbol{x}), \boldsymbol{z} \sim q_\phi(\boldsymbol{z}|\boldsymbol{x})}[\log p_\theta(\boldsymbol{z}|\boldsymbol{x})] - \mathbb{E}_{\boldsymbol{x} \sim p(\boldsymbol{x}), \boldsymbol{z} \sim q_\phi(\boldsymbol{z}|\boldsymbol{x})}[\log p(\boldsymbol{z})]$$

$$= \mathbb{E}_{\boldsymbol{x} \sim p(\boldsymbol{x}), \boldsymbol{z} \sim q_\phi(\boldsymbol{z}|\boldsymbol{x})}[\log p_\theta(\boldsymbol{z}|\boldsymbol{x})] - \mathbb{E}_{q(\boldsymbol{z})}[\log p(\boldsymbol{z})],$$

and the $q(\boldsymbol{z})$ is called the aggregated posterior distribution (Dieng et al., 2019; Makhzani et al., 2015; Mescheder et al., 2017), expressed as:

$$q(\boldsymbol{z}) = \int_{\boldsymbol{x}} q_\phi(\boldsymbol{z}|\boldsymbol{x})p(\boldsymbol{x}). \qquad (24)$$

Recall that Eq. 7 comprises two components, denoted as:

$$\mathbb{E}_{\boldsymbol{x} \sim p(\boldsymbol{x})}[\text{ELBO}(\boldsymbol{x})] = \overbrace{\mathbb{E}_{\boldsymbol{x} \sim p(\boldsymbol{x})}[\mathbb{E}_{\boldsymbol{z} \sim q_\phi(\boldsymbol{z}|\boldsymbol{x})} \log p_\theta(\boldsymbol{x}|\boldsymbol{z})]}^{L_1} - \overbrace{\mathbb{E}_{\boldsymbol{x} \sim p(\boldsymbol{x})}[D_{\text{KL}}(q_\phi(\boldsymbol{z}|\boldsymbol{x})||p(\boldsymbol{z}))]}^{L_2}.$$

$$(25)$$

Then, we have

$$L_1 = \mathbb{E}_{\boldsymbol{x} \sim p(\boldsymbol{x})}[\mathbb{E}_{\boldsymbol{z} \sim q_\phi(\boldsymbol{z}|\boldsymbol{x})} \log p_\theta(\boldsymbol{x}|\boldsymbol{z})]$$

$$= \mathbb{E}_{\boldsymbol{x} \sim p(\boldsymbol{x})}[\mathbb{E}_{\boldsymbol{z} \sim q_\phi(\boldsymbol{z}|\boldsymbol{x})} \log[\frac{p_\theta(\boldsymbol{z}|\boldsymbol{x})}{p(\boldsymbol{z})}p(\boldsymbol{x})]]$$

$$= \mathbb{E}_{\boldsymbol{x} \sim p(\boldsymbol{x})}\mathbb{E}_{\boldsymbol{z} \sim q_\phi(\boldsymbol{z}|\boldsymbol{x})} \log \frac{p_\theta(\boldsymbol{z}|\boldsymbol{x})}{p(\boldsymbol{z})} + \mathbb{E}_{\boldsymbol{x} \sim p(\boldsymbol{x})}\mathbb{E}_{\boldsymbol{z} \sim q_\phi(\boldsymbol{z}|\boldsymbol{x})} \log p(\boldsymbol{x}) \qquad (26)$$

$$= \mathbb{E}_{\boldsymbol{x} \sim p(\boldsymbol{x}), \boldsymbol{z} \sim q_\phi(\boldsymbol{z}|\boldsymbol{x})}[\log \frac{p_\theta(\boldsymbol{z}|\boldsymbol{x})}{p(\boldsymbol{z})}] + \mathbb{E}_{\boldsymbol{x} \sim p(\boldsymbol{x})} \log p(\boldsymbol{x})$$

$$= \hat{\mathcal{I}}_{q,p}(\boldsymbol{x}, \boldsymbol{z}) - \mathcal{H}_p(\boldsymbol{x}).$$

$$L_2 = \mathbb{E}_{\boldsymbol{x} \sim p(\boldsymbol{x})}[\mathbb{E}_{\boldsymbol{z} \sim q_\phi(\boldsymbol{z}|\boldsymbol{x})} \log \frac{q_\phi(\boldsymbol{z}|\boldsymbol{x})}{p(\boldsymbol{z})}]$$

$$= \mathbb{E}_{\boldsymbol{x} \sim p(\boldsymbol{x})}[\mathbb{E}_{\boldsymbol{z} \sim q_\phi(\boldsymbol{z}|\boldsymbol{x})} \log[\frac{q_\phi(\boldsymbol{z}|\boldsymbol{x})}{p(\boldsymbol{z})} \cdot \frac{q(\boldsymbol{z})}{q(\boldsymbol{z})}]]$$

$$= \mathbb{E}_{\boldsymbol{x} \sim p(\boldsymbol{x})}[\mathbb{E}_{\boldsymbol{z} \sim q_\phi(\boldsymbol{z}|\boldsymbol{x})} \log[\frac{q_\phi(\boldsymbol{z}|\boldsymbol{x})}{q(\boldsymbol{z})} \cdot \frac{q(\boldsymbol{z})}{p(\boldsymbol{z})}]] \qquad (27)$$

$$= \mathbb{E}_{\boldsymbol{x} \sim p(\boldsymbol{x})}[\mathbb{E}_{\boldsymbol{z} \sim q_\phi(\boldsymbol{z}|\boldsymbol{x})} \log \frac{q_\phi(\boldsymbol{z}|\boldsymbol{x})}{q(\boldsymbol{z})}] + \mathbb{E}_{\boldsymbol{x} \sim p(\boldsymbol{x})}[\mathbb{E}_{\boldsymbol{z} \sim q_\phi(\boldsymbol{z}|\boldsymbol{x})} \log[\frac{q(\boldsymbol{z})}{p(\boldsymbol{z})}]]$$

$$= \mathbb{E}_{\boldsymbol{x} \sim p(\boldsymbol{x}), \boldsymbol{z} \sim q_\phi(\boldsymbol{z}|\boldsymbol{x})}[\log \frac{q_\phi(\boldsymbol{z}|\boldsymbol{x})}{q(\boldsymbol{z})}] + \mathbb{E}_{\boldsymbol{z} \sim q(\boldsymbol{z})}[\log \frac{q(\boldsymbol{z})}{p(\boldsymbol{z})}]$$

$$= \hat{\mathcal{I}}_q(\boldsymbol{x}, \boldsymbol{z}) + D_{\text{KL}}(q(\boldsymbol{z})||p(\boldsymbol{z})).$$

Hence, we can achieve the following expression:

$$\mathbb{E}_{\boldsymbol{x} \sim p(\boldsymbol{x})}[\text{ELBO}(\boldsymbol{x})] = -D_{\text{KL}}(q(\boldsymbol{z})||p(\boldsymbol{z})) - [\mathcal{H}_p(\boldsymbol{x}) + \hat{\mathcal{I}}_q(\boldsymbol{x}, \boldsymbol{z}) - \hat{\mathcal{I}}_{q,p}(\boldsymbol{x}, \boldsymbol{z})]$$

$$= -D_{\text{KL}}(q(\boldsymbol{z})||p(\boldsymbol{z})) - \text{Ent-Mut}(\theta, \phi, p). \qquad (28)$$

## C.2 TOY EXAMPLES' DETAILS

**Single-modal case setup.** In this scenario, the data distribution is determined by a standard 2-dimensional Gaussian distribution $p(\boldsymbol{x}) = \mathcal{N}(\boldsymbol{x}|\boldsymbol{0}, \boldsymbol{\Sigma_x})$, where

$$\boldsymbol{\Sigma_x} = \begin{bmatrix} 1 & 0 \\ 0 & 1 \end{bmatrix}. \qquad (29)$$

In order to simulate the dimension-reduction property of VAE, we designate the dimension of the latent variable as 1-dimensional; that is, the variance $\mathbf{I}$ in $p(\boldsymbol{z})$ reduces to 1. Under this configuration, we $i.i.d.$ sample $N = 5000$ data points from the data distribution $p(\boldsymbol{x})$ to construct a training set. Each parameter's solutions are calculated analytically.

**Multi-modal case setup.** The data distribution is made by a mixture of two standard single-modal Gaussian distributions, $i.e.$, $p(\boldsymbol{x}) = \sum_{k=1}^{K} \pi_k \mathcal{N}(\boldsymbol{x}|\boldsymbol{\mu}_k, \boldsymbol{\Sigma}_k)$, where $K = 2$, $\pi_k = 1/2$ and

$$\boldsymbol{\mu}_1 = \begin{bmatrix} 3 \\ 3 \end{bmatrix}, \boldsymbol{\mu}_2 = \begin{bmatrix} -3 \\ -3 \end{bmatrix}, \boldsymbol{\Sigma}_1 = \begin{bmatrix} 1 & 0 \\ 0 & 1 \end{bmatrix}, \boldsymbol{\Sigma}_2 = \begin{bmatrix} 1 & 0 \\ 0 & 1 \end{bmatrix}. \tag{30}$$

The training set of this multi-modal case is built by $i.i.d.$ sampling from 5000 data points from each component Gaussian distribution $\mathcal{N}(\boldsymbol{x}|\boldsymbol{\mu}_k, \boldsymbol{\Sigma}_k)$, $i.e.$, 10000 data points in total.

## C.3 DERIVATION FOR SINGLE-MODAL CASE IN SECTION 3.2

Assume we have a dataset containing $N$ data samples $\{\boldsymbol{x}_1, \boldsymbol{x}_2, ..., \boldsymbol{x}_N\}$, $x_i \in \mathbb{R}^d$, $d = 2$, and we already know the groundtruth distribution of it, $i.e.$,

$$p(\boldsymbol{x}) = \mathcal{N}(\boldsymbol{x}|\mathbf{0}, \boldsymbol{\Sigma}_\mathbf{x}), \tag{31}$$

where $\boldsymbol{\Sigma}_x = \mathbf{I}$. We have a linear VAE model parameterized as:

$$p(\boldsymbol{z}) = \mathcal{N}(\boldsymbol{z}|\mathbf{0}, \mathbf{I}) \tag{32}$$
$$q_\phi(\boldsymbol{z}|\boldsymbol{x}) = \mathcal{N}(\boldsymbol{z}|\mathbf{A}\boldsymbol{x} + \mathbf{B}, \mathbf{C}) \tag{33}$$
$$p_\theta(\boldsymbol{x}|\boldsymbol{z}) = \mathcal{N}(\boldsymbol{x}|\mathbf{E}\boldsymbol{z} + \mathbf{F}, \sigma^2 \mathbf{I}), \tag{34}$$

where $p(\boldsymbol{z})$ is the prior distribution, $\boldsymbol{z} \in \mathbb{R}^q$, $q = 1$, $q_\phi(\boldsymbol{z}|\boldsymbol{x})$ is the approximated posterior distribution, and $p_\theta(\boldsymbol{x}|\boldsymbol{z})$ is the approximated likelihood distribution. Directly employing the knowledge from probabilistic Principal Component Analysis (pPCA) (Tipping & Bishop, 1999), we could get the maximum likelihood estimation of $p_\theta(\boldsymbol{x}|\boldsymbol{z})$:

$$\sigma_{\text{MLE}}^2 = \frac{1}{d - q} \sum_{j=q+1}^{d} \lambda_j \tag{35}$$

$$\mathbf{E}_{\text{MLE}} = \mathbf{U}_q \left( \boldsymbol{\Lambda}_q - \sigma_{\text{MLE}}^2 \right)^{1/2} \mathbf{R} \tag{36}$$

$$\mathbf{F}_{\text{MLE}} = \mathbf{0} \tag{37}$$

where $\lambda_{q+1}, ..., \lambda_d$ are the smallest eigenvalues of the sample covariance matrix $\mathbf{S} = \frac{1}{N} \sum_{n=1}^{N} \boldsymbol{x}\boldsymbol{x}^\top$, the $d \times q$ orthogonal matrix $\mathbf{U}_q$ is made by the $q$ dominant eigenvectors of $\mathbf{S}$, the diagonal matrix $\boldsymbol{\Lambda}_q$ contains the corresponding $q$ largest eigenvalues, and $\mathbf{R}$ is an arbitary $q \times q$ orthogonal matrix. Note that, when $q = 1$, we have $\mathbf{R} = \mathbf{I}$. After we get the parameters of $p_\theta(\boldsymbol{x}|\boldsymbol{z})$, we could get the $p(\boldsymbol{z}|\boldsymbol{x})$ by Bayes rule:

$$p(\boldsymbol{z}|\boldsymbol{x}) = \frac{p_\theta(\boldsymbol{x}|\boldsymbol{z})p(\boldsymbol{z})}{p(\boldsymbol{x})} \tag{38}$$
$$= \mathcal{N}(\boldsymbol{z}|\boldsymbol{\Sigma}_x^{-1}\mathbf{E}_{\text{MLE}}^\top \boldsymbol{x}, \sigma_{\text{MLE}}^2 \boldsymbol{\Sigma}_x^{-1}),$$

where $\boldsymbol{\Sigma}_x = \mathbf{E}_{\text{MLE}}^\top \mathbf{E}_{\text{MLE}} + \sigma_{\text{MLE}}^2 \mathbf{I}$. Thus, the maximum likelihood estimates of $q_\phi(\boldsymbol{z}|\boldsymbol{x})$'s parameters are:

$$\mathbf{A}_{\text{MLE}} = \boldsymbol{\Sigma}_x^{-1}\mathbf{E}_{\text{MLE}}^\top \tag{39}$$

$$\mathbf{B}_{\text{MLE}} = \mathbf{0} \tag{40}$$

$$\mathbf{C}_{\text{MLE}} = \sigma_{\text{MLE}}^2 \boldsymbol{\Sigma}_x^{-1}. \tag{41}$$

Although the maximum likelihood estimations are ascertained, it remains necessary to verify whether these estimations allow the ELBO to reach the global optimum. The derivation of ELBO is as follows:

$$\log p(\boldsymbol{x}) = \mathbb{E}_{q_\phi(\boldsymbol{z}|\boldsymbol{x})}[\log p(\boldsymbol{x}|\boldsymbol{z})] - D_{\text{KL}}(q_\phi(\boldsymbol{z}|\boldsymbol{x})||p(\boldsymbol{z})) + D_{\text{KL}}(q_\phi(\boldsymbol{z}|\boldsymbol{x})||p(\boldsymbol{z}|\boldsymbol{x}))$$
$$= \text{ELBO}(\boldsymbol{x}) + D_{\text{KL}}(q_\phi(\boldsymbol{z}|\boldsymbol{x})||p(\boldsymbol{z}|\boldsymbol{x})). \tag{42}$$

Given that $q_\phi(\boldsymbol{z}|\boldsymbol{x}) = \mathcal{N}(\boldsymbol{z}|\boldsymbol{\Sigma}_{\boldsymbol{x}}^{-1}\mathbf{E}_{\mathrm{MLE}}^\top\boldsymbol{x}, \sigma_{\mathrm{MLE}}^2\boldsymbol{\Sigma}_{\boldsymbol{x}}^{-1}) = p(\boldsymbol{z}|\boldsymbol{x})$, $D_{\mathrm{KL}}(q_\phi(\boldsymbol{z}|\boldsymbol{x})||p(\boldsymbol{x}|\boldsymbol{z}))$ becomes zero. Furthermore, any modifications to the parameters of $q_\phi$ would result in an increase of $D_{\mathrm{KL}}(q_\phi(\boldsymbol{z}|\boldsymbol{x})||p(\boldsymbol{x}|\boldsymbol{z}))$; in other words, it would result in a decrease of ELBO. Hence, the global optimum of the ELBO is attained when $\mathbf{A}_{\mathrm{MLE}} \sim \mathbf{E}_{\mathrm{MLE}}, \sigma_{\mathrm{MLE}}$ are implemented in the linear VAE. Moreover, in this situation, $\log p(\boldsymbol{x})$ equates to ELBO.

Finally, we could get the expression of the aggregated posterior distribution $q(\boldsymbol{z})$:

$$
\begin{aligned}
q(\boldsymbol{z}) &= \int_{\boldsymbol{x}} q_\phi(\boldsymbol{z}|\boldsymbol{x})p(\boldsymbol{x}) \\
&= \int_{\boldsymbol{x}} \mathcal{N}(\boldsymbol{z}|\boldsymbol{\Sigma}_{\boldsymbol{x}}^{-1}\mathbf{E}_{\mathrm{MLE}}^\top\boldsymbol{x}, \sigma_{\mathrm{MLE}}^2\boldsymbol{\Sigma}_{\boldsymbol{x}}^{-1})\mathcal{N}(\boldsymbol{x}|\boldsymbol{0}, \boldsymbol{\Sigma}_{\mathbf{x}}) \\
&= \int_{\boldsymbol{x}} \mathcal{N}(\boldsymbol{z}|\mathbf{I}^{-1}\mathbf{E}_{\mathrm{MLE}}^\top\boldsymbol{x}, \sigma_{\mathrm{MLE}}^2\mathbf{I}^{-1})\mathcal{N}(\boldsymbol{x}|\boldsymbol{0}, \mathbf{I}) \\
&= \int_{\boldsymbol{x}} \mathcal{N}(\boldsymbol{z}|\mathbf{E}_{\mathrm{MLE}}^\top\boldsymbol{x}, \sigma_{\mathrm{MLE}}^2\mathbf{I})\mathcal{N}(\boldsymbol{x}|\boldsymbol{0}, \mathbf{I}) \\
&= \mathcal{N}(\boldsymbol{0}, \mathbf{E}_{\mathrm{MLE}}^\top\mathbf{E}_{\mathrm{MLE}} + \sigma_{\mathrm{MLE}}^2\mathbf{I}) \\
&= \mathcal{N}(\boldsymbol{0}, \boldsymbol{\Sigma}_{\mathbf{x}}) \\
&= \mathcal{N}(\boldsymbol{0}, \mathbf{I}) \\
&= p(\boldsymbol{z}).
\end{aligned}
\tag{43}
$$

In summing up the single-modal case, our assertion is that $D_{\mathrm{KL}}[q(\boldsymbol{z})||p(\boldsymbol{z})] = 0$, indicating that the design of the prior distribution is appropriate and would not result in an *overestimation* of VAE.

## C.4 Derivation for Multi-modal Case in Section 3.2

Assume we have a distribution $p(\boldsymbol{x}) = \sum_{k=1}^K \pi_k\mathcal{N}(\boldsymbol{x}|\boldsymbol{\mu}_k, \boldsymbol{\Sigma}_k)$ and we build a dataset containing $K \times N$ data samples, which is made by sampling $N$ data samples from each $\mathcal{N}(\boldsymbol{x}|\boldsymbol{\mu}_k, \boldsymbol{\Sigma}_k)$. The parameterization setting of the $p(\boldsymbol{z})$, $q_\phi(\boldsymbol{z}|\boldsymbol{x})$, and $p_\theta(\boldsymbol{x}|\boldsymbol{z})$ is the same as the single-modal case in Section 3.2.

Deriving from the single-modal scenario, an analytical formulation of $D_{\mathrm{KL}}(q_\phi(\boldsymbol{z}|\boldsymbol{x})||p(\boldsymbol{z}|\boldsymbol{x}))$ is unattainable in the multi-modal case. Thus, it necessitates a derivation directly from the ELBO. Due to the fact that the global optimum of the decoder's parameters in the ELBO coincides with the global maximum of the marginal likelihood of the observed data (Lucas et al., 2019), we firstly commence with the derivation of the maximum likelihood estimation of $p_\theta(\boldsymbol{x}|\boldsymbol{z})$. Despite the feasibility of directly obtaining the maximum likelihood estimation of the parameters in $p_\theta(\boldsymbol{x}|\boldsymbol{z})$ by optimizing the integration $\hat{p}_\theta(\boldsymbol{x}) = \int_{\boldsymbol{z}} p_\theta(\boldsymbol{x}|\boldsymbol{z})p(\boldsymbol{z})$ using the observed data, we propose an additional clarification connecting this integration and the ELBO. With reference to the strictly tighter importance sampling on the ELBO (Burda et al., 2016), we can derive that

$$
\mathrm{ELBO}^s(\boldsymbol{x}) = \mathbb{E}_{q_\phi(\boldsymbol{z}|\boldsymbol{x})}[\log \frac{1}{S}\sum_{s=1}^S \frac{p_\theta(\boldsymbol{x}|\boldsymbol{z}^{(s)})p(\boldsymbol{z}^{(s)})}{q_\phi(\boldsymbol{z}^{(s)}|\boldsymbol{x})}].
\tag{44}
$$

Setting the number of instances $S = 1$, $\mathrm{ELBO}^s(\boldsymbol{x})$ equates to the regular $\mathrm{ELBO}(\boldsymbol{x})$. As $S$ approaches $+\infty$, it follows that

$$
\begin{aligned}
\mathrm{ELBO}^s(\boldsymbol{x}) &= \mathbb{E}_{q_\phi(\boldsymbol{z}|\boldsymbol{x})}[\log \mathbb{E}_{q_\phi(\boldsymbol{z}|\boldsymbol{x})}\frac{p_\theta(\boldsymbol{x}|\boldsymbol{z})p(\boldsymbol{z})}{q_\phi(\boldsymbol{z}|\boldsymbol{x})}] \\
&= \mathbb{E}_{q_\phi(\boldsymbol{z}|\boldsymbol{x})}[\log \int_{\boldsymbol{z}} q_\phi(\boldsymbol{z}|\boldsymbol{x})\frac{p_\theta(\boldsymbol{x}|\boldsymbol{z})p(\boldsymbol{z})}{q_\phi(\boldsymbol{z}|\boldsymbol{x})}d\boldsymbol{z}] \\
&= \mathbb{E}_{q_\phi(\boldsymbol{z}|\boldsymbol{x})}[\log \int_{\boldsymbol{z}} p_\theta(\boldsymbol{x}|\boldsymbol{z})p(\boldsymbol{z})d\boldsymbol{z}] \\
&= \log \int_{\boldsymbol{z}} p_\theta(\boldsymbol{x}|\boldsymbol{z})p(\boldsymbol{z})d\boldsymbol{z} \\
&= \log \hat{p}_\theta(\boldsymbol{x}).
\end{aligned}
\tag{45}
$$

The expression of $\hat{p}_\theta(\boldsymbol{x})$ is shown as:

$$
\begin{aligned}
\hat{p}_\theta(\boldsymbol{x}) &= \int_{\boldsymbol{z}} p_\theta(\boldsymbol{x}|\boldsymbol{z}) p(\boldsymbol{z}) \\
&= \int_{\boldsymbol{z}} \mathcal{N}(\boldsymbol{x}|\mathbf{E}\boldsymbol{z} + \mathbf{F}, \sigma^2 \mathbf{I}) \mathcal{N}(\mathbf{z}|\mathbf{0}, \mathbf{I}) \\
&= \mathcal{N}(\boldsymbol{x}|\mathbf{F}, \mathbf{E}\mathbf{E}^\top + \sigma^2 \mathbf{I}).
\end{aligned}
\tag{46}
$$

Then, the joint log-likelihood of the observed dataset $\{\boldsymbol{x}_i^{(k)}\}_{i=1,k=1}^{N,K}$ can be formulated as:

$$
\mathcal{L} = \sum_{k=1}^{K} \sum_{i=1}^{N} \log \hat{p}_\theta(\boldsymbol{x}_i^{(k)}) = -\frac{KNd}{2} \log(2\pi) - \frac{KN}{2} \log det(\mathbf{M}) - \frac{KN}{2} tr[\mathbf{M}^{-1}\mathbf{S}],
\tag{47}
$$

where $\mathbf{M} = \mathbf{E}\mathbf{E}^\top + \sigma^2 \mathbf{I}$ and $\mathbf{S} = \frac{1}{KN} \sum_{k=1}^{K} \sum_{i=1}^{N} (\boldsymbol{x}_i^{(k)} - \mathbf{F})(\boldsymbol{x}_i^{(k)} - \mathbf{F})^\top$.

Repeatly using the knowledge in pPCA again, we could get the maximum likelihood estimation of the parameters:

$$
(\sigma^*)^2 = \frac{1}{d-q} \sum_{j=q+1}^{d} \lambda_j
\tag{48}
$$

$$
\mathbf{E}^* = \mathbf{U}_q \left( \mathbf{\Lambda}_q - (\sigma^*)^2 \right)^{1/2} \mathbf{R}
\tag{49}
$$

$$
\mathbf{F}^* = \mathbf{0},
\tag{50}
$$

where $\lambda_{q+1}, ..., \lambda_d$ are the smallest eigenvalues of the sample covariance matrix $\mathbf{S} = \frac{1}{N} \sum_{n=1}^{N} \boldsymbol{x}\boldsymbol{x}^\top$, the $d \times q$ orthogonal matrix $\mathbf{U}_q$ is made by the $q$ dominant eigenvectors of $\mathbf{S}$, the diagonal matrix $\Lambda_q$ contains the corresponding $q$ largest eigenvalues, and $\mathbf{R}$ is an arbitary $q \times q$ orthogonal matrix. Note that, when $q = 1$, we have $\mathbf{R} = \mathbf{I}$. Actually, with the same $p(\boldsymbol{z})$ and a decoder $p_\theta(\boldsymbol{x}|\boldsymbol{z})$ parameterized by the same linear network, the expression of the maximum likelihood estimation of the $p_\theta(\boldsymbol{x}|\boldsymbol{z})$ in the multi-modal case is the same as the single-modal case.

In order to determine $q_\phi(\boldsymbol{z}|\boldsymbol{x})$'s parameters, we can initiate the process by identifying the stationary points of $q_\phi(\boldsymbol{z}|\boldsymbol{x})$ with respect to the ELBO. The ELBO can be analytically expressed as follows:

$$
\text{ELBO}(\boldsymbol{x}) = \overbrace{\mathbb{E}_{q_\phi(\boldsymbol{z}|\boldsymbol{x})}[\log p_\theta(\boldsymbol{x}|\boldsymbol{z})]}^{L_1} - \overbrace{D_{\text{KL}}[q_\phi(\boldsymbol{z}|\boldsymbol{x})\|p(\boldsymbol{z})]}^{L_2}
\tag{51}
$$

$$
\begin{aligned}
L_1 &= \mathbb{E}_{q_\phi(\boldsymbol{z}|\boldsymbol{x})}\left[ -\frac{(\mathbf{E}\boldsymbol{z} - \boldsymbol{x})^\top(\mathbf{E}\boldsymbol{z} - \boldsymbol{x})}{2\sigma^2} - \frac{d}{2} \log 2\pi\sigma^2 \right] \\
&= \mathbb{E}_{q_\phi(\boldsymbol{z}|\boldsymbol{x})}\left[ \frac{-(\mathbf{E}\boldsymbol{z})^\top(\mathbf{E}\boldsymbol{z}) + 2\boldsymbol{x}^\top \mathbf{E}\boldsymbol{z} - \boldsymbol{x}^\top\boldsymbol{x}}{2\sigma^2} - \frac{d}{2} \log(2\pi\sigma^2) \right] \\
&= \frac{1}{2\sigma^2}\left[ -tr(\mathbf{E}\mathbf{C}\mathbf{E}^\top) - (\mathbf{E}\mathbf{A}\boldsymbol{x} + \mathbf{E}\mathbf{B})^\top(\mathbf{E}\mathbf{A}\boldsymbol{x} + \mathbf{E}\mathbf{B}) + 2\boldsymbol{x}^\top(\mathbf{E}\mathbf{A}\boldsymbol{x} + \mathbf{E}\mathbf{B}) - \boldsymbol{x}^\top\boldsymbol{x} \right] \\
&\quad - \frac{d}{2} \log(2\pi\sigma^2)
\end{aligned}
\tag{52}
$$

$$
L_2 = \frac{1}{2}\left[ -\log det(\mathbf{C}) + (\mathbf{A}\boldsymbol{x} + \mathbf{B})^\top(\mathbf{A}\boldsymbol{x} + \mathbf{B}) + tr(\mathbf{C}) - q \right]
\tag{53}
$$

For a dataset consisting of $KN$ data samples, the stationary points with respect to the ELBO can be obtained through the following expressions:

$$
\frac{\partial(\sum^{KN} \text{ELBO}(\boldsymbol{x}))}{\partial \mathbf{A}} = KN\left[ -\mathbf{A}\mathbf{S} - \mathbf{B}\bar{\boldsymbol{x}}^\top - \frac{1}{\sigma^2}(\mathbf{E}^\top \mathbf{E}\mathbf{A}\mathbf{S}) - \frac{1}{\sigma^2}(\mathbf{E}^\top \mathbf{E}\mathbf{B}\bar{\boldsymbol{x}}^\top - \mathbf{E}^\top \mathbf{S}) \right] = \mathbf{0}
\tag{54}
$$

$$
\frac{\partial(\sum^{KN} \text{ELBO}(\boldsymbol{x}))}{\partial \mathbf{B}} = KN\left[ -\mathbf{A}\bar{\boldsymbol{x}} - \frac{1}{\sigma^2}\mathbf{E}^\top \mathbf{E}\mathbf{A}\bar{\boldsymbol{x}} + \frac{1}{\sigma^2}\mathbf{E}^\top \bar{\boldsymbol{x}} - (\mathbf{I} + \frac{\mathbf{E}^\top \mathbf{E}}{\sigma^2})\mathbf{B} \right] = \mathbf{0}
\tag{55}
$$

$$
\frac{\partial(\sum^{KN} \text{ELBO}(\boldsymbol{x}))}{\partial \mathbf{C}} = \frac{KN}{2}((\mathbf{C}^{-1})^\top - \mathbf{I} - \frac{1}{\sigma^2}(\mathbf{E}^\top \mathbf{E})) = \mathbf{0},
\tag{56}
$$

where $\mathbf{S} = \frac{1}{KN} \sum^{KN} \boldsymbol{x}\boldsymbol{x}^\top$ and $\bar{\boldsymbol{x}} = \frac{1}{KN} \sum^{KN} \boldsymbol{x}$. Upon further investigation, we have discovered that the stationary points of $\mathbf{A}$, $\mathbf{B}$, and $\mathbf{C}$ solely depend on the parameters $\mathbf{E}$ and $\sigma$. In mathematical terms, they can be expressed as:

$$\mathbf{A}^* = \frac{(\mathbf{I} + \frac{1}{\sigma^2}\mathbf{E}^\top\mathbf{E})^{-1}}{\sigma^2}\mathbf{E}^\top \tag{57}$$

$$\mathbf{B}^* = \mathbf{0} \tag{58}$$

$$\mathbf{C}^* = ((\mathbf{I} + \frac{1}{\sigma^2}\mathbf{E}^\top\mathbf{E})^\top)^{-1}. \tag{59}$$

Finally, we can derive the expression of $q(\boldsymbol{z})$ in this multi-modal case as follows:

$$
\begin{aligned}
q(\boldsymbol{z}) &= \int_{\boldsymbol{x}} q_\phi(\boldsymbol{z}|\boldsymbol{x})p(\boldsymbol{x}) \\
&= \int_{\boldsymbol{x}} \mathcal{N}(\boldsymbol{z}|\mathbf{A}^*\boldsymbol{x}, \mathbf{C}^*) \sum_{k=1}^{K} \pi_k \mathcal{N}(\boldsymbol{x}|\boldsymbol{\mu}_k, \boldsymbol{\Sigma}_k) \\
&= \sum_{k=1}^{K} \pi_k \int_{\boldsymbol{x}} \mathcal{N}(\boldsymbol{z}|\mathbf{A}^*\boldsymbol{x}, \mathbf{C}^*) \mathcal{N}(\boldsymbol{x}|\boldsymbol{\mu}_k, \boldsymbol{\Sigma}_k) \\
&= \sum_{k=1}^{K} \pi_k \mathcal{N}(\boldsymbol{z}|\mathbf{A}^*\boldsymbol{\mu}_k, \mathbf{A}^*\boldsymbol{\Sigma}_k(\mathbf{A}^*)^\top + \mathbf{C}^*) \\
&\neq p(\boldsymbol{z}).
\end{aligned}
\tag{60}
$$

In conclusion, we observe that $D_{\text{KL}}[q(\boldsymbol{z})||p(\boldsymbol{z})] \neq 0$, indicating that the design of the prior distribution $p(\boldsymbol{z})$ is not appropriate in this multi-modal case and may result in *overestimation* issue of VAE.

### C.5 IMPLEMENTATION DETAILS OF DEEP VAE IN SECTION 3.2

The non-linear deep VAE's encoder is implemented as a 3-layer MLP, which takes the 2D data points as inputs. The encoder consists of two linear layers with a hidden dimension of 10 and LeakyReLU activation function (Maas et al., 2013). The output layer, with a dimension of 2, does not have an activation function and provides the values for $\mu_z$ and $\log \sigma_z^2$ for each dimension of the latent variable.

For the decoder, it takes the sampled latent variable $\boldsymbol{z}$ through reparameterization and feeds it into two linear layers with a hidden dimension of 10 and LeakyReLU activation function. The final output is obtained by a linear layer without activation function, with a dimension of 4. The reconstruction likelihood is modeled as a Gaussian distribution, where the first two dimensions represent $\boldsymbol{\mu_x}$ (the mean of the reconstruction likelihood) and the remaining dimension represents $\log \sigma_{\boldsymbol{x}}^2$ (the log variance of the reconstruction likelihood).

The deep VAE is trained using the Adam optimizer (Kingma & Ba, 2015) with a learning rate of 1e-5. The training set consists of a total of 10,000 data points.

We also investigated the influence of dataset size (amount of training data) and model capacity (number of neural network layers) on the OOD detection performance of ELBO, using both the synthesized 2D multi-modal dataset and realistic image datasets ("FashionMNIST(ID) / MNIST(OOD)" and "CIFAR-10(ID) / SVHN(OOD)"). Our findings are illustrated in Figure 8 and Table 4. For the 2D multi-modal dataset, we sampled a data volume 10 times greater than its inherent distribution p(x) than the original configuration seen in Figure 2(a-b) of the main paper, increasing from 10,000 to 100,000 training samples. The VAE for this experiment utilized a 10-layer MLP as opposed to the original 3-layer MLP. Notably, the results from Figure 8(a) highlight that the is still not equal to p(z) = N (0, I) and Figure 8 (b) indicates the persistence of the *overestimation* problem in the non-linear deep VAE. For the practical image datasets, we varied the dataset size and model capacity (number of CNN layers) to investigate their effects on ELBO's OOD detection performance. However, results show that increasing the amount of data and the number of CNN layers does not yield significant improvements.

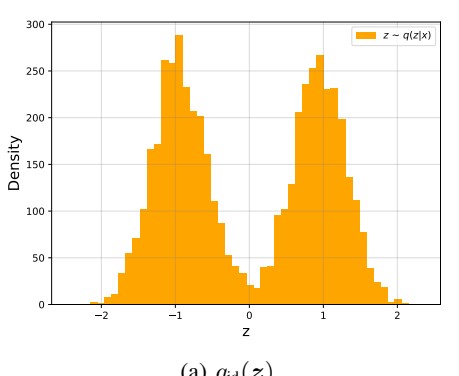
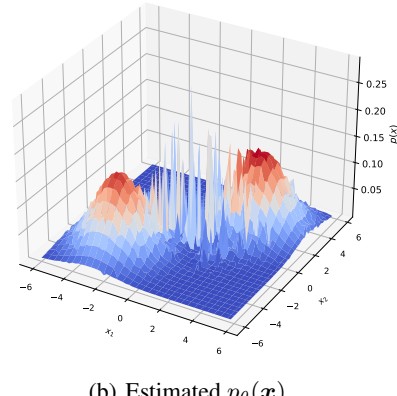

(a) $q_{\text{id}}(\boldsymbol{z})$          (b) Estimated $p_\theta(\boldsymbol{x})$

Figure 8: Visualization of $q_{\text{id}}(\boldsymbol{z})$ and estimated $p_\theta(\boldsymbol{x})$ by ELBO on a synthesized 2D multi-modal dataset. The data amount here is 10 times larger than in Figure 3 of the main paper, increasing from 10,000 to 100,000 samples. The VAE used is a non-linear deep one based on a 10-layer MLP, in contrast to the 3-layer MLP used in Figure 3 of the main paper; Results indicate that the $q_{id}(\boldsymbol{z})$ is still not equal to $p(\boldsymbol{z}) = \mathcal{N}(0, \mathbf{I})$ and the *overestimation* issue still exists.

Table 4: Ablation study examining the effects of dataset size (data amount) and model capacity (number of convolutional neural network (CNN) layers) on the OOD detection performance of ELBO. Results indicate that increasing the amount of data and the number of CNN layers does not yield significant improvements.

| FashionMNIST(ID) / MNIST(OOD) | | | | | CIFAR-10(ID) / SVHN(OOD) | | | | | |
|---|---|---|---|---|---|---|---|---|---|---|
| Num. of Layers | | | | | | Num. of Layers | | | | |
| Data Amount | 3 | 6 | 9 | 12 | 15 | Data Amount | 3 | 6 | 9 | 12 | 15 |
| 10000 | 9.45 | 14.0 | 13.2 | 14.2 | 14.6 | 10000 | 14.4 | 12.8 | 16.9 | 20.5 | 20.3 |
| 30000 | 16.3 | 14.5 | 15.3 | 14.5 | 15.8 | 30000 | 24.6 | 25.3 | 25.9 | 24.4 | 23.9 |
| 60000 | 23.5 | 25.1 | 23.0 | 20.3 | 19.8 | 50000 | 24.9 | 22.6 | 23.5 | 28.1 | 24.0 |

## D  DETAILS OF THE NON-SCALED ENTROPY-MUTUAL CALIBRATION METHOD

We provide a pseudo code here for calculating the $\mathcal{C}_{\text{non}}(\boldsymbol{x})$ of a testing sample $\boldsymbol{x}$ in Algorithm 1. Noted that, the maximum number of singular values $N$ should be larger than $n_{\text{id}}$.

---
**Algorithm 1** Non-scaled dataset entropy-mutual calibration $\mathcal{C}_{\text{non}}(\boldsymbol{x})$ algorithm
---
**Input:** Hyperparameter $n_{\text{id}}$ and its corresponding reconstruction error $\epsilon = \mathbb{E}_{\boldsymbol{x} \sim p_{\text{id}}} |\boldsymbol{x}_{\text{recon}} - \boldsymbol{x}|$, maximum number of singular values $N$, a testing sample $\boldsymbol{x}$.
**Ouput:** $\mathcal{C}_{\text{non}}(\boldsymbol{x})$.
Do SVD for the testing sample $\boldsymbol{x}$;
**for** $n_i = 1$ to $N$ **do**
    Calculate reconstruction error $\epsilon_i$ using $n_i$ singular values;
    **if** $\epsilon_i \leq \epsilon$ **then**
        **break**;
    **end if**
**end for**
**if** $n_i < n_{\text{id}}$ **then**
    Calculate $\mathcal{C}_{\text{non}}(\boldsymbol{x}) = n_i/n_{\text{id}}$;
**else**
    Calculate $\mathcal{C}_{\text{non}}(\boldsymbol{x}) = (n_{\text{id}} - (n_i - n_{\text{id}}))/n_{\text{id}}$;
**end if**
**return** $\mathcal{C}_{\text{non}}(\boldsymbol{x})$
---

# E DETAILS OF EXPERIMENTAL SETUP

## E.1 DESCRIPTION OF ALL DATASETS

In accordance with the existing literature (Nalisnick et al., 2019a;b; Morningstar et al., 2021) we evaluate our method against previous works using commonly acknowledged "**harder**" dataset pairs: FashionMNIST (ID) → MNIST (OOD), CIFAR-10 (ID) → SVHN (OOD), CelebA (ID) → CIFAR-10 (OOD), CelebA (ID) → CIFAR-100 (OOD), and CIFAR-10 (ID) → CIFAR-100 (OOD). Please note that, though CIFAR-10 and CIFAR-100 do have a few overlapping categories, the percentage of these overlapping categories in CIFAR-100 is limited and it is still regarded as a hard task (Morningstar et al., 2021; Nalisnick et al., 2019b). Additionally, (Choi et al., 2018; Morningstar et al., 2021) also identify detecting vertically flipped ID data as the OOD data to be a hard task. The suffixes "ID" and "OOD" represent in-distribution and out-of-distribution datasets, respectively.

To more comprehensively assess the generalization capabilities of these methods, we incorporate additional OOD datasets as follows. Notably, datasets featuring the suffix "-G" (*e.g.*, "CIFAR-10-G") have been converted to grayscale, resulting in a single-channel format.

For grayscale image datasets, we utilize the following datasets: FashionMNIST (Xiao et al., 2017), MNIST (LeCun et al., 1998), KMNIST (Clanuwat et al., 2018), notMNIST (Bulatov), Omniglot (Lake et al., 2015), and several grayscale datasets transformed from RGB datasets. **FashionMNIST** is a dataset consisting of 60,000 grayscale images of Zalando's article pictures for training, and 10,000 images for testing. Each image is 28x28 pixels and belongs to one of the 10 classes. **MNIST** is a widely used dataset containing 70,000 grayscale images of handwritten digits. It consists of a training set of 60,000 images and a test set of 10,000 images. Each image is 28x28 pixels. **KMNIST** is derived from the Kuzushiji Dataset and serves as a drop-in replacement for the MNIST dataset. It includes 70,000 grayscale images, each with a resolution of 28x28 pixels. **notMNIST** is a dataset composed of 547,838 grayscale images of glyphs extracted from publicly available fonts. The images are 28x28 pixels in size and cover letters A to J from various fonts. **Omniglot** contains 32,460 grayscale images of 1623 different handwritten characters from 50 distinct alphabets. Each image has a resolution of 28x28 pixels. Additionally, we have transformed several RGB datasets into grayscale versions, including CIFAR-10-G, CIFAR-100-G, SVHN-G, CelebA-G, and SUN-G.

For RGB datasets, we utilize the following datasets: CIFAR-10/CIFAR-100 (Krizhevsky & Hinton, 2009), SVHN (Netzer et al., 2011), CelebA (Liu et al.), Places365 (Zhou et al., 2017), Flower102 (Nilsback & Zisserman, 2008), LFWPeople (Huang et al., 2007), SUN (Xiao et al., 2010), GTSRB (Houben et al., 2013), STL-10 (Coates et al., 2011), and Texture (Cimpoi et al., 2014) datasets. **CIFAR-10** and **CIFAR-100** are datasets consisting of 32x32 color images. CIFAR-10 contains 50,000 training images and 10,000 testing images, with 10 different classes. CIFAR-100 has the same number of images but includes 100 classes. **SVHN** is a dataset obtained from Google Street View images, primarily used for recognizing digits and numbers in natural scene images. **CelebA** is a large-scale face attributes dataset containing over 200,000 celebrity images, each annotated with 40 attribute labels. **Places365** is a dataset that includes 1.8 million training images from 365 scene categories. The validation set contains 50 images per category, and the testing set contains 900 images per category. **Flower102** is an image classification dataset consisting of 102 flower categories, with each class containing between 40 and 258 images. The selected flowers are commonly found in the United Kingdom. **LFWPeople** contains more than 13,000 images of faces collected from the web, making it a popular dataset for face-related tasks. **SUN** is a large-scale scene recognition dataset, covering a wide range of scenes from abbey to zoo. **STL10** is an image recognition dataset designed for unsupervised feature learning. It includes labeled data from 10 categories and unlabeled data from additional classes. **GTSRB** is a dataset specifically developed for the task of German traffic sign recognition. **Texture** is an evolving collection of textured images in various real-world settings. All images from these datasets are resized to the dimensions of 32x32x3 before being used as input for the models.

## E.2 EVALUATION AND METRICS.

We adhere to the previous evaluation procedure (Havtorn et al., 2021; Li et al., 2022), where all methods are trained using the training split of the ID dataset, and their OOD detection performance is assessed on both the testing split of the ID dataset and the OOD dataset. In line with previous works

(Hendrycks & Gimpel, 2017; Alemi et al., 2018; Hendrycks et al., 2019), we employ evaluation metrics including the area under the receiver operating characteristic curve (AUROC ↑), the area under the precision-recall curve (AUPRC ↑), and the false positive rate at 80% true positive rate (FPR80 ↓). The arrows indicate the direction of improvement for each metric.

### E.3 DESCRIPTION OF ALL BASELINES

Following the categorization in $\mathcal{LLR}^{ada}$ (Li et al., 2022), we provide a detailed description of each baseline within the three categories:

**"Supervised":** Methods using in-distribution data labels $y$, which is the same as the "Label" category in $\mathcal{LLR}^{ada}$ (Li et al., 2022), including:

- Maximum softmax classification probability (CP) method (Hendrycks & Gimpel, 2017), denoted as "**CP**", and its variants: "**CP(OOD)**" with OOD data as a noise class, "**CP(Cal)**" with calibration on OOD data, and "**CP(Ent)**" with the entropy of softmax classification probability $p(y|x)$;
- Mahalanobis distance (**MD**) method (Lee et al., 2018);
- Latent Mahalanobis distance (**LMD**) method (Bulusu et al., 2020);
- Out-of-distribution image detection in neural networks (**ODIN**) method (Liang et al., 2018);
- Variational information bottleneck (**VIB**) method (Alemi et al., 2018);
- Deep ensembles (**DE**) method (Lakshminarayanan et al., 2017) with 20 classifiers;
- GradNorm (**GN**) method (Huang et al., 2021);
- LogitNorm (**LN**) method (Wei et al., 2022c);

**"Auxiliary":** Methods using auxiliary knowledge assumptions about ID or OOD data type, which is the same as the "Prior" category in $\mathcal{LLR}^{ada}$ (Li et al., 2022)), including:

- Likelihood Ratio (**LR**) method (Ren et al., 2019b) with different backbones, denoted as "**LR(PC)**" with backbone PixelCNN, "**LR(VAE)**" with VAE and "**LR(BC)**" with binary classifier);
- Outlier exposure (**OE**) method (Hendrycks et al., 2019);
- Input complexity (**IC**) method (Serrà et al., 2020) with different backbones, denoted as "**IC(PC++)**" with backbone PixelCNN++, "**IC(Glow)**" with backbone Glow and "**IC(HVAE)**" with backbone HVAE;
- Wild OOD detection sans-Supervision (**WOODS**) method (Katz-Samuels et al., 2022);
- Data-driven confidence minimization (**DCM**) methods (Choi et al., 2023).

**"Unsupervised":** Methods with no OOD-specific assumptions) including:

- Ensemble methods: WAIC method (Choi et al., 2018) with different backbones, denoted as "**WAIC (5Glow)**" with 5 Glow models, "**WAIC (5VAE)**" with 5 VAE models and "**WAIC (5PC)**" with 5 PixelCNN models;
  Not ensembles methods:
- Likelihood regret (**LRe**) method (Xiao et al., 2020) that utilize the difference in finetuning the VAE with ID and OOD data;
- Log-Likelihood Ratio (**HVK**) method (Havtorn et al., 2021) that employ the consistency between the specific low-level and high-level information in a hierarchical VAE;
- Adaptive Log-Likelihood Ratio ($\mathcal{LLR}^{ada}$) method (Li et al., 2022) that modify the training process to propagate more information to the high-level latent variables and then collect the consistency in all level of latent variables in a hierarchical VAE to do OOD detection.

The "**ELBO**" in all tables is the result of a standard VAE (Kingma & Welling, 2014) and our proposed methods PHP, DEC, and AVOID are based on this standard VAE. Other methods's performance are based on their own best settings reported in their original paper.

### E.4 DETAILS OF THE IMPLEMENTATION

The VAE's latent variable's dimension is set as 200 for all experiments with the encoder and decoder parameterized by a 3-layer convolutional neural network, respectively. The reconstruction likelihood distribution is modeled by a discretized mixture of logistics (Salimans et al., 2017). For optimization, we adopt the Adam optimizer (Kingma & Ba, 2015) with a learning rate of 1e-3. We train all models in comparison by setting the batch size as 128 and the max epoch as 1000 following (Havtorn et al., 2021; Li et al., 2022). All experiments are performed on a PC with an NVIDIA A100 GPU and implemented with PyTorch (Paszke et al., 2019).

The encoder of the VAE is implemented as a 3-layer convolutional network with kernel numbers of 32, 64, and 128, and strides of 1, 2, and 2, respectively. The ReLU (Krizhevsky et al., 2012) activation function is applied. The output layer consists of a linear layer that outputs the mean and log-variance of the latent variables, with a dimension of 200.

On the other hand, the decoder takes the reparameterized latent variables as input and utilizes a 3-layer transposed convolutional network. The network has kernel numbers of 128, 64, and 32, and strides of 2, 2, and 1, respectively. The ReLU activation function is used. Finally, the output layer is parameterized by a convolutional layer that models the distribution as a discretized mixture of logistics.

In the PHP method, an LSTM is employed as the backbone (Shi et al., 2015). The hidden size of the LSTM is set to 64, and the outputted hidden state is fed into a 3-layer linear network. The hidden sizes of the linear layers are 64, 32, and 2, respectively. The ReLU activation function is applied to the first two layers. The optimizer used for learning the $q(z)$ distribution is Adam, and the learning rate is set to 1e-4.

## F MORE EXPERIMENTAL RESULTS OF AVOID

We add more experimental results with models trained on CelebA (ID) in Table 5 and 6. Additionally, we add experiments on detecting vertically flipped data as OOD data in Table 7.

Table 5: Comparisons between our methods and other VAE-based OOD detection methods on the "harder tasks" (CelebA(ID) / CIFARs(OOD)). Bold numbers are superior results.

| CelebA(ID) / CIFAR-10(OOD) | | | | CelebA(ID) / CIFAR-100(OOD) | | | |
|---|---|---|---|---|---|---|---|
| Method | AUROC↑ | AUPRC↑ | FPR80↓ | Method | AUROC↑ | AUPRC↑ | FPR80↓ |
| ELBO (Kingma & Welling, 2014) | 27.8 ±1.30 | 37.5 ±1.33 | 96.3 ±0.91 | ELBO (Kingma & Welling, 2014) | 33.1 ±1.38 | 41.9 ±1.32 | 96.7 ±0.87 |
| HVK (Havtorn et al., 2021) | 40.1 ±2.00 | 43.8 ±2.43 | 88.1 ±2.18 | HVK (Havtorn et al., 2021) | 45.2 ±2.05 | 49.0 ±1.98 | 91.2 ±2.23 |
| $\mathcal{LLR}^{ada}$ (Li et al., 2022) | 58.0 ±0.67 | 62.5 ±0.70 | 77.3 ±0.36 | $\mathcal{LLR}^{ada}$ (Li et al., 2022) | 52.5 ±0.59 | 58.8 ±0.56 | 85.6 ±0.35 |
| *-Ours* | | | | *-Ours* | | | |
| PHP | 69.5 ±1.29 | 63.7 ±1.27 | 50.2 ±0.74 | PHP | 68.9 ±1.30 | 64.2 ±1.31 | 50.6 ±0.68 |
| DEC | 73.3 ±0.00 | 67.7 ±0.00 | 45.5 ±0.00 | DEC | 73.7 ±0.00 | 67.0 ±0.00 | 46.4 ±0.00 |
| AVOID | **75.6** ±1.35 | **70.3** ±1.38 | **43.4** ±0.43 | AVOID | **75.5** ±1.40 | **69.8** ±1.46 | **42.1** ±0.44 |

Table 6: Comparisons on more OOD datasets between our method and other VAE-based OOD detection methods with VAEs trained on CelebA(ID). Bold numbers are superior results.

| AUROC↑ with models trained on **CelebA (ID)** | | | | | | | | | |
|---|---|---|---|---|---|---|---|---|---|
| OOD datasets | SVHN | STL10 | Places365 | LFWPeople | SUN | GTSRB | Texture | Const | Random |
| ELBO (Kingma & Welling, 2014) | 27.2 | 56.9 | 50.2 | 52.2 | 27.1 | 67.9 | 54.5 | 1.24 | **100** |
| HVK (Havtorn et al., 2021) | 36.8 | 59.7 | 59.1 | **59.9** | 54.3 | 49.8 | 61.5 | 92.9 | 74.4 |
| $\mathcal{LLR}^{ada}$ (Li et al., 2022) | 91.2 | 61.5 | 55.7 | 58.6 | 58.8 | 42.3 | 68.1 | 90.2 | 73.4 |
| *-Ours* | | | | | | | | | |
| PHP | 56.9 | 59.9 | 63.5 | 52.5 | 67.2 | 72.0 | 63.2 | 53.4 | 100 |
| DEC | 99.7 | 60.1 | 60.9 | 55.7 | 66.1 | 67.8 | 68.5 | 97.0 | 100 |
| AVOID | **95.8** | **67.6** | **68.4** | 55.9 | **73.7** | **75.6** | **76.3** | **97.1** | **100** |

## G MORE ABLATION STUDY RESULTS ON VERIFYING THE POST-HOC PRIOR

We evaluate the effectiveness of the PHP method on additional datasets as shown in Table 8.

Table 7: Comparison with ELBO on detecting vertically flipped ("VFlip") data as OOD.

| AUROC ↑ in detecting VFlip data as OOD. | | | | | |
|---|---|---|---|---|---|
| Method | CelebA | CIFAR-10 | SVHN | FashionMNIST | MNIST |
| ELBO (Kingma & Welling, 2014) | 74.2 | 49.5 | 50.4 | 69.5 | 82.7 |
| AVOID(=PHP) | 85.7 | 53.7 | 52.7 | 86.2 | 84.9 |

Table 8: The comparisons of the OOD detection performance of our method on more datasets. The new score function only has **post-hoc prior** part.

| ID | FashionMNIST | | | ID | CIFAR-10 | | |
|---|---|---|---|---|---|---|---|
| OOD | AUROC ↑ | AUPRC ↑ | FPR80 ↓ | OOD | AUROC ↑ | AUPRC ↑ | PFR80 ↓ |
| | ELBO / PHP (ours) | | | | ELBO / PHP (ours) | | |
| KMNIST | 60.03 / **72.98** | 54.60 / **69.34** | 61.6 / **48.1** | CelebA | 57.27 / **70.91** | 54.51 / **72.16** | 69.03 / **52.95** |
| Omniglot | 99.86 / **99.90** | 99.89 / 99.89 | 0.00 / **0.00** | CIFAR-100 | 52.91 / **55.00** | 51.15 / **54.01** | 77.42 / **70.23** |
| notMNIST | 94.12 / **94.39** | 94.09 / **94.35** | 8.29 / **7.79** | Places365 | 57.24 / **57.36** | 56.96 / 56.55 | 73.13 / **52.95** |
| CIFAR-10-G | 98.01 / **98.84** | 98.24 / **99.13** | 1.20 / **0.30** | LFWPeople | 64.15 / **64.57** | 59.71 / **65.20** | 59.44 / 64.74 |
| CIFAR-100-G | 98.49 / **98.50** | 97.49 / **97.50** | 1.00 / **0.90** | SUN | 53.14 / **53.27** | 54.48 / **54.67** | 79.52 / **78.12** |
| SVHN-G | 95.61 / **96.00** | 96.20 / **97.13** | 3.00 / **0.60** | Texture | 37.86 / **43.38** | 40.93 / **43.99** | 82.22 / **80.12** |
| CelebA-G | 97.33 / **97.71** | 94.71 / **95.62** | 3.00 / **2.20** | Flowers102 | 67.68 / **67.76** | 64.68 / **64.75** | 57.94 / **57.63** |
| SUN-G | 99.16 / **99.26** | 99.39 / **99.40** | 0.00 / **0.00** | GTSRB | 39.50 / **52.62** | 41.73 / **50.81** | 86.61 / **75.12** |
| Const | 94.94 / **95.08** | 97.27 / **97.35** | 1.80 / **0.00** | Const | 0.001 / **15.70** | 30.71 / **30.78** | 100.0 / **86.62** |
| Random | 99.80 / **99.81** | 99.90 / **99.90** | 0.00 / **0.00** | Random | 71.81 / **72.52** | 82.89 / **83.42** | 85.71 / **85.00** |

# H    MORE ABLATION STUDY RESULTS ON VERIFYING THE DATASET ENTROPY-MUTUAL CALIBRATION

We evaluate the effectiveness of the DEC method on additional datasets as shown in Table 9.

Table 9: The comparisons of the OOD detection performance of our method on more datasets. The new score function only has **dataset entropy-mutual calibration** part.

| ID | FashionMNIST | | | ID | CIFAR-10 | | |
|---|---|---|---|---|---|---|---|
| OOD | AUROC ↑ | AUPRC ↑ | FPR80 ↓ | OOD | AUROC ↑ | AUPRC ↑ | PFR80 ↓ |
| | ELBO / DEC (ours) | | | | ELBO / DEC (ours) | | |
| KMNIST | 60.03 / **60.54** | 54.60 / **55.18** | 61.6 / **60.3** | CelebA | 57.27 / **69.00** | 54.51 / **61.83** | 69.03 / **50.93** |
| Omniglot | 99.86 / **99.91** | 99.89 / **99.94** | 0.00 / **0.00** | CIFAR-100 | 52.91 / **54.69** | 51.15 / **52.98** | 77.42 / 73.23 |
| notMNIST | 94.12 / **94.50** | 94.09 / 93.61 | 8.29 / **6.89** | Places365 | 57.24 / **68.14** | 56.96 / **65.16** | 73.13 / **64.26** |
| CIFAR-10-G | 98.01 / **99.31** | 98.24 / **99.25** | 1.20 / **0.40** | LFWPeople | 64.15 / **67.84** | 59.71 / **60.28** | 59.44 / **54.75** |
| CIFAR-100-G | 98.49 / **98.81** | 97.49 / **98.05** | 1.00 / **0.90** | SUN | 53.14 / **60.55** | 54.48 / **60.67** | 79.52 / **68.75** |
| SVHN-G | 95.61 / **97.06** | 96.20 / **97.92** | 3.00 / **0.00** | Texture | 37.86 / **70.36** | 40.93 / **60.02** | 82.22 / **64.16** |
| CelebA-G | 97.33 / **97.69** | 94.71 / **95.94** | 3.00 / **2.10** | Flowers102 | 67.68 / **75.59** | 64.68 / **77.84** | 57.94 / **46.48** |
| SUN-G | 99.16 / **99.58** | 99.39 / **99.67** | 0.00 / **0.00** | GTSRB | 39.50 / **48.35** | 41.73 / **45.59** | 86.61 / **73.83** |
| Const | 94.94 / **99.31** | 97.27 / **99.25** | 1.80 / **0.40** | Const | 0.001 / **76.20** | 30.71 / **83.27** | 100.0 / **58.04** |
| Random | 99.80 / **100.0** | 99.90 / **100.0** | 0.00 / **0.00** | Random | 71.81 / **99.53** | 82.89 / **99.73** | 85.71 / **0.000** |

# I    ERROR BAR

We conduct random experiments on all grayscale and RGB datasets for 5 trials using the trainable methods (ELBO, PHP, and AVOID methods). The average error rates are presented in Table 10 and 11, and it can be observed that the error rates are stable and small across these methods on gray image data and nature RGB image data.

# J    BROADER IMPACT

The impact of our research can be outlined in two key aspects:

Table 10: The error bar on the gray image dataset pair FashionMNIST (ID) / MNIST (OOD).

| Error bar on FashionMNIST (ID) / MNIST (OOD) | | | |
|---|---|---|---|
| Method | AUROC $\uparrow$ | AUPRC $\uparrow$ | FPR80 $\downarrow$ |
| ELBO (Kingma & Welling, 2014) | $23.5 \pm 0.82$ | $35.6 \pm 0.85$ | $98.5 \pm 0.38$ |
| HVK (Havtorn et al., 2021) | $98.4 \pm 0.79$ | $98.4 \pm 0.73$ | $1.3 \pm 0.04$ |
| $\mathcal{LLR}^{ada}$ (Li et al., 2022) | $97.0 \pm 0.58$ | $97.6 \pm 0.72$ | $0.9 \pm 0.03$ |
| -ours | | | |
| PHP | $89.7 \pm 0.54$ | $90.3 \pm 0.50$ | $13.3 \pm 0.24$ |
| DEC | $34.1 \pm 0.00$ | $40.7 \pm 0.00$ | $92.5 \pm 0.00$ |
| AVOID | $99.2 \pm 0.51$ | $99.4 \pm 0.60$ | $0.0 \pm 0.00$ |

Table 11: The error bar on the gray image dataset pair CIFAR-10 (ID) / SVHN (OOD).

| Error bar on CIFAR-10 (ID) / SVHN (OOD) | | | |
|---|---|---|---|
| Method | AUROC $\uparrow$ | AUPRC $\uparrow$ | FPR80 $\downarrow$ |
| ELBO (Kingma & Welling, 2014) | $24.9 \pm 1.41$ | $36.7 \pm 1.52$ | $94.6 \pm 0.96$ |
| HVK (Havtorn et al., 2021) | $89.1 \pm 2.32$ | $87.5 \pm 2.96$ | $17.2 \pm 2.00$ |
| $\mathcal{LLR}^{ada}$ (Li et al., 2022) | $92.6 \pm 0.41$ | $91.8 \pm 0.54$ | $11.1 \pm 0.27$ |
| -ours | | | |
| PHP | $39.6 \pm 1.37$ | $42.6 \pm 1.53$ | $85.7 \pm 0.69$ |
| DEC | $87.9 \pm 0.00$ | $89.9 \pm 0.00$ | $17.8 \pm 0.00$ |
| AVOID | $94.5 \pm 1.44$ | $95.3 \pm 1.48$ | $4.24 \pm 0.36$ |

- For Unsupervised OOD Detection: Our approach stands out due to its broad applicability and versatility. Unlike many conventional methods, it does not require labeled data and it can be applied to model the distribution of diverse data types using deep generative models. This is particularly useful in applications where labeled data is scarce or unavailable. Additionally, our method provides a universal solution to enhance OOD detection performance. This is achieved by offering an innovative perspective on the *overestimation* issue in VAE, which is not predicated on the data type.

- For the development of deep generative models: Our research offers valuable insights for the progression of deep generative models. By employing the KL divergence, $D_{\mathrm{KL}}(q(\boldsymbol{z})\|p(\boldsymbol{z}))$, our method can provide verification of whether a generative model has adequately learned to model the data distribution. These insights could potentially spark new developments and inspire more representative generative models, thereby furthering the field of deep learning research and applications.

In conclusion, our research holds promising potential to provide substantial contributions to both the realm of unsupervised OOD detection and the development of deep generative models.

## K  REBUTTAL

To ease the reading of reviewers, we have included all additional experiments for rebuttal in this section, which could be summarized as follows:

- (For reviewer **ms1o**, **G5ek**, **jWXz**, **F6GH**) in Appendix K.1, we demonstrate the insight for why the PHP method does not work that well in the CIFAR10(ID)/SVHN(OOD) case with UMAP (Uniform Manifold Approximation and Projection) visualization;

- (For reviewer **G5ek**) in Appendix K.2, we testify our methods' robustness in a reverse direction, i.e., MNIST (ID) / FashionMNIST (OOD) and SVHN (ID) / CIFAR-10 (OOD);

- (For reviewer **F6GH**) in Appendix K.3, we testify our methods' OOD detection performance with a model trained on the CIFAR-100 dataset;

- (For reviewer **ms1o**, **G5ek**, **jWXz**) in Appendix K.4, we testify our method's OOD detection performance with a more complex backbone, diffusion model;

- (For reviewer **ms1o**, **G5ek**) in Appendix K.5, we analyze the computation efficiency about our methods and ensemble methods reagrding the training time and inference time;

- (For reviewer **G5ek**, **jWXz**) in Appendix K.6, we testify the implementation of DEC method with other two image compressors, *i.e.*, JPEG and PNG compressor;

- (For reviewer **ms1o**) in Appendix K.7, we testify the noisy setting, which demonstrates the robustness of our method and could support our analysis about the Ent-Mut term.

## K.1 WHY DOES PHP NOT WORK WELL IN SOME CASES?

We provide a further analysis for the PHP method of when it would not work well, specifically why it works well in FashionMNIST (ID) / MNIST (0OD) dataset pair but not work well on CIFAR-10 (ID) / SVHN (OOD) dataset pair. Additionally, thanks for the reviewer's recommendation, we replace t-SNE with UMAP (Uniform Manifold Approximation and Projection) McInnes et al. (2018) for visualizing the latent representations. As the results shown in Figure 9, the latent variable's aggregated posterior distribution of FashionMNIST $q_{fashion}(z)$ is pretty distinguishable from that of MNIST $q_{mnist}(z)$ but the aggregated posterior distribution of CIFAR-10 $q_{cifar}(z)$ has some overlapping with SVHN $q_{svhn}(z)$, which may due to the shared low-level features across CIFAR-10 and SVHN datasets. Thus, the $D_{\text{KL}}[q_{svhn}(z)||\hat{q}_{cifar}(z)]$ could be smaller than $D_{\text{KL}}[q_{mnist}(z)||\hat{q}_{fashion}(z)]$, which indicates the reason for why PHP works well in FashionMNIST (ID) / MNIST (OOD) but not that well in CIFAR-10 (ID) / SVHN (OOD). This also inspires us that encoding more dataset-specific semantic information into the latent variables could further improve the performance of the PHP method.

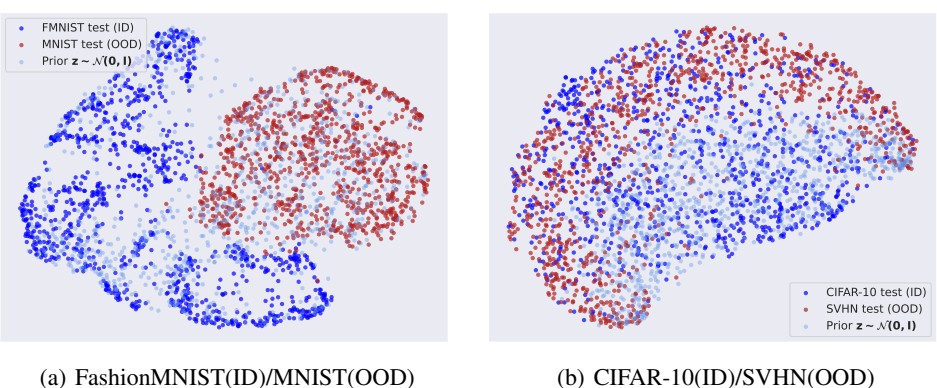

(a) FashionMNIST(ID)/MNIST(OOD)        (b) CIFAR-10(ID)/SVHN(OOD)

Figure 9: The UMAP visualization of latent representations on FashionMNIST(ID) / MNIST(OOD) and CIFAR-10(ID) / SVHN(OOD) dataset pairs.

## K.2 REVERSE TESTIFICATION

Since a good OOD detection method should improve or at least not hurt the performance when exchanging the ID and OOD datasets, we add three "reverse" experiments in Table 12 as an addition to Table 2, where models are trained on MNIST or SVHN and then testified on detecting FashionMNIST or CIFAR-10 as OOD data. The results in Table 12 could support our methods' robustness, because the PHP and DEC are calibration on the ELBO directly, which could improve or at least not hurt the performance of applying ELBO to do OOD detection. In contrast, the baselines HVK Havtorn et al. (2021) and $\mathcal{LLR}^{ada}$ Li et al. (2022) are not based on calibration on the ELBO but rely on the model's ability to extract consistent low-level and high-level information in the hierarchical VAEs. Though it could achieve competitive results once the hierarchical VAE is well-trained, it can hurt the OOD detection performance when the ELBO could already be a perfect OOD detector in cases like MNIST (ID) / FashionMNIST (OOD).

Table 12: **Reverse experiments.** The comparisons of our method with post-hoc prior (denoted as "PHP") or dataset entropy-mutual calibration (denoted as "DEC") individually and other unsupervised OOD detection methods. "PHP+DEC" is equal to our method "AVOID". Bold numbers are superior results.

| MNIST(ID)/FashionMNIST(OOD) | | | | SVHN(ID)/CIFAR-10(OOD) | | | |
|---|---|---|---|---|---|---|---|
| Method | AUROC↑ | AUPRC↑ | FPR80↓ | Method | AUROC↑ | AUPRC↑ | FPR80↓ |
| ELBO (Kingma & Welling, 2014) | **99.9**±0.0 | **99.9**±0.0 | **0.00**±0.0 | ELBO (Kingma & Welling, 2014) | **99.9**±0.0 | **99.9**±0.0 | **0.00**±0.0 |
| HVK (Havtorn et al., 2021) | 80.5±1.1 | 82.6±1.2 | 37.9±0.7 | HVK (Havtorn et al., 2021) | 64.2±1.0 | 58.0±1.2 | 56.8±0.8 |
| $\mathcal{LLR}^{ada}$(Li et al., 2022) | 83.0±1.1 | 86.2±1.1 | 27.0±0.6 | $\mathcal{LLR}^{ada}$(Li et al., 2022) | 34.7±1.4 | 39.7±1.3 | 90.6±0.8 |
| *-Ours:* | | | | *-Ours:* | | | |
| PHP | 99.9±0.0 | 99.9±0.0 | 0.00±0.0 | PHP | 99.9±0.0 | 99.9±0.0 | 0.00±0.0 |
| DEC | 99.9±0.0 | 99.9±0.0 | 0.00±0.0 | DEC | 99.9±0.0 | 99.9±0.0 | 0.00±0.0 |
| PHP+DEC | **99.9**±0.0 | **99.9**±0.0 | **0.00**±0.0 | PHP+DEC | **99.9**±0.0 | **99.9**±0.0 | **0.00**±0.0 |

### K.3 EXPERIMENT ON CIFAR-100

As CIFAR-100 is a more complex dataset compared to the CIFAR-10, testifying our methods' OOD detection performance with a model trained on CIFAR-100 could support our methods' robustness. The results in Table 13 demonstrate our methods' superiority than other baselines in OOD detection. Table 14, Table 15, and Table 16 evaluate the PHP, DEC, and their combination (AVOID)'s individual OOD detection performance on more datasets.

Table 13: Comparison of our methods and baselines of OOD detection performance on CIFAR-100 (ID) / SVHN (OOD). Bold are the best results.

| CIFAR-100 (ID) / SVHN (OOD) | | | |
|---|---|---|---|
| Method | AUROC ↑ | AUPRC ↑ | FPR80 ↓ |
| ELBO (Kingma & Welling, 2014) | 7.68 ±1.12 | 33.3 ± 1.20 | 99.7 ± 0.88 |
| HVK (Havtorn et al., 2021) | 52.3 ±1.65 | 53.6 ±1.30 | 79.9 ±0.75 |
| $\mathcal{LLR}^{ada}$ (Li et al., 2022) | 86.7 ±0.54 | 86.0 ±0.51 | 22.0 ±0.30 |
| -ours | | | |
| PHP | 24.4 ± 1.10 | 37.1 ±1.26 | 96.7 ± 0.56 |
| DEC | 81.8 ± 0.00 | 85.9 ± 0.00 | 31.3 ±0.00 |
| AVOID | **88.3** ± 1.21 | **88.7** ± 1.30 | **17.0** ± 0.53 |

Table 14: Comparison of OOD detection performance between our PHP method and ELBO with models trained on CIFAR-100 (ID). Bold are the best results.

| ID | CIFAR-100 | | |
|---|---|---|---|
| OOD | AUROC ↑ | AUPRC ↑ | FPR80 ↓ |
| | **ELBO / PHP(ours)** | | |
| CelebA | 58.2 / **65.0** | 56.0 / **64.9** | 65.8 / 65.9 |
| Places365 | 56.5 / **69.5** | 55.5 / **67.2** | 74.6 / **52.0** |
| LFWPeople | 63.9 / **74.3** | 58.4 / **72.4** | 61.3 / **46.6** |
| SUN | 58.3 / **60.0** | 55.6 / **57.9** | 61.0 / 60.6 |
| Texture | 52.7 / **55.4** | 48.4 / **51.5** | 66.1 / **64.4** |
| Flowers102 | 80.5 / **84.3** | 80.1 / **80.8** | 23.9 / **23.8** |
| GTSRB | 58.7 / **67.5** | 51.8 / **59.6** | 49.4 / **49.1** |
| Const | 0.00 / **2.40** | 0.00 / **31.8** | 100. / 100. |
| Random | 100. / **100.** | 100. / **100.** | 0.00 / **0.00** |

Table 15: Comparison of OOD detection performance between our DEC method and ELBO with models trained on CIFAR-100 (ID). Bold are the best results.

| ID | CIFAR-100 | | |
|---|---|---|---|
| OOD | AUROC↑ | AUPRC↑ | FPR80↓ |
| | ELBO / DEC(ours) | | |
| CelebA | 58.2 / **72.4** | 56.0 / **67.0** | 65.8 / **45.0** |
| Places365 | 56.5 / **71.7** | 55.5 / **66.5** | 74.6 / **45.8** |
| LFWPeople | 63.9 / **69.3** | 58.4 / **64.1** | 61.3 / **50.7** |
| SUN | 58.3 / **69.3** | 55.6 / **64.1** | 61.0 / **50.7** |
| Texture | 52.7 / **69.3** | 48.4 / **66.0** | 66.1 / **47.4** |
| Flowers102 | 80.5 / **87.3** | 80.1 / **81.0** | 23.9 / **19.1** |
| GTSRB | 58.7 / **73.9** | 51.8 / **68.1** | 49.4 / **41.0** |
| Const | 0.00 / **76.5** | 0.00 / **85.0** | 100. / **21.9** |
| Random | 100. / **100.** | 100. / **100.** | 0.00 / **0.00** |

Table 16: Comparison of OOD detection performance between our AVOID method and ELBO with models trained on CIFAR-100 (ID). Bold are the best results.

| ID | CIFAR-100 | | |
|---|---|---|---|
| OOD | AUROC↑ | AUPRC↑ | FPR80↓ |
| | ELBO / AVOID(ours) | | |
| CelebA | 58.2 / **74.8** | 56.0 / **68.6** | 65.8 / **35.7** |
| Places365 | 56.5 / **80.0** | 55.5 / **74.0** | 74.6 / **27.7** |
| LFWPeople | 63.9 / **81.5** | 58.4 / **75.4** | 61.3 / **26.3** |
| SUN | 58.3 / **70.6** | 55.6 / **62.6** | 61.0 / **39.1** |
| Texture | 52.7 / **75.0** | 48.4 / **65.8** | 66.1 / **31.5** |
| Flowers102 | 80.5 / **92.3** | 80.1 / **87.4** | 23.9 / **9.20** |
| GTSRB | 58.7 / **75.3** | 51.8 / **68.8** | 49.4 / **35.3** |
| Const | 0.00 / **79.8** | 0.00 / **80.8** | 100. / **22.1** |
| Random | 100. / **100.** | 100. / **100.** | 0.00 / **0.00** |

### K.4 EXPERIMENTS ON DIFFUSION MODELS

As the diffusion model is a more powerful and complex deep generative model, which also provides a variational evidence lower bound (ELBO) as an estimation of the data marginal log-likelihood Ho et al. (2020), we testify our method with replace the original backbone (VAE) to a diffusion model. We first directly testify the OOD detection performance of the ELBO of a trained diffusion model with 1000 time steps Ho et al. (2020), *i.e.*, $\{x_t\}_{t=0}^{T=1000}$ where $x_0$ is the input data $x$ and $x_1, ..., x_T$ could be seen as the latent variables. As Table 17 shows, directly applying ELBO of a diffusion model could still suffer from the *overestimation* issue. Therefore, it would be interesting to see whether our method could also alleviate the *overestimation* issue in a diffusion model.

To apply our AVOID method to diffusion models, we need to fit its aggregated posterior distribution $q_{id}(z)$ at first. Let us recall the ELBO of a $T$ time step diffusion model, expressed as

$$\text{ELBO}(x) = \mathbb{E}_{q(x_1|x_0)} \log p_\theta(x_0|x_1) - \sum_{t=2}^{T} D_{\text{KL}}[q(x_{t-1}|x_t, x_0)||p_\theta(x_{t-1}|x_t)] - D_{\text{KL}}[q(x_T|x_0)||p(x_T)],$$

(61)

where $q(\boldsymbol{x}_1, \boldsymbol{x}_2, ..., \boldsymbol{x}_T)$ is decomposed to a product of $T$ terms $q(\boldsymbol{x}_{t-1}|\boldsymbol{x}_t)$ for every time step, *i.e.*,

$$q(\boldsymbol{x}_1, \boldsymbol{x}_2, ..., \boldsymbol{x}_T) = q(\boldsymbol{x}_T) \prod_{t=2}^{T} q(\boldsymbol{x}_{t-1}|\boldsymbol{x}_t). \tag{62}$$

To fit the $q(\boldsymbol{x}_1, \boldsymbol{x}_2, ..., \boldsymbol{x}_T)$ for applying PHP method, we need $T$ individual LSTMs to fit every step's $q(\boldsymbol{x}_{t-1}|\boldsymbol{x}_t)$. However, as the dimension of $\boldsymbol{x}_t$ is the same as the input data, it could be difficult or even impossible to fit the $q(\boldsymbol{x}_{t-1}|\boldsymbol{x}_t)$ well.

Thanks for the reviewers' deep recognition of this difficulty and recommending the variants of the diffusion model, *i.e.*, latent diffusion model. For latent diffusion models Rombach et al. (2022), to improve the computation efficiency, the $\boldsymbol{x}_1$ is replaced by a low-dimension latent variable $\boldsymbol{z}_1$ encoded by a auto-encoder, then the following $\boldsymbol{x}_2, ..., \boldsymbol{x}_T$ is replaced by $\boldsymbol{z}_2, ..., \boldsymbol{z}_T$. Therefore, the ELBO of the latent diffusion model is expressed by

$$\text{ELBO}(\boldsymbol{x}) = \mathbb{E}_{q_\phi(\boldsymbol{z}_1|\boldsymbol{x}_0)} \log p_\theta(\boldsymbol{x}_0|\boldsymbol{z}_1) - \sum_{t=2}^{T} D_{\text{KL}}[q(\boldsymbol{z}_{t-1}|\boldsymbol{z}_t, \boldsymbol{x}_0)||p_\theta(\boldsymbol{z}_{t-1}|\boldsymbol{z}_t)] - D_{\text{KL}}[q(\boldsymbol{z}_T|\boldsymbol{x}_0)||p(\boldsymbol{z}_T)], \tag{63}$$

and the latent $q(\boldsymbol{z}_1, \boldsymbol{z}_2, ..., \boldsymbol{z}_T)$ is decomposed to a product of $T$ terms $q(\boldsymbol{z}_{t-1}|\boldsymbol{z}_t)$ for every time step, *i.e.*,

$$q(\boldsymbol{z}_1, \boldsymbol{z}_2, ..., \boldsymbol{z}_T) = q(\boldsymbol{z}_T) \prod_{t=2}^{T} q(\boldsymbol{z}_{t-1}|\boldsymbol{z}_t). \tag{64}$$

To fit the $q(\boldsymbol{z}_1, \boldsymbol{z}_2, ..., \boldsymbol{z}_T)$ for applying PHP method, we need $T$ individual LSTMs $\hat{q}_{id}(\boldsymbol{z}_{t-1}|\boldsymbol{z}_t)$ to fit every step's $q(\boldsymbol{z}_{t-1}|\boldsymbol{z}_t)$ on the training set and then change the ELBO to

$$\text{PHP}(\boldsymbol{x}) = \log p_\theta(\boldsymbol{x}_0|\boldsymbol{z}_1) - \sum_{t=2}^{T} D_{\text{KL}}[q(\boldsymbol{z}_{t-1}|\boldsymbol{z}_t, \boldsymbol{x}_0)||\hat{q}_{id}(\boldsymbol{z}_{t-1}|\boldsymbol{z}_t)] - D_{\text{KL}}[q(\boldsymbol{z}_T|\boldsymbol{x}_0)||\hat{q}_{id}(\boldsymbol{z}_T)]. \tag{65}$$

However, the $T$ could still be too large, *e.g.*, when $T = 1000$, we still need to train 1000 individual models to fit $q(\boldsymbol{z}_{t-1}|\boldsymbol{z}_t)$, which we think is very awkward and low-efficiency in computation. Thus, we consider a shallow latent diffusion model, termed "Shallow-Diffusion", where the $T$ is set as 10. Besides, we let it directly maximize the ELBO as its training objective with a trainable encoder, which could be also interpreted as a hierarchical VAE Luo (2022). With a much more complex backbone than VAE, we report the OOD detection results in Table 17. As results show, our method could still largely alleviate the *overestimation* issue.

Table 17: The comparisons of our method with a 10-layer latent diffusion model, termed "Shallow-Diffuion", with post-hoc prior (denoted as "PHP") or dataset entropy-mutual calibration (denoted as "DEC") individually and other unsupervised OOD detection methods. "PHP+DEC" is equal to our method "AVOID". Bold numbers are superior results.

| FashionMNIST(ID)/MNIST(OOD) | | | | CIFAR-10(ID)/SVHN(OOD) | | | |
|---|---|---|---|---|---|---|---|
| Method | AUROC↑ | AUPRC↑ | FPR80↓ | Method | AUROC↑ | AUPRC↑ | FPR80↓ |
| Diffusion | 18.0 | 34.1 | 97.4 | Diffusion | 19.3 | 35.0 | 97.2 |
| Shallow-Diffusion | 11.5 | 34.6 | 98.6 | Shallow-Diffusion | 4.11 | 31.14 | 99.80 |
| PHP | 80.9 | 83.5 | 29.5 | PHP | 43.0 | 46.0 | 86.2 |
| DEC | 34.7 | 39.7 | 90.6 | DEC | 88.9 | 90.1 | 12.8 |
| PHP+DEC | **85.9** | **87.5** | **19.5** | PHP+DEC | **95.0** | **95.1** | **4.11** |

## K.5 Computation efficiency

We conduct comparisons for the computation efficiency from two perspectives: inference time and training time, where results are shown in Table 18 and Table 19, respectively.

First, we testify to the computation efficiency regarding the inference by recording the average inference time in scoring a data point on an A100 GPU. The time is averaged by sampling 100 data points

from each of all the used gray-scale datasets or RGB nature datasets. Thanks to the reviewer's suggestion, in addition to the originally implemented DEC method based on SVD, termed "DEC(SVD)", we implement three different variants of the DEC method for comparison: DEC(SVD-bs), DEC(JPEG), and DEC(PNG). "DEC(SVD-bs)" means a binary search version of the "DEC(SVD)", where the $n_i$ is determined by binary searching from 1 to $2 * n_i d$. "DEC(JPEG)" and "DEC(PNG)" are implementing DEC with different compressors, JPEG compressor and PNG compressor. They could provide a value $bit(\boldsymbol{x})$ indicating the complexity of an image $\boldsymbol{x}$, then we could get the average $bit(\boldsymbol{x})$ of the training set, termed "$bit_{id}$". Thus, we could replace the $n_i$ with $bit(\boldsymbol{x})$ to implement the $\mathcal{C}_{non}(\boldsymbol{x})$ DEC method as follows:

$$\mathcal{C}_{non}(\boldsymbol{x}) = \begin{cases} (bit(\boldsymbol{x})/bit_{id}), & \text{if} \quad bit(\boldsymbol{x}) < bit_{id}, \\ (bit_{id} - (bit(\boldsymbol{x}) - bit_{id}))/bit_{id}, & \text{if} \quad bit(\boldsymbol{x}) \geq bit_{id}, \end{cases} \tag{66}$$

which could avoid the low-efficiency searching of $n_i$ in the DEC(SVD) method.

For the ensemble method WAIC Choi et al. (2018), it relies on an ensemble of $N$ generative models $\{\log p_{\theta_i}(\boldsymbol{x})\}_{i=1}^N$ to detect the OOD data with a score function:

$$\mathcal{S}^{waic}(\boldsymbol{x}) = \mathbb{E}_i(\log p_{\theta_i}(\boldsymbol{x})) - \text{Var}_i(\log p_{\theta_i}(\boldsymbol{x})). \tag{67}$$

Though the time cost in scoring with Eq. 67 could be ignored, its computation efficiency is much lower than our methods since it needs to train $N$ independent models and run inference for all the models to get the score. To achieve better performance, the ensemble number $N$ should be large enough that leads to worse computation efficiency, *e.g.*, when $N = 5$, we need 5 times the computation resource of the ELBO method to get such 5 models and run the inference.

As the results shown in Table 18, our method AVOID with SVD implementing the DEC method is only approximately 1.5x computation time than the vanilla ELBO, which is still fast. For the training efficiency shown in Table 19, the PHP method still does not need much computation resources. For the computation efficiency of the ensemble method WAIC, it largely depends on the computation resource.

Table 18: Computation efficiency under the metric of average inference time for a data example on an NVIDIA A100 GPU. The inference time for ensemble methods means it could cost the least time if all the models are doing inference at the same time with sufficient computation resources like many GPUs but will cost $N$ times longer than it if only one model of the ensemble could be inferred at one time due to limited computation resources.

| Avg. inference time (ms) for a data example | | |
|---|---|---|
| Method | Gray-scale data | RGB nature data |
| ELBO | 9.33 | 9.58 |
| Ensemble of $N$ VAEs | $(9.33, N \times 9.33]$ | $(9.58, N \times 9.58]$ |
| PHP | 10.6 | 11.0 |
| DEC(SVD) | 13.3 | 14.4 |
| DEC(SVD-bs) | 11.0 | 11.6 |
| DEC(JPEG) | 10.0 | 11.4 |
| DEC(PNG) | 10.1 | 10.7 |
| AVOID(SVD) | 14.3 | 14.6 |

Table 19: Computation efficiency under the metric of average training time for different methods on an NVIDIA A100 GPU. The training time for ensemble methods means it could cost the least time if all the models are trained at the same time with sufficient computation resources like many GPUs but will cost $N$ times longer than it if only one model of the ensemble could be trained at one time due to limited computation resources.

| Avg. training time of a model | | |
|---|---|---|
| | Gray-scale data | RGB nature data |
| ELBO | $\approx 5h$ | $\approx 7h$ |
| PHP | $\approx 0.5h$ | $\approx 0.5h$ |
| Ensemble of $N$ VAEs | $\approx 5h \sim N \times 5h$ | $\approx 7h \sim N \times 7h$ |

## K.6  DIFFERENT COMPRESSOR

We kindly emphasize that the SVD is only one of the choices for implementing the DEC method, which is simply implemented to support our analysis of the *overestimation* issue. Other compressors like JPEG and PNG compressors could also be directly applied to implement the DEC method. We compare their performance in Table 20. Actually, we find that the relationship between the data type and compressor plays an important role in its performance, *e.g*, the ratio of the average bit, *bits*(FashionMNIST)/*bits*(MNIST), compressed by JEPG is only 1.12, which leads to its poor performance; in contrast, the ratio of the average bit, *bits*(FashionMNIST)/*bits*(MNIST), compressed by PNG is 1.92, which is much larger and leads to its best performance on this dataset pair.

Table 20: OOD detection performance of DEC method with different compressors.

| FashionMNIST(ID)/MNIST(OOD) | | | | CIFAR-10(ID)/SVHN(OOD) | | | |
|---|---|---|---|---|---|---|---|
| Method | AUROC↑ | AUPRC↑ | FPR80↓ | Method | AUROC↑ | AUPRC↑ | FPR80↓ |
| SVD | 34.1 | 40.7 | 92.5 | SVD | 87.8 | 89,9 | 17.8 |
| JPEG | 31.1 | 36.6 | 93.0 | JPEG | 90.4 | 88.6 | 12.5 |
| PNG | 67.2 | 68.6 | 45.8 | PNG | 80.5 | 75.4 | 26.4 |

## K.7  EVALUATION ON NOISY DATA

It would be interesting to add different scales of random noise to the original in-distribution data and see what would happen in detecting the noisy data as OOD data. Intuitively, adding random noise to the data could largely increase the entropy of the distribution of in-distribution data, which could lead to better OOD detection performance.

We demonstrate the changing process when we gradually add noise to the data with scale $\alpha$ from 0 to 1, expressed as

$$\boldsymbol{x}_{noisy} \leftarrow \boldsymbol{x} + \alpha \times \epsilon, \epsilon \sim \mathcal{N}(0, \mathbf{I}). \tag{68}$$

As Table 21 and Table 22 show, along with the scale of noise increases, the OOD detection performance of all methods becomes better, supporting our analysis of the factor 2 for the *overestimation* issue. To be more specific, $\mathcal{H}_{p_{noisy}}(\boldsymbol{x}_{noisy}) > \mathcal{H}_{p_{ID}}(\boldsymbol{x}_{ID})$ contributes to Ent-Mut$(\theta, \phi, p_{noisy}) >$ Ent-Mut$(\theta, \phi, p_{ID})$, and further leads to the gap $\mathcal{G} > 0$ in Eq. 8, *i.e.*, the *overestimation* issue does not occur in these cases.

Table 21: OOD detection performance under the metric AUROC ↑ on the FashionMNIST(ID)/Noisy-FashionMNIST(OOD) dataset pair in different noisy scales. "Comp." denotes the ratio of average bits between ID and OOD datasets, *i.e.*, $bits$(Fashion)/$bits$(Noisy-Fashion), where the bits are measured by JPEG compressor.

| FashionMNIST(ID) / Noisy-FashionMNIST(OOD) | | | | | |
|---|---|---|---|---|---|
| noise scale | ELBO | PHP | DEC | AVOID | Comp. |
| 0.0 | 50.0 | 50.0 | 50.0 | 50.0 | 1.0 |
| 0.1 | 92.0 | 93.8 | 99.4 | 99.7 | 1.26 |
| 0.2 | 97.7 | 98.2 | 99.5 | 100. | 1.38 |
| 0.3 | 99.4 | 99.5 | 99.5 | 100. | 1.46 |
| 0.4 | 99.9 | 99.9 | 99.9 | 100. | 1.53 |
| 0.5 | 99.9 | 99.9 | 99.9 | 100. | 1.58 |
| 0.6 | 100. | 100. | 100. | 100. | 1.63 |
| 0.7 | 100. | 100. | 100. | 100. | 1.65 |
| 0.8 | 100. | 100. | 100. | 100. | 1.67 |
| 0.9 | 100. | 100. | 100. | 100. | 1.68 |
| 1.0 | 100. | 100. | 100. | 100. | 1.69 |

Table 22: OOD detection performance under the metric AUROC ↑ on the CIFAR(ID)/Noisy-CIFAR-10(OOD) dataset pair in different noisy scales. "Comp." denotes the ratio of average bits between ID and OOD datasets, *i.e.*, $bits$(CIFAR10)$/bits$(Noisy-CIFAR10), where the bits are measured by JPEG compressor.

| noise scale | CIFAR10(ID) / Noisy-CIFAR10(OOD) | | | | |
|---|---|---|---|---|---|
| | ELBO | PHP | DEC | AVOID | Comp. |
| 0.0 | 50.0 | 50.0 | 50.0 | 50.0 | 1.0 |
| 0.1 | 94.3 | 94.7 | 98.1 | 98.5 | 1.26 |
| 0.2 | 99.2 | 99.4 | 99.9 | 99.9 | 1.39 |
| 0.3 | 99.9 | 99.9 | 99.9 | 99.9 | 1.48 |
| 0.4 | 100. | 100. | 100. | 100. | 1.53 |
| 0.5 | 100. | 100. | 100. | 100. | 1.57 |
| 0.6 | 100. | 100. | 100. | 100. | 1.59 |
| 0.7 | 100. | 100. | 100. | 100. | 1.61 |
| 0.8 | 100. | 100. | 100. | 100. | 1.63 |
| 0.9 | 100. | 100. | 100. | 100. | 1.64 |
| 1.0 | 100. | 100. | 100. | 100. | 1.65 |

## K.8 DISCUSSION OF SET-LEVEL AND INSTANCE-LEVEL DEFINITION OF *overestimation*

We provide a discussion of a more strict instance-level definition of *overestimation* here. Let's still start from a toy example, assuming there is an ID distribution $x \sim p_{id}(\boldsymbol{x})$, *e.g.*, $\mathcal{N}(10, 5)$, and also an OOD distribution $x \sim p_{ood}(\boldsymbol{x})$, *e.g.*, $\mathcal{N}(9, 5)$, we need to develop an ID-OOD classifier $D(\boldsymbol{x})$ to determine a data sample is OOD or not, specifically $D(\boldsymbol{x})$ = ID if $\mathcal{S}(\boldsymbol{x}) > T$ or $D(\boldsymbol{x})$ = OOD if $\mathcal{S}(\boldsymbol{x}) \leq T$, where $T$ could be a threshold such that 95% of ID samples are classified as ID.

The core of designing an ideal ID-OOD classifier for unsupervised OOD detection is the choice of score function $\mathcal{S}(\boldsymbol{x})$. Under the context of generative models, we expect $\mathcal{S}(\boldsymbol{x})$ to own the property $\mathcal{G} = \mathbb{E}_{\boldsymbol{x} \sim p_{id}(\boldsymbol{x})}[\mathcal{S}(\boldsymbol{x})] - \mathbb{E}_{\boldsymbol{x} \sim p_{ood}(\boldsymbol{x})}[\mathcal{S}(\boldsymbol{x})] > 0$, so that most of ID samples can be classified accurately after setting a threshold $T$. We admit there will still be some ID/OOD samples to be misclassified even if $\mathcal{G} > 0$ and the selection of $T$ can lead to varying degrees of misclassification, but from the perspective of the mean value of the data distribution, a larger $\mathcal{G} > 0$ indicates that $\mathcal{S}(\boldsymbol{x})$ tends to assign higher scores to most of ID samples than that of OOD samples.

The aforementioned content is the original statement in our first version of the manuscript. Then, as for the stricter case you mentioned, we have $p(\mathcal{S}(\boldsymbol{x}) > T | x \sim p_{id}(\boldsymbol{x}))$ and $p(\mathcal{S}(\boldsymbol{x}) > T | x \sim p_{ood}(\boldsymbol{x}))$, and target of designing $\mathcal{S}(\boldsymbol{x})$ is to make $p(\mathcal{S}(\boldsymbol{x}) > T | x \sim p_{id}(\boldsymbol{x}))$ large but $p(\mathcal{S}(\boldsymbol{x}) > T | x \sim p_{ood}(\boldsymbol{x}))$ small. In other words, we need to make the confidence of classifying ID samples with ID labels larger than that of OOD samples, specifically $p(\mathcal{S}(\boldsymbol{x}) > T | x \sim p_{id}(\boldsymbol{x})) > p(\mathcal{S}(\boldsymbol{x}) > T | x \sim p_{ood}(\boldsymbol{x}))$, also equals to $p(\mathcal{S}(\boldsymbol{x}) > T | x \sim p_{id}(\boldsymbol{x})) - p(\mathcal{S}(\boldsymbol{x}) > T | x \sim p_{ood}(\boldsymbol{x})) > 0$.

At the **instance level, an ideal $\mathcal{S}(\boldsymbol{x})$ is to make the lowest value of ID samples larger than the highest value of OOD samples.** At the **set level** (or distribution level), an ideal $\mathcal{S}(\boldsymbol{x})$ is to enlarge the distance between the distributions of ID and OOD samples, under the condition that the ID samples tend to own higher $\mathcal{S}(\boldsymbol{x})$ scores. In our work, we tend to focus on the second scenario to enlarge the distribution distance, which can be instantiated as the distance between mean values of distributions. Thus, in our case, the stricter defination $p(\mathcal{S}(\boldsymbol{x}) > T | x \sim p_{id}(\boldsymbol{x})) - p(\mathcal{S}(\boldsymbol{x}) > T | x \sim p_{ood}(\boldsymbol{x})) > 0$ can be also transfered into $\mathcal{G} = \mathbb{E}_{\boldsymbol{x} \sim p_{id}(\boldsymbol{x})}[\mathcal{S}(\boldsymbol{x})] - \mathbb{E}_{\boldsymbol{x} \sim p_{ood}(\boldsymbol{x})}[\mathcal{S}(\boldsymbol{x})] > 0$ during the implementation.

Thus, we assume that the following theoretical analysis, which is focused on the set-level definition in the manuscript, can still hold even with the stricter definition. To be more specific, the two factors, 1) the mismatch between $q(\boldsymbol{z})$ and $p(\boldsymbol{z})$ and 2) the gap in dataset entropy-mutual integration, could still cause some out-of-distribution (OOD) data samples to be scored higher than some in-distribution (ID) data with the score function as the ELBO.

As for the **instance-level** theoretical analysis, we admit it remains a great challenge in the field of OOD detection, especially for unsupervised OOD detection with generative models. We are very willing to challenge this problem in the future.

