# OpenReview forum: "AVOID: Alleviating VAE's Overestimation in Unsupervised OOD Detection"
_ICLR.cc/2024/Conference — Submitted to ICLR 2024_

### Official Review · Reviewer_F6GH · 2023-10-22

**Soundness:** 3 good
**Presentation:** 3 good
**Contribution:** 3 good
**Rating:** 6
**Confidence:** 4

**Summary:**

The paper discusses the over-estimation problem in VAEs: VAEs often assign a higher likelihood to OOD datapoints than ID datapoints. In analyzing this issue, the paper brings forth two different possible reasons, namely (1) the prior choice $p(z)$ being Gaussian, (2) the entropy of ID data being much higher than OOD data. The paper does this analysis by decomposing the ELBO, and then proposes two approaches to mitigate these two factors. Combining these two approaches give an unified way of handling a VAE’s overestimation problem, and the authors evaluate their approach in a suite of unsupervised OOD detection tasks.

**Strengths:**

1. The decomposition of the ELBO, and the two factors given for VAEs over-estimation, is novel to the best of my knowledge. It is also a clever way of analyzing the problem.
2. Section 3.2, further analysis on Factor I. I like the simple to complex examples provided to show how $p(z)$ being a standard normal distribution can be an improper choice of prior when modeling a complex distribution.
3. The paper is well written and easy to read.
4. The authors have conducted a thorough set of experiments and ablations.

**Weaknesses:**

**(Motivation)** Could the authors explain the motivation behind choosing unsupervised OOD detection and specifically VAEs? Typically, there are no shortage of labeled datapoints from the training distribution, so a few use-cases would be helpful. Also is there a particular reason for focusing on VAEs? Some prior successful unsupervised OOD detection method, such as DoSE [1], works on both VAEs and Glow, and LMD [2] uses diffusion models for OOD detection (diffusion models being the typical generative model of choice over VAEs these days).

**(Notation, section 2.1)** Why is $x = \textrm{ID}$ or $x = \textrm{OOD}$? A better notation is $D(x) = \textrm{ID}$ if $S(x) > T$, and so on, where $D$ is the ID-OOD classifier.

**(Definition 1, VAE’s overestimation)** The authors define over-estimation when the expected ELBO over the OOD distribution is larger than that on the ID distribution. This however, is a weak definition in the sense that it gives no guarantee about arbitrary samples from these distributions. For example, it is quite possible that due to overlaps between ELBO on ID and OOD data, there is no overestimation but a big fraction of the ID/OOD samples are misclassified.

For example, assume that the score function on the ID distribution has a distribution $N(10, 5)$, and that on the OOD distribution has a distribution of $N(9, 5)$. Then by definition 1, there is no overestimation issue in this case. However, it is easy to see that if we choose the threshold $T$ such that 95% of ID samples are classified as ID, then a big percentage of OOD samples would also be classified as ID. A better definition of over-confidence would take into account the threshold $T$, $P(S(x) < T | x \sim p_{id})$ and $P(S(x) < T | x \sim p_{ood})$.

While I understand that this definition is chosen to facilitate the theoretical discussion/make the proofs easier, it is important to acknowledge this weakness in the paper.

**(More notations)** I am assuming $p_{\theta}(x|z)$ is the decoder distribution, and $p(x)$ represents the true distribution of $x$. This needs to be clearly stated and used in a careful manner.

**(Equation 5)** Could the authors clarify the term $H_{q, p}(z|x) = -E_{p(x)q_{\phi}(z|x)} [log(p_{\theta}(z|x)]$ (equation 5)? The expectation is taken over the true distribution $p(x)$ but the term inside is the model distribution $p_{\theta}(z|x)$. This is not the conditional entropy of the variable $z|x$ in the usual sense, and some explanation/clarification of notation would be useful. It is possible the model distribution is not close to the true distribution. Is there some sort of assumption that they are?

**(Contribution 1 in introduction: dataset entropy mutual integration)** In the introduction, dataset entropy mutual integration is coined as:

>> “sum of the dataset entropy and the mutual information terms between the inputs and latent variables”

However, in page 4, we see that:

$$\textrm{Ent-mut} = H_p(x) + I_q(x, z) - I_{q, p}(x, z)$$

So it is not technically a sum of entropy and mutual information, as we subtract $I_{q, p}(x, z)$ in the expression. Also what is the relevance/meaning of $I_q(x, z) - I_{q, p}(x, z)$?

**(Table 1 and 2)** No error bar or uncertainty estimation is given. A lot of the methods have similar numbers, and without an error bar, it is hard to discern the results. I would request the authors to

Run the experiments for 3 seeds, for each baseline, and report the standard error.
Bold the top performing method, and also any method whose average performance > lower bound on the top performing method’s performance. Any equivalent formulation is also good.

**I see that Table 10 and 11 in the appendix have the associated error bars**, but mentioning/referencing them in the main paper would be important.

**(Lack of self-containedness)**

I was looking for limitations of the paper, and it is mentioned in the appendix K. I would request this to be moved to the conclusion section to make the paper more self-contained.


**(Nit: overclaiming)**

Section 3.2
>> the prior distribution $p(z) = N(0, I)$ is an improper choice for VAE when modeling a complex data distribution $p(x)$

This is overclaiming:
1. what is the measure of complexity of $p(x)$? Its entropy/differential entropy?
2. We have seen some examples when the prior being $p(z) = N(0, I)$ leads to a bad outcome. This does not prove this statement in a general sense. Better way to say this, use “may be an improper choice” instead of “is an improper choice”, unless the authors have a more specific theorem to present.

[1] Density of States Estimation for Out-of-Distribution Detection, https://arxiv.org/abs/2006.09273

[2] Unsupervised Out-of-Distribution Detection with Diffusion Inpainting, https://openreview.net/forum?id=HiX1ybkFMl

**Questions:**

**(Understanding Post-Hoc prior)**

Just to make sure I understand the method correctly: for PHP, one first trains a VAE using ELBO from equation 2:

$$ELBO(x) = E_{q_{\phi}(z|x)} [\text{log}p_{theta}(x|z)] - D_{KL}(q_{\phi}(z|x)||p(z))$$

And only once this is done, PHP trains a second LSTM to learn $\hat{q_{id}}(z)$ to match the learned $q_{id}(z)$ from the regular VAE training?

**(Error related to LSTM estimation)**

Does the method assume that the LSTM learned distribution $\hat{q_{id}}(z))$ matches $q_{id}(z)$? What happens when this match is not well, or $D_{KL}(\hat{q_{id}}(z)||q_{id}(z))$ is high? Then it seems that PHP should not work. This seems like a key assumption for PHP to work well.

**(Table 1)**

Why are the methods in supervised/auxiliary column different between FashionMNIST/MNIST and CIFAR-10/SVHN?


**(Table 10 and 11)**

Why are the error rates of DEC 0 in table 10 and 11?


**(Additional experimental results)**

Would it be possible to produce table 1, but with CIFAR-100 used as the ID dataset? Most OOD detection papers report numbers on CIFAR-100, and it is regarded as a harder task than CIFAR-10.


**(Computational resources)**

How much additional computation time is required for this method, including training a separate LSTM for PHP? How does this compare with other baselines?

---

> ### Author Response · Authors · 2023-11-22
> **(Part 1) Thanks for your insightful and constructive feedback!**
>
> **Motivation:** Thanks for giving us this chance to explain our motivation for focusing on unsupervised OOD detection.
>
> 1) A **real-world use case** could be autonomous driving and we want to identify whether the current driving state is similar or strange to the training set for ensuring the safety of decision-making.
> Although we could collect a lot of images or videos of driving in different scenarios, it would cost huge manual efforts to annotate each frame with a proper label.
> In that case, unsupervised OOD detection could avoid the labeling process and directly model the training set's distribution. Actually, under the setting of offline RL, a core direction of research focus is to identify the out-of-distribution state-action pairs and assign them a lower reward [1]. Moreover, as the RL setting always has a lot of states, e.g., millions of time steps, it could be impossible to label every state.
>
> 2) **Motivation for VAE**: VAE is estimating a lower bound (ELBO) of the marginal data log-likelihood unlike exact likelihood models like flow models and autoregressive models, thus the analyses on the relationship between ELBO and OOD detection performance are challenging and lacking. We tried to fill this research gap by identifying the two factors that contribute to the overestimation issue in ELBO-based OOD detection.
> Besides, we need to highlight that the training objective of diffusion models is also based on ELBO [2, 3], thus our analysis and proposed methods could be directly applied to diffusion models, where we have included the experiments on diffusion models in Appendix K.4.
>
>    [1] Fujimoto, Scott, David Meger, and Doina Precup. “Off-policy deep reinforcement learning without exploration.” ICML, 2019.
>
>    [2] Ho, Jonathan, Ajay Jain, and Pieter Abbeel. "Denoising diffusion probabilistic models." NeurIPS 2020.
>
>    [3] Luo, Calvin. "Understanding diffusion models: A unified perspective." arXiv preprint arXiv:2208.11970 (2022).
>
> **Notation:** Thanks for your suggestion and we have revised the notation with blue color.
>
> **Definition 1:**
> Thanks for your thoughtful suggestions.
> We highly agree with your statement that even if $\mathcal{G}=\mathbb{E}\_{x\sim p_{id}}[\text{ELBO(x)}] - \mathbb{E}\_{x\sim p_{ood}}[\text{ELBO(x)}]>0$, there is still a certain chance that the overestimation issue could still happen on some OOD data samples.
> However, if $\mathcal{G}\leq 0$, most of the OOD data samples will be assigned unexpectedly higher ELBO values, which could be seen as **a worst case** of the failures for ELBO-based OOD detection.
> Thus, we propose to address this issue $\mathcal{G}\leq 0$ at first to develop a promising detection method for ELBO-based generative models, which is our focus in this paper.
>
> Thanks for your constructive suggestions. We agree that a definition of over-confidence would be a stricter definition and we will add it in our revision.
> We also appreciate your understanding that the original definition is to make our theoretical analysis easier to understand, but we need to highlight that the conclusion of theoretical analysis in our paper still holds even if the definition is changed.
>
> **More notations:** Yes, $p_\theta(x|z)$ denotes the decoder distribution paramterized with $\theta$ and $p(x)$ is the true distribution of $x$.
> We will highlight their definitions in our revision, and we need to highlight that the estimated data distribution $p_\theta(x)=\int_z p_\theta(x|z)p(z)$ may not be equal to $p(x)$ if the VAE is not a perfect generative model.

---

> > ### Author Response · Authors · 2023-11-22
> > **(Part 2)**
> >
> > **Equation 5:**
> > Thanks for your thoughtful suggestion.
> > It is correct that the definition of $\mathcal{H}\_{p,q}(z|x)$ is not consistent with its definition in the usual sense, because it is a symbol with a specific definition in our paper.
> > Sorry for this misunderstanding and we have revised the notation of $\mathcal{H}\_{q,p}$, $\mathcal{I}\_{q,p}$, $\mathcal{H}\_{q}$, $\mathcal{I}\_{q}$ to $\hat{\mathcal{H}}\_{q,p}$, $\hat{\mathcal{I}}\_{q,p}$, $\hat{\mathcal{H}}\_{q}$, $\hat{\mathcal{I}}\_{q}$ with blue color in the revised paper.
> >
> > Here is a clarification of the notations:
> > The usual definition of $\mathcal{I}\_{q,p}(x,z)$ is
> > $$\mathcal{I}\_{q,p}(x,z) = \int_x\int_z p(x)q_\phi(z|x)\log \frac{q_\phi(z|x)p(x)}{p(x)q_\phi(z)}=\mathbb{E}\_{p(x)q_\phi(z|x)}\log q_\phi(z|x) - \mathbb{E}\_{q_\phi(z)}\log q_\phi(z)=-\mathcal{H}\_{q,p}(z|x) + \mathcal{H}\_{q}(z)$$,
> >
> > The purpose of **introducing the $\hat{\mathcal{I}}_{q,p}(x,z)$ is ONLY for a brief notation of the terms in the expectation of the ELBO**, i.e.,
> > $$\hat{\mathcal{I}}\_{q,p}(x,z) := \mathbb{E}\_{p(x)q_\phi(z|x)}[\log \frac{p_\theta(z|x)}{p(z)}]$$,
> > which is not equal to $\mathcal{I}\_{q,p}(x,z)$ unless $p_\theta(z|x)=q_\phi(z|x)$ and $p(z)=q_\phi(z)$. Note that $q(z)$ is a brief notation of $q_\phi(z)$ in the paper.
> >
> > Similarly, we use $\hat{\mathcal{I}}\_{q}(x,z)$ for a brief notation of another term in the expectation of the ELBO, i.e.,
> > $$\hat{\mathcal{I}}\_{q}(x,z) := \mathbb{E}\_{p(x)q_\phi(z|x)}[\log\frac{q_\phi(z|x)}{q(z)}]$$,
> > which is not equal to $\mathcal{I}\_q(x,z)$ unless $q_\phi(x)=p(x)$.
> >
> > We have revised the definition in our manuscript and we emphasize that **the revision is **NOT** related to the effectiveness of our analysis and proposed methods.**
> >
> > **Dataset entropy mutual intergration & meaning of $\mathcal{I}\_q(x,z)-\mathcal{I}\_{q,p}(x,z)$:**
> > Yes, following the above response, it would be more precise to say "a sum of entropy and mutual information gap" and $\hat{\mathcal{I}}\_{q,p}(x,z)-\hat{\mathcal{I}}\_{q}(x,z)$ represents the gap between the $q_{\phi}(z|x)$ and $p_{\theta}(z|x)$, $q(z)$ and $p(z)$.
> >
> > **Table 1 and 2:**
> > Thanks for the notification, we have included the error bars in Table 2 (FashionMNIST(ID)/MNIST(OOD) and CIFAR10(ID)/SVHN(OOD) dataset pairs) and Table 5 of Appendix F (CelebA(ID)/CIFARs(OOD)). Besides, we have also added additional experiments on CIFAR100(ID)/SVHN(OOD) in Table 13 of Appendix K.3.
> > All results of our methods and baselines are obtained by randomly running 5 times. From the results, the performance of all these methods is relatively stable.
> >
> > Thanks for your suggestion on bolding the methods whose average performance is better than the lower bound on the top-performing method’s performance.
> >
> > **Lack of self-containedness:**
> > Thanks for your suggestion and we have included the limitation of our methods in the conclusion section, which has been noted with a blue color.
> >
> >
> > **Overclaiming**
> > Wow, we sincerely appreciate your efforts in reviewing and improving the quality of our paper.
> > And we have modified this sentence to "may be an improper choice" to make the statement more rigorous.
> >
> > As for the measure of complexity of $p(x)$, in our understanding, we may empirically use the entropy or the number of modes to measure the complexity of the given distribution.
> >
> > **Understanding post-hoc prior:**
> > Yes, your understanding is correct.
> > After training a VAE using ELBO, the PHP method will secondly train an LSTM to learn $\hat{q}\_{id}(z)$ to match the aggregated posterior distribution $q_{id}(z)$ of latent representations of ID data samples.
> >
> > **Error related to LSTM:**
> > Actually, it's not necessary to have a perfect matching to achieve decent OOD detection performance, because the key insight is to enlarge $D_{\text{KL}}[q_{ood}(z)||\hat{q}\_{id}(z)]$ but reduce $D_{\text{KL}}[q_{id}(z)||\hat{q}\_{id}(z)]$, where a normal approximator is enough to bring promising benefits to OOD detection performance.
> > Moreover, we admit that a perfect approximator can bring further performance gains.
> >
> > **Table 1:**
> > Thanks for your careful review.
> > To make a comparison, most of the results are cited from other papers directly to make readers understand that the unsupervised methods could achieve comparable performance with the supervised/auxiliary ones.
> >
> > **Table 10 & 11:**
> > SVD (Singular Value Decomposition) is a deterministic method without randomness.
> > Therefore, the experimental results of multiple rounds of DEC with SVD are the same, resulting in an error rate of 0.

---

> > > ### Author Response · Authors · 2023-11-22
> > > **(Part 3)**
> > >
> > > **Additional experiments on CIFAR-100**
> > > Yes, thanks for your suggestion and we have included the experimental results on the CIFAR-100 dataset in Appendix K.3.
> > > From the results, we can find that our methods can still outperform other baselines.
> > >
> > >
> > > **Computation resources**
> > >
> > > Thanks, we have included the analysis of computation efficiency in Appendix K.5, including the time cost of training a separate LSTM and also calculating the values of PHP and DEC terms.
> > > Briefly, considering the dimension of latent representation will be much less than that of raw feature, the time cost of training an LSTM on the latent space will be much less than that of training a VAE on the input data space, as shown in Table 19.
> > > As for the computation time of calculating the values of PHP and DEC terms, from the results shown in Table.18, we can find that the computation time for calculating AVOID is only 1.5 times that of ELBO, which indicates that our method won't bring too much additional computation cost. Besides, it could be faster to implement DEC with other methods like different compressors and binary searching schemes.
> > >
> > > ---------
> > >
> > > Lastly, we want to express our deepest gratitude for your meticulous and insightful comments, which have significantly enhanced the quality of our work. If there are any further questions or concerns, we are eager and ready to address them.

---

> > > ### Comment · Reviewer_F6GH · 2023-11-22
> > > **Further comment to authors**
> > >
> > > Dear authors,
> > >
> > > Thanks for taking the time to go through my comments and addressing them. I have two more questions/suggestions:
> > >
> > > **(Further Question 1)**
> > > > Thanks for your constructive suggestions. We agree that a definition of over-confidence would be a stricter definition and we will add it in our revision. We also appreciate your understanding that the original definition is to make our theoretical analysis easier to understand, but we need to highlight that the conclusion of theoretical analysis in our paper still holds even if the definition is changed.
> > >
> > > Could you clarify what you mean by the conclusion of theoretical analysis remains the same? Could you highlight the modified definition according to what I said, the conclusion of the analysis precisely, and some argument for why it would remain the same? Sorry to be strict about this, I just want to make sure we differentiate between what holds strictly/formally (if this, then that), and what is more of an intuitive understanding (if this, then intuitively this makes sense).
> > >
> > > **(Further Question 2)**
> > > > Wow, we sincerely appreciate your efforts in reviewing and improving the quality of our paper. And we have modified this sentence to "may be an improper choice" to make the statement more rigorous.
> > >
> > > I think a more formal statement here, with some simplified toy setup, would make the paper really strong. Maybe assume that the true distribution $p(x)$ is a mixture of Gaussians, and in this case the prior distribution $p(z) \sim N(0, I)$, and we result in over-estimation. I believe you currently show this empirically, but some proof in a simplified toy setup may give the reader a better understanding of the dynamics.
> > >
> > > I hope the authors will add these to the final version. Other than that, the authors have addressed all my concerns, and I have increased the score from 5 to 6.

---

> > > > ### Author Response · Authors · 2023-11-23
> > > > **Thanks for your further response!**
> > > >
> > > > We are very pleased to have addressed most of your concerns! We hope to adequately address the remaining two questions as well.
> > > >
> > > > -------
> > > >
> > > > **Further Question 1:** Sorry for the late revision, we have added an extra section to the newly updated paper in Appendix K.8 for a discussion on the "set-level and instance-level definition of overestimation", which is also noticed with the blue color in Section 3.1 (following Definition 1). Here is a detailed discussion:
> > > >
> > > > Let's still start from the toy example in your first-round review, assuming there is an ID distribution $x\sim p_{id}(x)$, e.g., $\mathcal{N}(10,5)$, and also an OOD distribution $x\sim p_{ood}(x)$, e.g., $\mathcal{N}(9,5)$, we need to develop an ID-OOD classifier $D(x)$ to determine a data sample is OOD or not, specifically $D(x)=\text{ID}$ if $S(x)>T$ or $D(x)=\text{OOD}$ if $S(x)<T$.
> > > >
> > > > The core of designing an ideal ID-OOD classifier for unsupervised OOD detection is the choice of score function $S(x)$.
> > > > Under the context of generative models, we expect $S(x)$ to own the property $\mathcal{G}=\mathbb{E}\_{x\sim p_{id}(x)}[S(x)]-\mathbb{E}\_{x\sim p_{ood}(x)}[S(x)]>0$, so that most of the ID samples can be classified accurately after setting a threshold $T$.
> > > > We admit there will still be some ID/OOD samples to be misclassified even if $\mathcal{G}>0$ and the selection of $T$ can lead to varying degrees of misclassification, but from the perspective of the mean value of the data distribution, a larger $\mathcal{G}>0$ indicates that $S(x)$ tends to assign higher scores to most of the ID samples than that of OOD samples.
> > > >
> > > > The aforementioned content is the original statement in our first version of the manuscript and should be the non-strict case in your words.
> > > > Then, as for the stricter case you mentioned, we have $p(S(x)>T|x\sim p_{id}(x))$ and $p(S(x)>T|x\sim p_{ood}(x))$, and target of designing $S(x)$ is to make $p(S(x)>T|x\sim p_{id}(x))$ large but $p(S(x)>T|x\sim p_{ood}(x))$ small.
> > > > In other words, we need to make the confidence of classifying ID samples with ID labels larger than that of OOD samples, specifically $p(S(x)>T|x\sim p_{id}(x))>p(S(x)>T|x\sim p_{ood}(x))$, also equals to $p(S(x)>T|x\sim p_{id}(x))-p(S(x)>T|x\sim p_{ood}(x))>0$.
> > > >
> > > > At the **instance level, an ideal $S(x)$ is to make the lowest value of ID samples larger than the highest value of OOD samples.**
> > > > At the **set level** (or distribution level), an ideal $S(x)$ is to enlarge the distance between the distributions of ID and OOD samples, under the condition that the ID samples tend to own higher $S(x)$ scores.
> > > > In our work, we tend to focus on the second scenario to enlarge the distribution distance, which can be instantiated as the distance between mean values of distributions.
> > > > Thus, in our case, the stricter defination $p(S(x)>T|x\sim p_{id}(x))-p(S(x)>T|x\sim p_{ood}(x))>0$ can be also transfered into $\mathcal{G}=\mathbb{E}\_{x\sim p_{id}(x)}[S(x)]-\mathbb{E}\_{x\sim p_{ood}(x)}[S(x)]>0$ during the implementation.
> > > >
> > > > Thus, we assume that the following theoretical analysis, which is focused on the normal definition in the manuscript, can still hold even with the stricter definition you mentioned. To be more specific, the two factors, 1) the mismatch between $q(z)$ and $p(z)$ and 2) the gap in dataset entropy-mutual integration,  could still cause some OOD data samples to be scored higher than some ID data with the scoring function as the ELBO.
> > > >
> > > > **As for the instance-level theoretical analysis, we admit it remains a great challenge in the field of OOD detection, especially for unsupervised OOD detection with generative models. We are very willing to challenge this problem in the future.**
> > > >
> > > > ------
> > > >
> > > > **Further Question 2:**
> > > >
> > > > Thanks for your thoughtful suggestion! Yes, we highly agree that the suggested "mixture of Gaussian equipped with prior $\mathcal{N}(0, I)$" case could be very helpful to demonstrate the dynamic in resulting the overestimation. Actually, we conduct this toy experiment in part "When the design of prior is NOT proper?" of Section 3.2, where the ground truth $p(x)$ is a mixture of two Gaussians as shown in Figure 1(e).
> > > > In this case, the overestimation occurs with ELBO based on $p(z)=\mathcal{N}(0, I)$ in Figure 1(g) but could be alleviated by replacing $p(z)$ to $q(z)$ in Figure 1(h).
> > > > The following part "More empirical studies on non-linear VAEs for the improper design of prior" in Section 3.2 is to use a deep non-linear VAE to model the mixture of Gaussian, but the overestimation could still happen.
> > > >
> > > > We sincerely apologize for the poor organization of Section 3.2. We will emphasize the key ideas based on your suggestions and engage in a more comprehensive discussion of this case in the final version.

---

### Official Review · Reviewer_jWXz · 2023-10-31

**Soundness:** 3 good
**Presentation:** 3 good
**Contribution:** 3 good
**Rating:** 6
**Confidence:** 4

**Summary:**

The paper proposes a new anomaly score for OOD detection with VAEs: rather than use the ELBO, the paper proposes to
1. replace the prior $p(\mathbf{z})$ in the KL divergence term in the ELBO with the aggregated posterior $\hat{q}_{id}(\mathbf{z})$, which they call the post-hoc prior (PHP) method, and
2. add a term $\mathcal{C}(\mathbf{x}) = \mathbb{E}_{p_{id}}[PHP(\mathcal{x})] \frac{\mathcal{C}_{non}(\mathbf{x})}{\mathbb{E}_{p_{id}}[\mathcal{C}_{non}(\mathbf{x})]}$, which they call the dataset entropy-mutual calibration (DEC) method.

On the two most classic OOD detection failures for DGMs (i.e., FMNIST vs. MNIST, CIFAR-10 vs. SVHN), using just one of PHP or DEC succeeds on one benchmark but fails on the other, while using both PHP and DEC combined (which they call AVOID) succeeds on both benchmarks. On the Celeb-A vs. CIFAR-10 and Celeb-A vs. CIFAR-100 tasks, PHP, DEC, and AVOID perform better than other VAE-based detection methods. The authors also show that AVOID performs better than the ELBO on various OOD detection tasks with FMNIST, CIFAR-10, or Celeb-A as the in-distribution dataset.

**Strengths:**

1. Originality: To my knowledge, the proposed method is novel.
2. Quality: The experiments consider a wide range of baseline methods and ablations.
3. Clarity: The paper does a good job of breaking down the presentation particularly for factor 1, e.g. from analysis to proposed method.
4. Significance: The paper tackles the important question of understanding and improving failures in OOD detection by VAEs. In particular, I find it interesting that OOD detection with VAEs improves when substituting in the aggregate posterior for the prior in the ELBO. This result seems to provide a good example of how accounting for estimation error can improve OOD detection.

**Weaknesses:**

1. The motivation behind DEC is unclear, and its presentation is a bit circuitous. For instance, why is the non-scaled calibration function defined as it is in Eq 19, especially when $n_i \geq n_{id}$? There are many definitions that could satisfy property 1 (Eq 15). Also, if the point of the scaling is to be comparable to the Ent-Mut terms, why scale by Ent-Mut of the ID distribution plus a KL term rather than just the Ent-Mut term directly? It would help if, for instance, the authors can show that the performance is robust to various choices in DEC (or motivate the specific choices made).
2. The experiments show that PHP and DEC by themselves each only minimally improve OOD detection performance in some cases. In addition, the experiments only consider certain pairs but not their reverse (e.g., FMNIST vs. MNIST but not MNIST vs. FMNIST). Considering the latter can increase confidence that the proposed solution is not overfitting on a particular type of OOD detection task.

A few smaller comments:
1. The use of "counterfactual" in the third paragraph in the intro seem incorrect.
2. Eq 24 in the appendix looks wrong (though I think just due to typo; not something I noticed to affect any other part of the paper).

**Questions:**

1. (Repeated from above) Why is the non-scaled calibration function defined as it is in Eq 19, especially when $n_i \geq n_{id}$? There are many definitions that could satisfy property 1 (Eq 15). Also, if the point of the scaling is to be comparable to the Ent-Mut terms, why scale by Ent-Mut of the ID distribution plus a KL term rather than just the Ent-Mut term directly?
2. Do the authors have any hypotheses as to why PHP and DEC by themselves each only minimally improve OOD detection performance in some cases?
3. What are the results of this method on "reverse" dataset pairs (e.g. ID MNIST vs. OOD FMNIST)?

---

> ### Author Response · Authors · 2023-11-22
> **Thanks for the thoughtful insights and constructive comments!**
>
> We are deeply grateful for the time and effort you've dedicated to reviewing our work. Your thoughtful insights and constructive comments are immensely valuable to us!
>
> --------
>
> **W1 & Q1:**
> We much appreciate your deep thinking of the DEC method. Yes, we acknowledge that there could be many definitions that could satisfy property 1.
> Eq. 15 is only one of these definitions and is based on the SVD compressor. "$n_i \geq n_{id}$" means when a data sample is more complex than the in-distribution data measured by SVD, it would need more singular values to achieve the reconstruction error $|x_{recon}-x|<\epsilon$ with the SVD method. The "$(n_{id} - (n_i -n_{id}))/n_{id}$" guarantees that if and only if when $n_i=n_{id}$, $\mathcal{C}\_{\text{non}}(x)=1$, and for either $n_i>n_{id}$ or $n_i < n_{id}$, the $\mathcal{C}\_{\text{non}}(x)<1$. In that case, we could make sure the in-distribution data samples own a large $\mathcal{C}_{\text{non}}(x)$.
> We have also included the experiments of replacing the SVD method with other image compressors, JPEG and PNG compressors, to implement the DEC method in Appendix K.6.
>
> We acknowledge that DEC based on the data compressors could have limitations. For instance, it could have no effect in alleviating the overestimation when detecting a vertically flipped in-distribution data as OOD, in which case we could only rely on the PHP method as shown in Table 7.
>
> For concerns of "**why scale by Ent-Mut of the ID distribution plus a KL term**", we may guess it could be a misunderstanding since you cannot scale to Ent-Mut term directly, which is intractable.
> Recall $$\text{Ent-Mut}(\theta, \phi, p) = \mathcal{H}\_p(x)+\mathcal{I}\_{q}(x,z)-\mathcal{I}\_{q,p}(x,z)$$, to our knowledge, we cannot accurately estimate $\mathcal{H}\_p(x)$ without knowing the underlying ground truth distribution of a dataset. Besides, $\mathcal{I}\_{q}(x,z)-\mathcal{I}\_{q,p}(x,z)$ is an error term that related to the non-perfect VAE and this term is also unknown as we do not know the expression of the distribution of $p_\theta(z|x)$ in Eq. (5).
>
> However, we can estimate the Ent-Mut term with the PHP method. As explained in Eq. (20), when $q_{id}(x)$ could be estimated by $\hat{q}\_{id}(z)$, the expectation on the ELBO becomes the Ent-Mut term. We admit with your potential underlying concern about the KL term $D_{\text{KL}}[q_{id}(x)||\hat{q}_{id}(z)]$ may still not be equal to 0. Our analysis for the overestimation issue appeals for further advanced methods in estimating it, and there are already existing works focusing on it, [1, 2] claims that a high-dimension input data is actually supported on a low-dimension manifold of $z$ and we could estimate it well if we find a proper dimension for $z$. Thus, our analysis could pave the way for further improvement in VAE-based OOD detection methods.
>
> [1] Dai, Bin, and David Wipf. “Diagnosing and Enhancing VAE Models.” International Conference on Learning Representations. 2018
>
> [2] Loaiza-Ganem, Gabriel, et al. "Diagnosing and fixing manifold overfitting in deep generative models." arXiv preprint arXiv:2204.07172 (2022).
>
> **W2 & Q3:**
> Thanks for your suggestion and we have added the "**reverse experiments**" to Appendix K.2. Results show that our method could still work well in these cases.
>
>
> **Smaller comments**:
> Thanks for your careful reading and we have revised them in the paper noted with blue color.
>
> **Q2:**
> Yes, we could provide an explanation for it.
> For the PHP method, we add an experiment on Appendix K.1, where we can find the $q_{fashion}(z)$ is very distinguishable to $q_{mnist}(z)$ but the $q_{cifar10}(z)$ is not that distinguishable to $q_{svhn}(z)$, this could explain why PHP works better in FashionMNIST(ID)/MNIST(OOD) dataset pair but not that well in CIFAR10(ID)/SVHN(OOD) dataset pair. This provides an insight that learning a dataset-specific semantic latent space could achieve better OOD detection performance with the PHP method.
>
> For the DEC method, we have previously conducted a "VFlip" experiment in Table 7 of Appendix F. In this experiment, the OOD data is set as the vertically flipped in-distribution data. Thus, the DEC method based on compressors cannot work at all, but our method could still rely on the PHP method to achieve promising performance on OOD detection.

---

### Official Review · Reviewer_G5ek · 2023-11-01

**Soundness:** 2 fair
**Presentation:** 4 excellent
**Contribution:** 3 good
**Rating:** 6
**Confidence:** 4

**Summary:**

This paper addresses the problem of generative models, specifically VAEs, when applied to OOD detection tasks. It is based upon the observation that the ELBO used in VAEs, even though it is a reasonable candidate, is an unreliable metric for OOD detection. Moreover, it tries to change the metric to come to a more reliable score to perform OOD detection.

The paper breaks down the ELBO of a dataset (being in- or out-of-distribution) into two components: (i) a negative KL divergence between the aggregate posterior $q(z)$ and the prior $p(z)$, and (ii) a negative term related to dataset entropy and mutual information between inputs and latent variables.
It identifies the cause of ELBO's poor performance in OOD detection, noting that the KL divergence in (i) is often overestimated and the negative entropy term in (ii) is often inflated for simpler datasets that we perform OOD detection on (for example doing OOD detection on MNIST when a model is trained on FashionMNIST). The work proposes a post-hoc correction to adjust the former (PHP) and it introduces a method (DEC) to correct the small OOD entropies by utilizing a complexity measure inspired by Serra et al. (2020).
Implementing these adjustments successfully improves OOD detection for VAEs in challenging scenarios, such as differentiating between datasets like FashionMNIST and MNIST, or CIFAR10 and SVHN.

All in all, the study tries to address an intriguing problem, but at this stage, I cannot accept it and I will explain why in the following. I would be willing to increase my score to above the acceptance threshold if the authors can address all the important issues and suggestions that I have written in the following.

**Strengths:**

1) The paper is well-written and easy to follow.
2) The problem of DGMs failure in OOD detection is intriguing and has been observed not only in VAEs, but almost all the likelihood-based deep generative models. Therefore, any contribution in this field is valuable.
3) Breaking down the ELBO term is interesting from a theoretical standpoint. Although this observation is not entirely novel, as I will explain in the weaknesses, the theory is certainly sound and the method improves the OOD detection performance by large for the tasks it has considered.
4) The toy examples are very informative and interesting.

**Weaknesses:**

1) **(Important)** The OOD detection pathology is one-sided, meaning that it happens when you train a model on a relatively complex dataset and test it for OOD on a simpler one. However, it usually does not hold the other way around. An important feature that a good OOD detection method should have is that even though it fixes the pathological direction, it does not hinder the performance in the reverse direction. That being said, please provide the results when running the framework in the reverse direction where a VAE is trained on MNIST and tested for OOD on FashionMNIST. Similarly, when a model is trained on SVHN and tested for OOD on CIFAR10. It is well-known that many such methods hinder the performance in the other direction, one would need to make sure this does not happen for the current algorithm.
2) **(Important)** Although the entropy and KL divergence breakdown is touted as novel, it is not entirely novel! Caterini et al. (2022) have considered breaking down the likelihood term into an entropy and a KL divergence term. In fact, in the special case where ELBO equals the likelihood and the encoder and decoder provide perfect mappings, the mutual information terms cancel out and the entropy term is pointed out in Caterini et al. (2022). However, this paper is not mentioned at all. It should most certainly be added to the next iteration of the paper.
3) The PHP method needs to train an entirely new model which can be time-consuming. Ideally, your generative model already has a good understanding of what in-distribution means and should be able to perform OOD detection even without additional training.
4) The DEC method is tested specifically on image data and the SVD-based algorithm also seems like a sort of “outside help”. Even though similar methods have been proposed in the past to fix the entropy term, such as Serra et al. (2020), I still believe that the DGM should already have the information required for OOD detection without the need for any extra model training or running modality dependent algorithms.
5) Performing OOD detection requires adjusting an $n_{id}$ hyperparameter which is task-dependent, and there is no guarantee that one setting of this hyperparameter generalizes to all.
6) Please cite the relevant study by Schirrmeister et al. (2020) on the reason behind the OOD detection anomaly for invertible networks.
7) This study only considers VAEs. It would be interesting to see how the method acts in a broader context when a latent space is involved. For example, latent diffusion models are such examples. Even though other generative models might be out of the scope of the paper, it should be pointed out as a limitation of this study.

**References**

Caterini, Anthony L., and Gabriel Loaiza-Ganem. "Entropic issues in likelihood-based ood detection." I (Still) Can't Believe It's Not Better! Workshop at NeurIPS 2021. PMLR, 2022.

Schirrmeister, Robin, et al. "Understanding anomaly detection with deep invertible networks through hierarchies of distributions and features." Advances in Neural Information Processing Systems 33 (2020): 21038-21049.

**Questions:**

1) **(Important)** Although the PHP method is quite novel, it reminds me of the studies where an extra flow model is trained on the latent space of a VAE to alleviate the bias that the aggregate posterior should be Gaussian. Loaiza-Ganem et al. (2022) (which has not been referenced in this paper) show that a VAE model that has been trained again with a flow on top can perform much better in OOD detection. Could you provide some discussion on the connections between your study and theirs? And how yours is novel?

2) **(Important)** Could you please generate a figure similar to Figure 3 for the CIFAR10 (vs) SVHN OOD detection task where the PHP method does not improve the performance by large?

3) The computation of $C_{non}$ seems rather contrived! I might have missed it but what is the rationale behind the formula when $n_i \ge n_{id}$? Why not choose a simple fix such as the compression scores proposed by Serra et al. (2020)? I am referring to an ensemble of FLIF, JPEG, and PNG compressions and computing the bit count for the compressed images.  Also, can I ask for a runtime analysis of computing $C(x)$? This is more of a suggestion, but I believe you can improve algorithm 1 (in Appendix D) for computing $n_i$ by performing a binary search rather than iterating over all the possible values. Since the number of potential singular values $N$ can be as high as the number of pixels, it is important to be efficient.

**References**

Loaiza-Ganem, Gabriel, et al. "Diagnosing and fixing manifold overfitting in deep generative models." arXiv preprint arXiv:2204.07172 (2022).

---

> ### Author Response · Authors · 2023-11-22
> **(Part 1) Thanks for the valuable comments!**
>
> We sincerely appreciate the time and effort you've put into reviewing our work. Your comments are incredibly valuable to us! We have taken your concerns into consideration and hope our responses effectively address your concerns.
>
> -------------------
>
> **W1:**
> We absolutely agree with your suggestion that an ideal OOD detection model should be bidirectionally effective.
> Actually, the property you mentioned is one of the advantages of our method and we highlight the results of bidirectional comparison has been included in **Figure 5(c-d)** of first-time submission, indicating that our method can still be effective in detecting FashionMNIST as OOD when trained on MNIST as you suggest.
> On the contrary, the previous SOTA method $\mathcal{LLR}^{ada}$ tends to perform even worse than the basic OOD method with ELBO.
> To make a more comprehansive comparison, we have added more experimental results to demonstrate our method's **bidirectional effectiveness in the Table 12** of Appendix K.2.
> The key insight is that, unlike baselines like HVK and $\mathcal{LLR}^{ada}$, the PHP and DEC are calibrations on the ELBO directly, which could improve or at least not hurt the performance of applying ELBO. Thus, when ELBO could already work well in some cases, our method could also perform well.
>
> **W2:**
> Thanks for your recommendation of this interesting paper (Caterini et al. [1]) and we will cite this paper in our revision.
>
>
> Firstly, we admit that the paper you recommend is relevant to ours, but we also need to highlight that **the breakdown target of our method is different from the paper [1]**, which will be discussed below:
>
> 1) Caterini et al. breakdown the expectation on the **log-likelihood $\log p_\theta(x)$** into two terms
> $$\mathbb{E}\_{x\sim P}[\log p_\theta(x)]=-D_{\text{KL}}[P||P_\theta]-\mathcal{H}[P]$$, which is not specified to VAEs.
>
> 2) In our work, we directly breakdown the expectation on the **ELBO (lower bound of $\log p_\theta(x)$)** as shown in Eq. (7):
> $$\mathbb{E}\_{x\sim P}[\text{ELBO}(x)]=-D_{\text{KL}}[q(z)||p(z)] - [\mathcal{H}[P]+\mathcal{I}\_q(x,z)-\mathcal{I}\_{q,p}(x,z)]$$
> , which provides a lens to see what is happening in $D_{\text{KL}}[P||P_\theta]$ and thus operatable methods would be inspired.
>
> Additionally, we want to show our sincereest gratitude for your insightful thinking of our paper as you claim the **"special case"** where ELBO equals the likelihood.
> Yes, it is correct that $\text{ELBO}(x)=\log p_\theta(x) =\log P_{ID}(x)$ when there is a perfect generative model to mimic the data distribution, and there could be
>
> $$\mathbb{E}\_{x\sim P_{ID}}[\log p_\theta(x)]-\mathbb{E}\_{y \sim P_{OOD}}[\log p_\theta(y)]$$
> $$=\mathbb{E}\_{x\sim P_{ID}}[\text{ELBO}(x)]-\mathbb{E}\_{y \sim P_{OOD}}[\text{ELBO}(y)]$$
> $$=[-D_{KL}(P_{ID}(x)||p_\theta(x))-\mathcal{H}(P_{ID})] - [-D_{KL}(P_{OOD}(y)||p_\theta(y)) - \mathcal{H}(P_{ood})]$$
> $$=-\mathcal{H}(P_{ID}) + \mathcal{H}(P_{OOD}) + D_{KL}[P_{OOD}(y)||p_\theta(y)].$$
>
> In that case, we agree with you that calibration on the entropy term could further improve the OOD detection performance.
>
> Paper [1] also provides insights of the likelihood-ratio methods for removing the influence of entropy term, but that relies on an assumption of a perfect VAE model satisfying $P_\theta=P_{ID}$ and find a reference model $R_\phi$ to satisfy $D_{KL}[P_{OOD}||R_\phi]\leq D_{KL}[P_{ID}||R_\phi]$.
>
> However, we feel very sad to emphasize that **acquiring a perfect model $P_\theta=P_{ID}$ could be difficult and even impossible** for a long time in the future though it is theoretically possible [2,3].
> Therefore, further **breaking down the $D_{\text{KL}}[P||P_\theta]$ for ELBO-based methods is extreamly necessary**.
>
> In summary, **the novelty of our analysis is to inspire an operatable way to alleviate the overestimation issue with a theoretical guarantee for a non-perfect ELBO-based generative model, e.g., VAE,**
> which could be shorted as:
> 1) We could further investigate which factor will lead to a bad $D_{\text{KL}}[P||P_\theta]$, i.e., the mismatch between the aggerated posterior $q(z)$ and prior $p(z)$ or the mutual information gap $\mathcal{I}\_q(x,z)-\mathcal{I}\_{q,p}(x,z)$;
> 2) Our analysis provides a tracable way to fix the overestimation of VAEs even when it is not a perfect model to estimate $P_{ID}$, i.e., using the PHP method to replace the $p(z)$ with approximated $q_{id}(z)$ and DEC method to calibrate the Ent-Mut term.
>
>
> [1] Caterini, Anthony L., and Gabriel Loaiza-Ganem. "Entropic issues in likelihood-based ood detection." I (Still) Can't Believe It's Not Better! Workshop at NeurIPS 2021. PMLR, 2022.
>
> [2] Dai, Bin, and David Wipf. "Diagnosing and Enhancing VAE Models." International Conference on Learning Representations. 2018.
>
> [3] Dai, Bin, Li Kevin Wenliang, and David Wipf. "On the Value of Infinite Gradients in Variational Autoencoder Models." Advances in Neural Information Processing Systems. 2021.

---

> > ### Author Response · Authors · 2023-11-22
> > **(Part 2)**
> >
> > **W3:** Thanks for your comments! we suppose that the two points you mentioned are not in conflict.
> > We agree with the point that an ideal generative model should be able to perform OOD detection without additional training, and it can already use ELBO or likelihood to achieve the perfect performance of OOD detection.
> > However, in practice, there is currently no such a perfect generative model.  We admit that existing imperfect generative models can already use ELBO or likelihood to achieve not-so-bad OOD detection performance, but there are still requirements to develop a series of post-hoc methods to further improve the performance with these generative models.
> >
> > As for the computation cost of the PHP method, we need to highlight that the new model is trained to model the aggregated posterior of latent representations, whose dimension will be much smaller than the dimension of raw features. Thus, we suppose the PHP method will not bring too many computational burdens.
> > The experiments for the computation efficiency regarding the training time and inference time are added to Appendix K.5.
> >
> > **W4:**
> > Refer our response to weakness 3, we agree with your point that "the DGM should already have the information required for OOD detection without the need for any extra model training or running modality-dependent algorithms".
> > However, the post-hoc method can be used as an effective and straightforward remedy to improve the OOD detection performance of these imperfect generative models.
> >
> > **W5:**
> > Thanks for your thoughtful comment! We evaluate the effect of different settings of $n_{id}$ in Figure 6(c), where the different settings of $n_{id}$ could generally alleviate the overestimation issue and improve the OOD detection performance.
> > We admit the best $n_{id}$ will change from task to task. However, for a given ID dataset, the value of $n_{id}$ will be fixed and will not change as the OOD dataset changes. Thus, in practice, after setting the value of $n_{id}$ for a given ID dataset, our method can be applied to various OOD datasets.
> > Note that SVD is only one of the choices for DEC, and implementing the DEC with other compressors like PNG and JPEG could avoid this concern, for which we show the corresponding performance in Table 20 of Appendix K.6.
> >
> > **W6:**
> > Thanks for the notification, we have cited and discussed this work in the revised paper.
> >
> >
> > **W7:**
> > Thanks for your suggestions, we have included the experiments of applying our methods on a latent diffusion model in Table 17 of Appendix K.4.
> > As the experimental results shown in Table 17, we can find that our method can still work well on other generative models with more complex architectures.
> >
> > **Q1:**
> > Thanks for recommending this paper [4] to us and we will include a discussion of the relevance between this paper and our work in the revision!
> > In our consideration, this paper provides an interesting analysis of the failure of likelihood-based models from a perspective of manifold overfitting.
> > Manifold overfitting is an issue that occurs when using a generative model to estimate the input data likelihood in high-dimensional space, i.e., the dimension of the inputs. It hinders recovering the true data distribution $P^*$ that is actually supported on a low-dimension manifold.
> > More interestingly, it is possible to achieve large likelihoods while being close to any $P^+$, rather than to $P^*$.
> > Therefore, the authors [4] propose a two-step correctness theorem to alleviate the influence brought by the manifold overfitting. Specifically, the input data is firstly projected to a low-dimension representation $z$ and secondly, we apply a generative model to learn the low-dimension representation.
> >
> > For its relationship to our method, we admit that it is similar to the core idea behind the PHP method, where the mismatch between $q(z)$ and $p(z)$ could bring VAE's overestimation issue.
> > However, our analysis reveals that **the mismatch between $q(z)$ and $p(z)$ is not the only factor** that could contribute to the overestimation, where the Ent-Mut term could also lead to the overestimation. Therefore, **our analysis and methods could be seen as a more comprehensive remedy to the VAE's overestimation issue**.
> >
> > By the way, we need to highlight that we have cited a similar theoretical paper [2] that also proposes to use a two-step approach to learn a better VAE for OOD detection.
> >
> > ([2] Dai, Bin, and David Wipf. "Diagnosing and Enhancing VAE Models." International Conference on Learning Representations. 2018.)
> >
> > **Q2:**
> > Thanks for your suggestion! We have included the UMAP visualization of latent representations on FashionMNIST(ID) /MNIST(OOD) and CIFAR-10(ID)/SVHN(OOD) dataset pairs as shown in Fig.9 of Appendix K.1.
> > From the visualization results, the intuition that the PHP method does not improve the performance by large could due to the latent representation of CIFAR and SVHN is not that distinguishable than that of FashionMNIST and MNIST.

---

> > > ### Author Response · Authors · 2023-11-22
> > > **(Part 3)**
> > >
> > > **Q3:**
> > > Thanks for your suggestion! we set a maximum value of $n_i$ as 2*$n_{id}$, which avoids heavy computation of $\mathcal{C}\
> > > _{\text{non}}$. For the computation efficiency analysis, we add an experiment in Appendix K.5, where the DEC method is implemented in various ways including a binary search version of SVD, JPEG-based DEC, and PNG-based DEC method.
> > > Besides, for OOD detection performance of DEC based on different compressors is testified in Table 20 of Appendix K.6.
> > >
> > >
> > >
> > > ------
> > >
> > > We sincerely appreciate your insightful comments, which have significantly contributed to improving the quality of our work. Recognizing the vital importance of your evaluation, we hope that our responses and the detailed experiments presented in Appendix K adequately address your concerns, particularly those highlighted as "Important". Should there be any additional issues or questions, we would be more than glad to address them.

---

> ### Comment · Reviewer_G5ek · 2023-11-23
> **Thank you!**
>
> I appreciate the considerable effort the authors have put into revising the manuscript. My primary concerns, identified as (W1) and (W2), have been adequately addressed. Additionally, the revised compression algorithms for estimating DEC are more practical compared to the previous SVD-based method, which necessitated setting a $n_{\text{id}}$ for each model. Consequently, I will raise my review score.
>
> The addition of experiments involving diffusions is a noteworthy enhancement. I suggest, for further improvement, to consider formulating a diffusion as a probability flow ODE, rather than a VAE. This approach enables the actual computation of data likelihood $ p_\theta(\cdot)$ as mentioned in Song et al. [A]. For the comparisons in Table 17 of your final paper version, it would be beneficial to include this likelihood value as an OOD score and compare it against your method as a baseline. Please check [B] for reference, they perform the likelihood ratio on top of the ODE formulation of the diffusions. Although the model in Goodier and Campbell [B] differs from yours, they do report significantly higher AUC-ROC values.
>
> **References:**
>
> [A] Song, Yang, et al. *Maximum likelihood training of score-based diffusion models*. Advances in Neural Information Processing Systems 34 (2021): 1415-1428.
>
> [B] Goodier, Joseph, and Neill DF Campbell *Likelihood-based Out-of-Distribution Detection with Denoising Diffusion Probabilistic Models.* arXiv preprint arXiv:2310.17432 (2023).

---

> > ### Author Response · Authors · 2023-11-23
> > **Thanks for your further response!**
> >
> > We are very pleased to have addressed your primary concerns!
> >
> > We are grateful for the insightful suggestion to enhance our experiments with diffusion models. We will formulate a diffusion as a probability flow ODE for likelihood calculation and compare it with our methods in the next revision. Additionally, we will incorporate a wide range of baselines, including the proposed likelihood ratio method built on top of the ODE formulation of the diffusions [B]. We promise to include these experiments in the final version of our paper.

---

### Official Review · Reviewer_ms1o · 2023-11-01

**Soundness:** 3 good
**Presentation:** 3 good
**Contribution:** 3 good
**Rating:** 6
**Confidence:** 4

**Summary:**

The authors propose a new approach to address overestimation in unsupervised Out-Of-Distribution (OOD) detection using Variational Autoencoders (VAEs). In their investigation, they found two main factors that contribute to overestimation in OOD, (1) improper design of the prior, and (2) gap in entropy-mutual integration between in-distribution and out-distribution. The proposed approach uses a new score function to address the two problems.
The paper presents extensive experiments to validate the effectiveness of the proposed method, suggesting a competitive performance compared to literature methods. An ablation study is also presented to evaluate the behavior of the main components of the proposed approach: post-hoc prior and dataset entropy-mutual calibration.

**Strengths:**

I found the performed evaluation of the factors that cause overestimation in OOD detection very interesting. The unsupervised approach using VAEs seems promising and robust.
The authors also designed a comprehensive and extensive set of experiments, including ablation studies, to evaluate the contributions of individual components of the proposed method.
The paper is well-organized, I liked the approach of breaking down the problem, its causes, the solution, and the experimental validation in a logical sequence.
I believe that OOD detection represents a significant challenge in the field of machine learning. This is especially true for safety-critical applications as we are every day more dependent on automatic decision-making.

**Weaknesses:**

Besides the large number of experiments, I think the main experiments are centered on or in variations of specific datasets like FashionMNIST and CIFAR-10.

I agree with the authors that the ablation study provides insights, but I don't think the contributions of the PHP and DEC components are still clear, especially when combined.

The authors should clearly address the computational efficiency of the proposed method compared to standard VAEs and other literature approaches to the problem.

I don't like to rely on t-SNE plots to make claims about the differences between in-distribution and out-of-distribution data.  While it is a popular choice for high-dimensional data visualization, I think UMAP offers several advantages as it preserves more of the global structure of the data and is more reproducible as it offers more intuitive hyperparameters.

I was wondering, as the authors separate ensemble methods from non-ensemble methods in the "Unsupervised" category, it would be interesting to understand how the proposed method performs against ensemble methods. I was considering the performance in terms of performance and computational cost.

I do believe the authors benefit from a discussion on scenarios where the proposed method might not work well. This gives readers a more balanced view and sets expectations correctly. I suggest the authors explore more of that.

**Questions:**

Would the method work with more complex architectures, or is it specifically tailored to standard VAEs?

Have the authors evaluated the robustness of the proposed idea against adversarial examples or noisy datasets?

How does overestimation influence real-world decisions or systems that rely on OOD detection? I think contextualizing that in the paper will help the reader to understand better the proposed approach.

---

> ### Author Response · Authors · 2023-11-22
> **Thanks for your valuable suggestions!**
>
> Thanks for your valuable suggestions! To better address your concerns about our method, please allow us to interchange the order of weaknesses in your comments.
>
> **W1: UMAP Visualization**
>
> Thanks for the suggestion of replacing t-SNE with UMAP visualization, and we have included the UMAP visualization of latent representations on FashionMNIST(ID) /MNIST(OOD) and CIFAR-10(ID)/SVHN(OOD) dataset pairs as shown in Fig.9 of Appendix K.1.
>
> **W2: When would the method not work well**
>
> Combining the results shown in Table 2 and also the visualization of latent representations in Fig.9, we can find that the PHP method works well in FashinMNIST(ID)/MNIST(OOD), but does not work well in CIFAR-10(ID)/SVHN(OOD).
> The underlying reason is that the aggregated posterior q(z) of FashionMNIST is quite distinguishable from that of MNIST, but there is a significant portion of the regions overlap between the aggregated posteriors of CIFAR-10(ID) and SVHN(OOD), which leads to the performance gains brought by PHP method is not very noticeable as shown in Table 2.
>
> For the DEC method, it will not work well when the image complexity of ID/OOD datasets is similar, e.g. FashinMNIST(ID)/MNIST(OOD) shown in Table 2 and a special case in Table 7. For the special case of Table 7 that we detect the vertically flipped in-distribution data as OOD data, the DEC method based on SVD or other image compressors does not work, but we can still rely on the PHP method to work in this special case.
>
> **W3: Contribution of PHP and DEC**
>
> Continue analyzing the experimental results in Table 2, for CIFAR-10(ID)/SVHN(OOD), we can find that the DEC method can bring about an improvement in OOD detection performance, especially when PHP is not working well.
> Conversely, for FashinMNIST(ID)/MNIST(OOD), when the DEC method is not working well because of the relatively similar image complexity between ID/OOD datasets, the PHP method can bring about performance benefits.
>
> Therefore, the complementarity between PHP and DEC methods can ensure that our combined approach works well in the majority of OOD detection scenarios.
>
> **W4: Computational efficiency of the proposed method**
>
> Thanks for your suggestion! We need to emphasize that both PHP and DEC methods are post-hoc methods with trained VAEs, which will not cause too many additional computation burdens.
> To make an intuitive efficiency comparison with performing OOD detection on VAEs with ELBO, we testify to the computation efficiency from two perspectives: training time and inference time, in Table 18 and Table 19 of Appendix K.5.
> Results show that the average computation time in scoring a data point on an A100 GPU is approximately 1.5x computation time than vanilla ELBO, which is still a very small number and could be much faster if we change the SVD to other image compressors or a binary search version. The additional time for training the LSTM of the PHP method is also very fast, taking about 30 min to converge.
>
> **W5: Comparison with ensemble methods**
>
> Thanks for your suggestions, we reproduce the ensemble method WAIC with an ensemble of 5 VAEs to test their performance (added to Table 2) and computation efficiency (in Table 18 and Table 19 of Appendix K.5).
>
> **Q1: Complex architecture**
>
> To demonstrate that our method can also work well on generative models with more complex architectures, we have included the experimental results of applying our method to the recently popular diffusion model in Table 17 of Appendix K.4.
> From the results, we can find that our method still works well on OOD detection with trained diffusion models, demonstrating the generalizability of our approach. More experimental details can be found in K.4.
>
> **Q2: Robustness on noisy data**
> To evaluate the robustness of the proposed method on noisy datasets, we manually introduce varying scales of random noise into input data samples to create OOD datasets at different noise levels. Then, we apply our method to these noisy datasets and obtain the experimental results as shown in Table 21 and Table 22 of Appendix K.7. We can find that our method still works well on these noisy datasets.
>
> **Q3: Influence on real-world decisions or systems that rely on OOD detection**
>
> A real-world use case could be autonomous driving we hope to identify whether the current driving state is strange to the training set for ensuring the safety of decision-making.
> In this case, we should give lower confidence to the strange (OOD) states, otherwise, incorrect high confidence may hurt the safety.
> Actually, this could be the setting of offline RL, where a core direction of research focus is to identify the out-of-distribution state-action pairs and assign them a lower reward [1]. Moreover, as the RL setting always has a lot of states, e.g., millions of steps, it could be impossible to label every state, which appeals to unsupervised OOD detection.
>
> [1] Fujimoto et al. "Off-policy deep reinforcement learning without exploration." ICML, 2019.

---

### Author Response · Authors · 2023-11-22
**For all reviewers**

We apologize for the delay in our response because we have made every effort to conduct as many of the (requested) additional experiments as possible. These experiments aim to further support our analysis of the overestimation issue in **ELBO-based** OOD detection and the advantages of our proposed method.

We've included all these experiments in **Appendix K ("Rebuttal")** of the revised paper, summarized as follows:
1) For Reviewer **ms1o**, **G5ek**, **jWXz**, **F6GH**: in Appendix K.1, we demonstrate the insight for why the PHP method does not work that well in the CIFAR10(ID)/SVHN(OOD) case with UMAP (Uniform Manifold Approximation and Projection) visualization;
2) For Reviewer **G5ek**: in Appendix K.2, we testify our methods' robustness in a reverse direction, i.e., MNIST (ID) / FashionMNIST (OOD) and SVHN (ID) / CIFAR-10 (OOD);
3) For Reviewer **F6GH**: in Appendix K.3, we testify our methods' OOD detection performance with a model trained on the CIFAR-100 dataset;
4) For Reviewer **ms1o, G5ek, jWXz**: in Appendix K.4, we testify our method's OOD detection performance with a more complex backbone, a latent diffusion model;
5) For Reviewer **ms1o, G5ek**: in Appendix K.5, we analyze the computation efficiency about our methods and ensemble methods reagrding the training time and inference time;
6) For Reviewer **G5ek, jWXz**: in Appendix k.6, we testify the implementation of DEC method with other two image compressors, i.e., JPEG and PNG compressor;
7) For Reviewer **ms1o**: in Appendix K.7, we testify the noisy setting, which demonstrates the robustness of our method and could support our analysis about the Ent-Mut term.

We hope these additional experiments and our responses can address each of your concerns about our work.
We acknowledge that the breadth of our additional experiments might impose a significant review burden, especially given the constrained timeline of the rebuttal period. We sincerely apologize for any inconvenience this may cause and sincerely appreciate your time and efforts in evaluating our work.

---

### Meta-Review · Area_Chair_h9Qc · 2023-12-08

**Metareview:**

This paper deals with the phenomenon that deep generative models tend to assign unintuitive likelihood scores to out-of-distribution data.  One would expect a generative model to provide an ideal score function for OOD data, namely p(X), the likelihood that the data comes from the training distribution given the model.  However, in practice researchers have observed that OOD data can often even have a higher likelihood than in-distribution data.  The authors focus on variational autoencoders here, and note that the ELBO should be a strong score function - i.e. a lower bound on the data likelihood marginalized over model parameters.  However, the ELBO also exhibits this issue with overestimation of the likelihood of OOD data.  The authors suggest that this discrepancy comes from the prior in the model and a gap in entropy and mutual information terms.  The authors develop an alternative score to mitigate these issues and demonstrate improved performance on a variety of standard vision based OOD benchmarks.

The reviewers unanimously gave a score of 6, which is a borderline accept.  One reviewer felt that there were some significant issues with the paper (.e.g the claims were too strong given previous literature) but this seems to have been resolved during the discussion phase.  In terms of strengths, the reviewers found the paper well written, technically sound and found the presented experiments thorough.  In terms of weaknesses, the reviewers seemed to find the problem tackled somewhat narrow, i.e. it only applies to VAEs and evaluation was only done on vision tasks.

There has been a massive resurgence in generative models recently, and dealing with OOD data remains a major challenge.  Therefore the general motivation seems strong and this could be quite interesting to the community.  Unfortunately, the focus on VAEs and use of only synthetic vision benchmarks seems to really undersell the impact.  That might explain why reviewers were quite reserved in their scores, despite that they viewed the work as technically correct.  The experiments all seem like proof of concept (MNIST, CIFAR) when there are quite exciting real OOD problems and benchmarks, particularly for generative models.  Therefore, the recommendation is to reject. Hopefully the reviews will be helpful in strengthening the paper for a future submission.

**Justification For Why Not Higher Score:**

I wouldn't be upset if the paper was bumped up. However, I think we should draw a line in the sand with just toy experiments.  There are a variety of more interesting benchmarks for OOD detection in generative models.  If the theoretical contribution was really insightful or strong I'd be ok with proof of concept experiments, but otherwise I think we need to move past MNIST, CIFAR.

**Justification For Why Not Lower Score:**

NA

---

### Decision · Program_Chairs · 2024-01-16

Reject